# Proteomic analysis of archival breast cancer clinical specimens identifies biological subtypes with distinct survival outcomes

Karama Asleh [1,2,9], Gian Luca Negri [3,9], Sandra E. Spencer Miko [3], Shane Colborne[3], Christopher S. Hughes [4], Xiu Q. Wang[1], Dongxia Gao[1], C. Blake Gilks[5,6], Stephen K. L. Chia[7], Torsten O. Nielsen [1,5] & Gregg B. Morin [3,8 ✉]

Despite advances in genomic classification of breast cancer, current clinical tests and treatment decisions are commonly based on protein level information. Formalin-fixed paraffin-embedded (FFPE) tissue specimens with extended clinical outcomes are widely available. Here, we perform comprehensive proteomic profiling of 300 FFPE breast cancer surgical specimens, 75 of each PAM50 subtype, from patients diagnosed in 2008-2013 (n = 178) and 1986-1992 (n = 122) with linked clinical outcomes. These two cohorts are analyzed separately, and we quantify 4214 proteins across all 300 samples. Within the aggressive PAM50-classified basal-like cases, proteomic profiling reveals two groups with one having characteristic immune hot expression features and highly favorable survival. Her2-Enriched cases separate into heterogeneous groups differing by extracellular matrix, lipid metabolism, and immune-response features. Within 88 triple-negative breast cancers, four proteomic clusters display features of basal-immune hot, basal-immune cold, mesenchymal, and luminal with disparate survival outcomes. Our proteomic analysis characterizes the heterogeneity of breast cancer in a clinically-applicable manner, identifies potential biomarkers and therapeutic targets, and provides a resource for clinical breast cancer classification.

[1] Genetic Pathology Evaluation Centre, Department of Pathology and Laboratory Medicine, University of British Columbia, Vancouver, BC, Canada. [2] Interdisciplinary Oncology Program, Faculty of Medicine, University of British Columbia, Vancouver, BC, Canada. [3] Canada's Michael Smith Genome Sciences Centre, BC Cancer Research Institute, University of British Columbia, Vancouver, BC, Canada. [4] Department of Molecular Oncology, BC Cancer Research Institute, University of British Columbia, Vancouver, BC, Canada. [5] Division of Anatomical Pathology, Vancouver General Hospital, University of British Columbia, Vancouver, Canada. [6] Canadian Immunohistochemistry Quality Control, University of British Columbia, Vancouver, BC, Canada. [7] Division of Medical Oncology, British Columbia Cancer Centre, University of British Columbia, Vancouver, BC, Canada. [8] Department of Medical Genetics, University of British Columbia, Vancouver, BC, Canada. [9] These authors contributed equally: Karama Asleh, Gian Luca Negri. ✉email: gmorin@bcgsc.ca

The genomic classification of breast cancers into intrinsic subtypes has remarkably advanced breast cancer diagnosis and further refined prognosis and prediction of patients' outcomes[1–4]. This achievement has been largely attributed to the higher precision of these classifications over the in-practice commonly used immunohistochemical (IHC) methods that are semi-quantitative and measure only a few biomarkers.

Genomic classifications of breast cancer, such as the PAM50 RNA-based gene signature[4,5], are increasingly used as a gold-standard to identify intrinsic breast cancer subtypes and recommend biomarkers for clinical use[6,7]. Importantly, these classifications do not always guide therapeutic choices, due to the extensive heterogeneity that still characterizes breast cancers beyond their DNA or RNA profiles, especially within the aggressive subgroup of triple-negative breast cancer (TNBC)[8–12]. In addition, DNA alterations or changes in RNA expression do not always translate into protein expression patterns that display the biological changes in cell function at the level where targeted therapies and clinical diagnostic tests work[13].

Newer classifications based on protein expression profiling have been proposed to more reliably reveal the functional phenotypic differences that underpin breast cancer heterogeneity. Large-scale proteomic characterization of breast cancer in the Cancer Genome Atlas (TCGA) was first attempted using reverse phase protein arrays (RPPA) to quantify the expression of 171 cancer-related proteins and phosphoproteins[8]. This study recapitulated the presence of the four main mRNA-based subtypes of breast cancer and described two additional distinct subgroups (reactive I and II) characterized by stromal features. The RPPA analysis provided key insights into the heterogeneity of breast cancer at the protein level, but it had a restricted number of proteins and the inherent difficulties of antibody-based quantification. To classify breast tumors in a more in-depth and comprehensive manner, the Clinical Proteomic Tumor Analysis Consortium (CPTAC) performed a mass spectrometry (MS) based analysis using fresh frozen materials from 77 TCGA breast cancer specimens, demonstrating the presence of three distinct proteome subgroups[14]. Two subgroups recapitulated the luminal and basal PAM50 RNA subtypes, while an additional stromal enriched subgroup was a mixture of PAM50 subtypes and was highly concordant with a RPPA protein-defined subgroup displaying stromal features. A recent CPTAC analysis of 122 TCGA breast cancer specimens described the presence of four proteome subgroups that correlated with their PAM50 subtypes, but also illustrated heterogeneity within luminal A, luminal B and Her2-Enriched PAM50 assignments[15]. Another MS-based study that analyzed nine breast cancer specimens from each of the four PAM50 subtypes also delineated heterogeneity in breast cancers, describing six proteome subgroups that partially recapitulated the PAM50 subtypes but subdivided the basal subtype into two distinct subsets[16].

While earlier studies provided high-quality analytical data to classify breast cancers, the limited number of cases analyzed, and the lack of clinical outcome association, is insufficient to characterize the true biological heterogeneity of breast cancers in relation to clinical behavior and treatment response. Furthermore, the relatively large amount of fresh-frozen tissue required in these studies is not typically available from patients, restricting the application of these methods in the clinical setting and precluding application to collections of sufficient age to provide extended survival outcome data. Currently, most biobanks do not maintain archived frozen materials with sufficient follow-up for conducting prognostic studies[17]. Thus, an approach that is compatible with standard formalin-fixed paraffin-embedded (FFPE) clinical specimens with linked long-term outcome data would provide information more easily translated into clinical actionable tests.

In this work, we utilize our highly sensitive MS-based methodology termed Single-Pot, Solid-Phase-enhanced, Sample Preparation-Clinical Tissue Proteomics (SP3-CTP) capable of capturing biological features in FFPE tumor samples[18–20]. This method can be used to query large FFPE material cohorts linked to outcome data, enabling comprehensive quantification of protein expression from lower input quantities of routinely available patient specimens, and employing a highly efficient workflow[21,22]. We report a broad scale global proteome profiling of 300 well-characterized archival FFPE breast cancer specimens and link results with detailed clinical outcome, IHC, and PAM50 RNA-based intrinsic subtypes. We demonstrate the presence of a distinct proteome group characterized by high expression of immune-response proteins and favorable clinical outcomes. We characterize the proteomic heterogeneity among a subset of 88 TNBC cases that display protein features of four subtypes of TNBC[23], which have disparate clinical outcomes. Our data identify potential biomarkers for existing chemotherapies or emerging immunotherapies.

## Results

**Proteomic analysis of FFPE breast cancer tissue samples and characteristics of study cohorts**. A total of 300 archival FFPE breast tumor primary tissues, representing 75 from each of the RNA PAM50 subtypes[4], and 38 normal reduction mammoplasty samples, were obtained (Fig. 1a, b). Samples were assembled with an original aim to be analyzed as one cohort, thus the MS data were obtained per this design, from patients diagnosed with invasive breast cancer using tissue obtained prior to adjuvant systemic therapy in 2008–2013 ($n = 178$; the 08–13 cohort) and 1986–1992 ($n = 122$; the 86–92 cohort). The 08–13 cohort included 75 basal-like, 62 Her2-Enriched, 30 luminal B, and 11 luminal A PAM50 defined cases. The 86–92 cohort provided the long-term outcome data required to gather sufficient outcome events for luminal A breast cancer cases and included 64 luminal A, 45 luminal B, and 13 Her2-Enriched PAM50 cases (Fig. 1b).

FFPE samples were macro-dissected from 3 to 6 sections to obtain >80% tumor content and analyzed using the SP3-CTP multiplex MS proteomics protocol[19,24] (Supplementary Fig. 1a). Digested peptides were labeled with a stable isotope labeled tandem mass tag (TMT) and the whole cohort was run in TMT-11-multiplex sets; each set had two of each PAM50 subtype, one normal sample, a standard SuperMix (see Methods), and a pool of all samples (Supplementary Fig. 1b, Supplementary Data 1a). The pooled control also included an isoDoping peptide library (Supplementary Data 1b) comprised of 706 synthetic peptides, with 3–5 unique peptides for 179 query proteins, whose ion signals add to their cognate peptide ion signals in the experimental channels to ensure that the combined MS1 ion signal for each peptide is above the MS2 selection threshold (Supplementary Fig. 1c). The targeted proteins included the PAM50 signature genes[4], and clinically relevant proteins expected to be present in low abundance (Supplementary Data 1b). Overall, 38 11-plex sets encompassing the 300 tumor and 38 normal samples were analyzed, with one 11-plex having 2 technical replicates of the four PAM50 subtypes (Supplementary Data 1a). The proteomic analysis quantified 9088 proteins in total (mean ~6500 proteins/11-plex) and 4214 proteins were quantified across all samples (Supplementary Fig. 2a–d). See also Source Data File.

The isoDoping method provided high coverage quantification of the targeted proteins; 164 proteins out of the 179 proteins were detected with 2 or more peptides. Among these, 93% of proteins were quantified in 85% of the samples (Supplementary Fig. 2e), including 49/50 PAM50 proteins. 74 proteins, including 14 PAM50 proteins, required at least 1 isoDoping peptide for successful identification by two peptides. The average signal to noise ratio, before normalization, of SuperMix and isoDoping

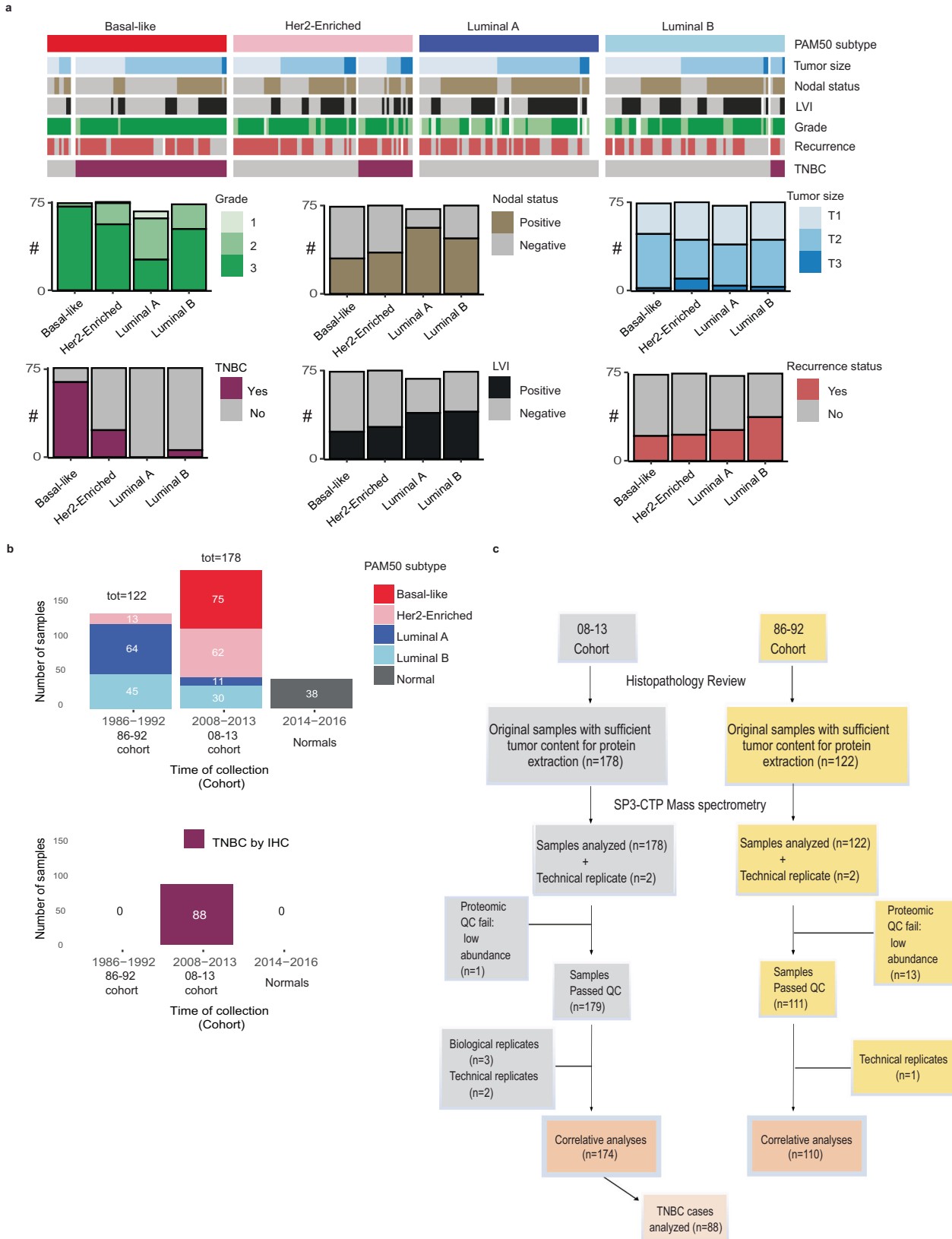

according to the sample type is displayed in Supplementary Fig. 2f, g.

We used the uniform manifold approximation and projection[25] non-linear dimensionality reduction method to determine if cases would separate based on their set characteristics. The normal and SuperMix sets were clearly separated from the tumor cohort (Supplementary Fig. 3a). However, likely due to the differences in collection techniques, pre-analytical handling, and fixation

**Fig. 1 Proteomic analysis of FFPE breast cancer tissue samples. a** The clinical features of the 300-tumor study cohort across the four PAM50 breast cancer subtypes. Samples were assembled from patients diagnosed with invasive breast cancer using tissue obtained prior to adjuvant systemic therapy in 2008–2013 ($n = 178$; the 08–13 cohort) and 1986–1992 ($n = 122$; the 86–92 cohort). The MS data were obtained with the 08–13 and 86–92 samples intermixed (see Fig. S1b batch design), however these two cohorts were analyzed separately. Pathological primary tumor size was defined as (T1 ≤ 2 cm), (T2 2–5 cm), (T3 > 5 cm); recurrence, (local, regional, distant). The feature list is in Supplementary Data 1c. LVI lymphovascular invasion, TNBC triple-negative breast cancer. **b** The distribution of the PAM50 subtypes for the 300 tumor samples described in **a** across the 86–92 and 08–13 cohorts. The study also included 38 normal breast reduction mammoplasty samples. Within the 08–13 cohort, a set of 88 cases were classified as TNBC by IHC and were analyzed as a separate cohort. **c** CONSORT flow diagram depicting the workflow numbers for the cases included in the study cohorts. TNBC triple-negative breast cancer. Source data are provided as a Source Data file.

procedures used for the 86–92 cohort samples they displayed an overall reduced signal and were depleted in lysine containing peptides, a known consequence of formalin fixation[17], leading to a batch effect observed between the 86–92 and 08–13 cohorts (Supplementary Fig. 3b–d). Thus, separate analyses were conducted for the 08–13 and 86–92 cohorts. In the 08–13 cohort, 174 cases passed quality control and were used for the subsequent analysis (Fig. 1c). The cases in the 08–13 cohort were treated in accordance with contemporary guidelines[26] and contained cases from all four PAM50 subtypes, including all 75 basal-like and 88 TNBC cases (Fig. 1b, c, Supplementary Data 1c). High reproducibility was observed between the biological replicates (referring to different specimens taken from the same patient) ($n = 3$, mean $r = 0.71$) and the technical replicates ($n = 3$, mean $r = 0.88$) (Supplementary Fig. 4a, b). All the technical replicates and 2 out of 3 biological replicates clustered adjacent to each other, while the 3rd biological replicates clustered very closely together, a variance in line with expectations for intratumoral regional sampling (Supplementary Fig. 4c). An overview clustering of all the samples included in our study showed that the 38 SuperMix replicates had the highest correlation across the 38 plexes (range 0.68–0.81) when compared to the breast tumors and normal samples (Supplementary Fig. 4c, d).

Since the 08–13 cohort provided meaningful insights into the clinically challenging TNBC and basal-like breast cancer subtypes, this cohort was the focus of our analyses. Within the 08–13 cohort, 88 cases were classified as TNBC by IHC and were analyzed as a separate cohort (Fig. 1b, c).

**Proteome analysis reveals distinct breast cancer subtypes with differential immune responses and clinical outcomes**. We examined the overlap between the proteomic data and PAM50 classification in the 08–13 cohort; we observed a good separation of the basal-like subtype while luminal A, luminal B and Her2-Enriched subtypes clustered together (Fig. 2a).

To identify distinct protein-based subtypes, we performed an unsupervised clustering using the consensus clustering algorithm[27] on the 25% most highly variable proteins ($n = 1054$) based on median absolute deviation (Supplementary Data 2a). Four robustly segregated groups were identified based on inspection of the consensus matrix and delta plots examining the change in consensus cumulative distribution function (CDF) area (Supplementary Fig. 5a–c). These four proteome-based clusters partially recapitulated the RNA-based PAM50 subtypes (Fig. 2b, c, Supplementary Data 2b). Cluster-1 ($n = 34$) consisted mostly of luminal B ($n = 18$) and Her2-Enriched ($n = 13$) PAM50 cases. Clusters-2 ($n = 50$) was significantly enriched for basal-like subtype ($n = 41$), included few Her2-Enriched, but had no luminal cases ($p$-value $< 1.16e{-}11$, Fisher's test). Cluster-3 ($n = 47$) was primarily basal-like cases ($n = 31$) but included Her2-Enriched cases ($n = 14$) ($p$-value $< 1.3e{-}4$, Fisher's test). Cluster-4 ($n = 43$) was mostly Her2-Enriched ($n = 26$) but included luminal A ($n = 8$) and luminal B ($n = 8$) cases ($p$-value $< 1.9e{-}4$, Fisher's test). Notably, 72/73 basal-like PAM50 tumors were subdivided into Cluster-2 and -3 (Fig. 2b, c), where Cluster-3 displayed the

most favorable recurrence-free survival (RFS) and overall survival (OS), whereas Cluster-2 had the worst outcomes (Fig. 3a, b). The survival differences between Cluster-2 and -3 were also observed when only the basal-like cases were examined (Supplementary Fig. 6a).

PAM50 assigned Her2-Enriched cases were distributed across the four proteome clusters, supporting the heterogeneity of these cases using mRNA-based classification[15,16,28]. IHC classified Her2+ cases ($n = 49$) were mostly luminal B and Her2-Enriched by PAM50 assignment. When these 49 cases were examined using proteins found in *ERBB2* amplicons[15,29], 14 had overall low abundance of ERBB2 and flanking proteins, while the remaining 35 cases had high expression of ERBB2 and/or other flanking proteins. ERBB2 and the adjacent GRB7 were co-expressed in the majority of these 35 cases; however, the ERBB2 amplicon proteins did not correlate with the main proteome clusters (Supplementary Fig. 6b).

We characterized Cluster-3 by computing the differentially abundant proteins that most significantly distinguish Cluster-3 from others ($n = 339$, with a log2 fold change (FC) > 0.20, adjusted Benjamini–Hochberg (BHadj) $p < 0.05$) (Supplementary Data 2c). Cluster-3 showed a high abundance of proteins involved in immune-response related pathways, including transporter proteins associated with antigen processing and presentation on MHC class I (TAP1, TAP2, TAPBP), subunits of the immuno-proteasome (PSMB9, PSMB10, PSME1, PMSE2), MHC class II proteins (HLA-DQA1, HLA-DQB1) and its chaperone CD74, interferon (IFN) γ signaling with high expression of STAT1, GBP1, GBP2, and type I IFN signaling (IFIT1, IFIT2, MX1, OAS2, OAS3) (Fig. 3c; Supplementary Data 2c). Gene set enrichment analysis (GSEA) showed that Cluster-3 was significantly enriched for immune-response processes, thus characterized as immune hot compared to other clusters (BHadj $p < 0.05$) (Fig. 3d, Supplementary Data 2d). Further enrichment analyses showed that Cluster-1 (luminal B and Her2-Enriched) had upregulation of fatty acid metabolism, catabolic, and oxidation-reduction associated processes ($n = 212$, log2FC > 0.2, BHadj $p < 0.05$ (Fig. 3d, Supplementary Data 2c, e). Cluster-2 (mostly basal-like), which exhibited the poorest clinical outcomes (Fig. 3a, b), was enriched for stromal and extracellular matrix (ECM), including collagen organization, blood coagulation, and angiogenesis processes ($n = 167$, log2FC > 0.2, BHadj $p < 0.05$) (Fig. 3d, Supplementary Data 2c, f). Cluster-2 also had elevated DNA replication and repair functions, and low expression of immune-response related pathways. Cluster-4 (Her2-Enriched and mixed luminals) was enriched for stromal and ECM components, blood coagulation, humoral immune response (complement and immunoglobulins), and hormone receptor binding ($n = 426$, log2FC > 0.2, BHadj $p < 0.05$), but showed low abundance of DNA damage repair proteins (Fig. 3d, Supplementary Data 2c, g).

**Expression of key subtype specific breast cancer proteins is consistent with the biological characteristics of each proteome**

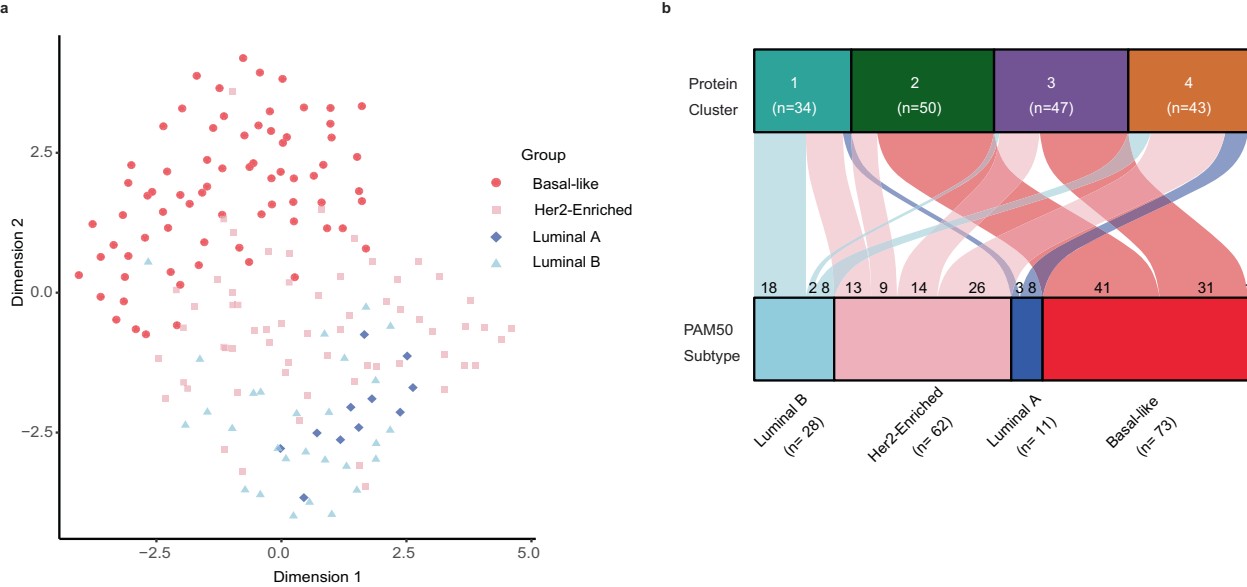

**Fig. 2 Proteome unsupervised consensus clustering reveals distinct breast cancer subtypes. a** Uniform Manifold Approximation and Projection of the 08–13 cohort for the basal-like, luminal A, luminal B, and Her2-Enriched PAM50 subtypes based on all proteins quantified in every sample (4214). **b** Alluvial plot shows the relationship between PAM50 subtypes and the four proteomic consensus clusters in the 08–13 cohort. **c** Consensus clustering of 174 cases, based on the relative abundance of 1054 most variant proteins. Immune related is defined based on the protein function as involved in immune-response biological process and for each protein cluster, the most representative terms displayed on the heatmap were selected based on g:profiler[85] enrichment analysis. Source data are provided as a Source Data file.

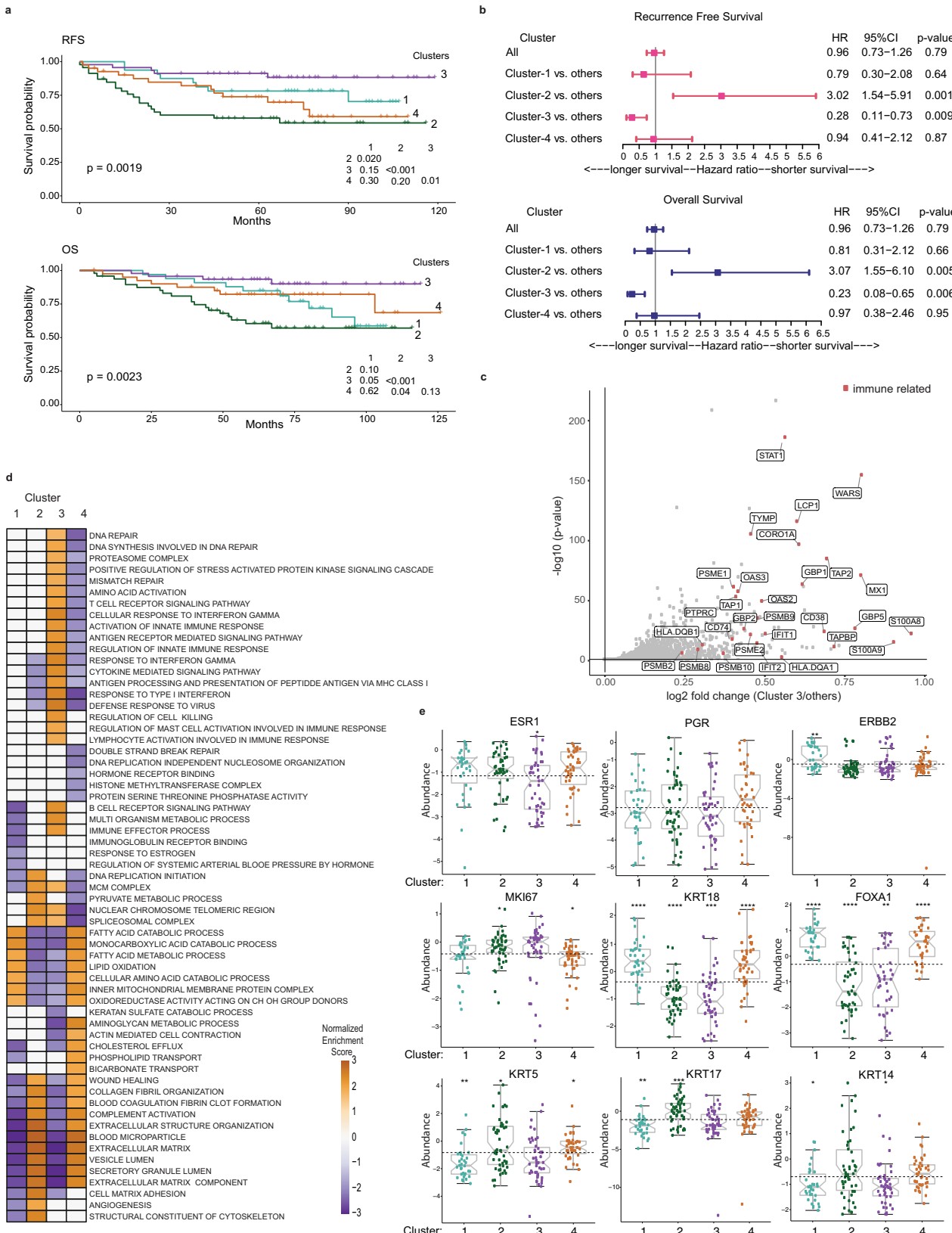

**cluster**. We examined whether the proteomic abundance of hall-mark breast cancer biomarkers, including the 49 observed PAM50 proteins, were consistent with the biological characteristics of each proteome cluster. High expression of basal cytokeratins (KRT5, KRT17, KRT14) were observed in Cluster-2, and the proliferation marker MKI67 was higher in Cluster-2, but lower in Cluster-4

(Fig. 3e). Luminal biomarkers including ESR1, FOXA1, and KRT18 were significantly lower in Cluster-3, but higher in Cluster-1 and -4 (Wilcoxon $p < 0.05$, $p < 0.01$, and $p < 0.001$, respectively) (Fig. 3e). Her2 biomarkers (ERBB2, GRB7) were significantly high in Cluster-1 (Wilcoxon $p < 0.01$ and $p < 0.05$, respectively) (Fig. 3e, Supplementary Fig. 6c). PRKDC, a biomarker reported by CPTAC[14] as the

**Fig. 3 Key characteristics of the different proteomic breast cancer clusters in the O8–13 cohort. a** Kaplan Meier plots show RFS and OS for the four proteomic clusters. **b** Forest plots for multivariate survival analyses of RFS and OS in the four proteomic clusters. The error bars represent 95% confidence interval (CI) with hazard ratio (HR) result displayed as a plotted box. Results are derived from Cox regression models and stratified log-rank tests with 2-sided $p$-values at a significance level of 0.05. Results are unadjusted for multiple comparisons. **c** Volcano plot showing differentially expressed proteins between Cluster-3 (immune hot) vs. the other clusters. Immune-related proteins with log2 fold > 0.2 and adjusted BH $p < 0.05$ are highlighted red. Results are derived from peptide-level expression-change averaging (PECA) analysis, using modified $t$-test adjusted for multiple comparisons using the Benjamini–Hochberg method. **d** Gene set enrichment analysis (GSEA) of selected significant biological processes between the four proteomic clusters (adjusted $p < 0.05$). The enriched processes are listed in Supplementary Data 2. **e** Relative protein abundance of key subtype specific breast cancer proteins across the four proteomic clusters. Boxplots show the median (center bar), and the third and first quartiles (upper and lower edges, respectively) of protein expression. Horizontal dotted line is the base mean. Boxplot whiskers range extends to the most extreme data point which is no more than 1.5 times the interquartile range from the box. Asterisks show the pairwise significance of the mean in each group against "all" as a reference: (*$p < 0.05$), (**$p < 0.01$), (***$p < 0.001$), (****$p < 0.0001$). Results are derived from a 2-sided $t$-test of the means of each cluster compared to all. Protein abundance values are based on log2 ratio for PSMs abundances divided by the relative PIS value in each TMT plex. For each protein, the median ratio of the 5 most abundant PSMs was used as relative abundance. See also Supplementary Fig. 6c. Source data are provided as a Source Data file.

most specific for the basal-like subtype, was significantly higher in Cluster-2 (Wilcoxon $p < 0.001$) (Supplementary Fig. 6c). Some key individual proteins recapitulated the major RNA PAM50 subtypes[4] (Supplementary Fig. 6d), including significantly high expression of FOXA1, MAPT and NAT1 in luminal A and B, ERBB2 and BLVRA in Her2-Enriched, and ANLN, MKI67 and PHGDH in basal-like PAM50 subtypes (Wilcoxon $p < 0.05$).

**Clinicopathological features of the different proteome breast cancer subtypes.** Most cases in Cluster-2 and -3 were associated with ER, PR and Her2 negativity by IHC clinical tests, high proliferation index (Ki67), and the core basal phenotype (defined as ER−, PR−, Her2− and [EGFR+ or CK5+])[30] (Supplementary Data 2h). Cluster-2 and -3 cases were mostly treated with chemotherapy without added hormonal therapy when compared to Cluster-1 and -4. Cluster-2 had higher recurrence rates compared to other clusters and was associated with metastasis to brain and lung that characterize the clinical behavior of aggressive basal-like subsets, while Cluster-4 was associated with metastasis to bone (Supplementary Data 2h) (Chi-square $p < 0.05$). The association between the different clusters and survival outcomes remained significant in a multivariate analysis adjusted for clinicopathological variables (Fig. 3b, Supplementary Table 1) (log-rank $p < 0.05$).

When testing the association between the MS data for ESR1, PGR, HER2 and their IHC categories, results were significant for HER2 ($p < 0.0001$) and ER ($p = 0.02$) IHC expression (Supplementary Fig. 7).

**Morphological expression of key immune biomarkers characterizes the proteome immune hot cluster.** Since the immune hot Cluster-3 had the most favorable outcomes, we examined whether it contained high levels of tumor infiltrating lymphocytes (TILs). Using International TIL working group guidelines[31], we scored the stromal TILs on the corresponding hematoxylin and eosin (H&E) slides as a continuous parameter and by low (0–50%) and high (≥50%) categories (Fig. 4a, Supplementary Data 2h). Continuous and categorical TILs scores were significantly higher in the immune hot cluster compared to other clusters (Figs. 2c, 4a, Supplementary Data 2h). The immune hot cluster also had significantly higher CD8+ TILs in the intratumoral compartment compared to other clusters (Wilcoxon $p < 0.0001$) (Fig. 4a). Since Cluster-2 and -4 were significantly enriched for ECM proteins, we assessed the percentage of ECM compartment within the total macro-dissected tumor area. Cluster-4 cases showed the highest percentage of ECM and were more stromal-enriched when compared to other clusters (Wilcoxon $p < 0.01$) (Fig. 4a).

We assessed proteins characteristic of the immune hot cluster by IHC. We selected four that were among the top differentially expressed proteins between the immune hot cluster vs. others (Supplementary Data 2c), had available antibodies applicable to FFPE; and had a practical scoring methodology on carcinoma cells by IHC; TAP1 (MHC class I), HLA-DQA1 (MHC class II), IFIT2 (type I interferon signaling) and S100A8 (Fig. 4b, c). In addition, these proteins were not highly expressed in the normal reduction mammoplasty samples. The expression of S100A8, TAP1, and HLA-DQA1 were significantly higher in the immune hot cluster compared to others (Wilcoxon $p < 0.05$, $p < 0.001$, and $p < 0.001$, respectively) (Fig. 4c). As expected from the functional enrichment, Cluster-2 displayed the lowest expression of S100A8, IFIT2, and HLA-DQA1 compared to the other clusters (Wilcoxon $p < 0.001$, $p < 0.05$, and $p < 0.001$, respectively) (Fig. 4c). When assessing the correlation between the MS data and the IHC scores for the validated biomarkers, a low-moderate correlation was noted (Supplementary Fig. 7).

**Combined IHC expression of antigen presentation proteins TAP1 and HLA-DQA1 correlate with survival.** We assessed the prognostic value of the TAP1, HLA-DQA1, IFIT2 and S100A8. Tumors with high expression of TAP1 or HLA-DQA1 as single biomarkers displayed significantly better RFS (log-rank $p < 0.05$) (Supplementary Fig. 8a). IFIT2 and S100A8 did not have significant associations with RFS. Since TAP1 and HLA-DQA1 function in antigen presentation processes that allow T cells to recognize and kill tumor cells, we examined the prognostic value of their combined IHC values. Tumors with high IHC expression for both TAP1 and HLA-DQA1 showed the most favorable RFS, while the subgroup with low expression for both had the worst RFS (log-rank $p = 0.016$). The subgroups with a high expression for only one of these biomarkers were characterized with intermediate RFS (Supplementary Fig. 8b). 70% (21/30) of the cases classified as (TAP1 high/HLA-DQA1 high) were in Cluster-3, while 90% (76/84) of (TAP1 low/HLA-DQA1 low) cases were in other clusters (Chi-square $p$-value < 0.00001) (Supplementary Data 2h).

We subsequently confirmed our observations on an independent, clinically similar set of 176 breast cancer cases and showed that high expression of HLA-DQA1 as a single biomarker had a significantly better survival (log-rank $p = 0.02$) and a trend was seen for high TAP1 as a single biomarker (log-rank $p = 0.09$). These data also confirmed that tumors with high IHC expression for both TAP1 and HLA-DQA1 showed the most favorable survival, while the subgroup with low expression for both had the worst RFS (log-rank $p = 0.05$) (Supplementary Table 2; Supplementary Fig. 9).

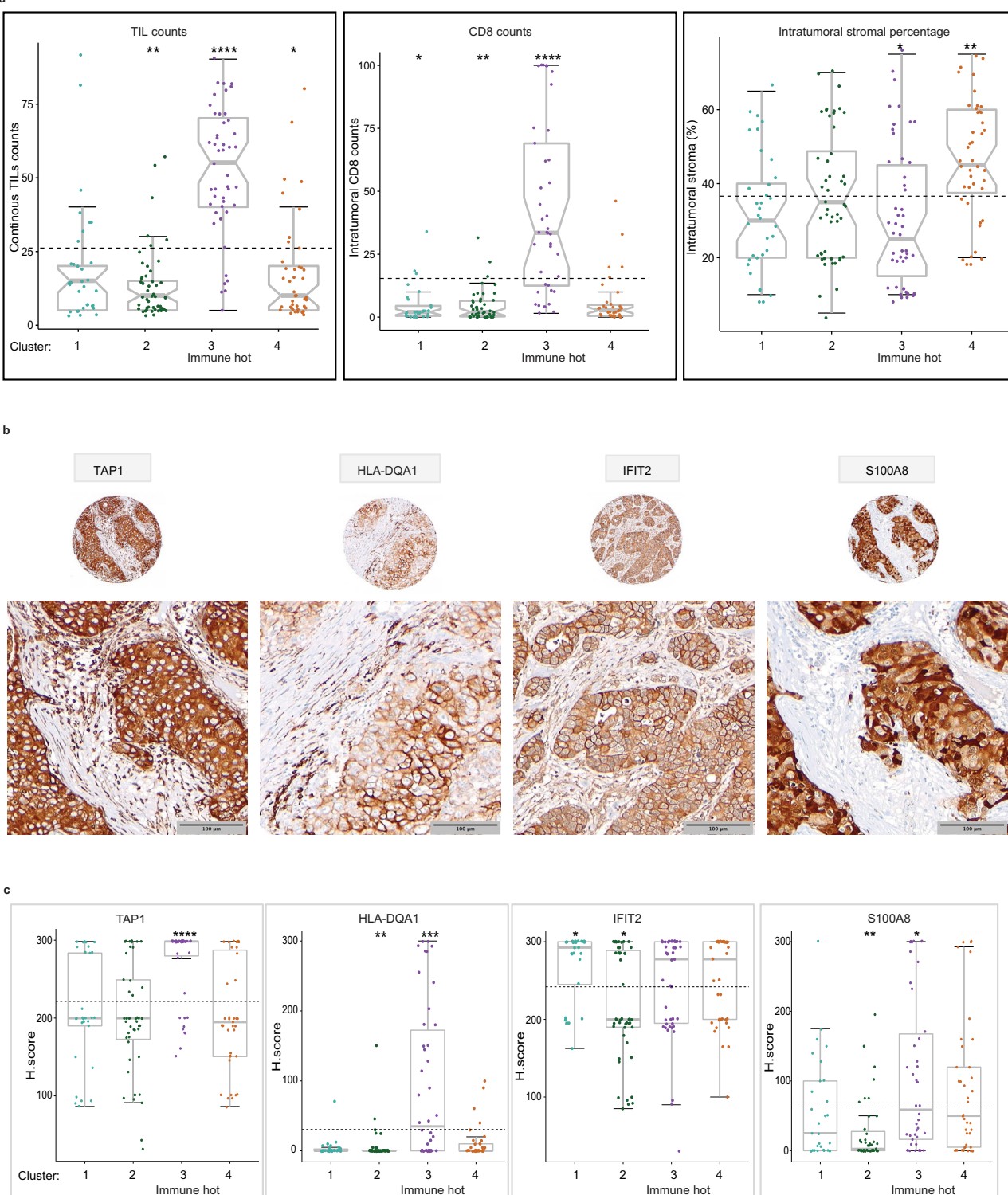

**Comparison with previous breast cancer proteomics studies**. In order to validate findings with available published datasets, two previous proteomic datasets were used (a) CPTAC breast tumor cohort by Krug et al.[15] and (b) OSLO2 breast cancer landscape cohort by Johansson et al.[16].

Consensus clustering of CPTAC breast tumor proteomes using 939 proteins that overlap with the 1054 mostly highly variant proteins in our 08–13 cohort identified four main proteome clusters that highly resembled the original NMF clusters of

LumA-I, LumB-I, Basal-I, HER2-I and consistent with Krug et al. showed that some luminal A PAM50 cases were included in the LumB-I NMF cluster. Our analysis demonstrated the existence of subsets enriched for immune-response pathways at the proteome level and these included basal-like and Her2-Enriched subtypes. In contrast to the 08–13 cohort, these subsets were not captured as separate and defined clusters by CPTAC analysis. Consistent with our analysis on the 08–13 cohort, stromal pathways were enriched in luminal A tumors and lipid metabolism were

**Fig. 4 Morphological expression of key immune biomarkers characterizes the proteome immune hot cluster in the 08–13 cohort. a** Intratumoral percentage distribution of stromal TILs by H&E (left panel), CD8[+] TILs by IHC (middle panel), and stroma by H&E (right panel) for tumor sections. The horizontal dotted line is the base mean. Boxplots show the median (center bar), and the third and first quartiles (upper and lower edges, respectively). Boxplot whiskers range extends to the most extreme data point which is no more than 1.5 times the interquartile range from the box. Asterisks show the pairwise significance of the mean in each group against "all" as a reference: (*$p < 0.05$), (**$p < 0.01$), (***$p < 0.001$), (****$p < 0.0001$). Results are derived from a Wilcoxon test with 2-sided $p$-value. **b** Representative images of IHC expression of four proteins highly expressed in the immune hot cluster at ×20 and ×40 magnification. Scale bar = 100 μm. **c** Verification of proteomic expression of four proteins highly expressed in Cluster-3 (immune hot) by immunohistochemistry. Scores use the H scoring system (intensity × positivity) for cytoplasmic staining in the invasive tumor cells. Boxplots show the median (center bar), and the third and first quartiles (upper and lower edges, respectively). Boxplot whiskers range extends to the most extreme data point which is no more than 1.5 times the interquartile range from the box. Asterisks show the pairwise significance of the mean in each group against "all" as a reference: (*$p < 0.05$), (**$p < 0.01$), (***$p < 0.001$), (****$p < 0.0001$). TILs tumor infiltrating lymphocytes. Source data are provided as a Source Data file.

enriched within luminal B and Her2-Enriched tumors (Supplementary Fig. 10a).

To validate our findings on the 36 cases of the 4 main subtypes (9 for each PAM50 type) in the OSLO2 landscape cohort, by applying consensus clustering using the 775 proteins that overlap with the 1054 mostly highly variant proteins of our 08–13 cohort, highly resembled four of the main consensus core tumor clusters (CoTCs) and their biological functions as reported in Johansson et al. These clusters consisted of CoTC1 (basal-like immune cold), CoTC2 (basal-like immune hot), CoTC3 with few CoTC6 cases (luminal A-enriched) and CoTC6 (luminal B and Her2-Enriched). Importantly, the immune distinctions within the basal-like subtype were entirely reproduced using our highly variant proteins showing that the two basal-like samples of OSL.3EB and OSL.449 (CoTC2) were consistently classified as basal-immune hot cluster when compared to other basal cases characterized as basal-immune cold. These findings are displayed in Supplementary Fig. 10b.

**Triple-negative breast cancer comprises of four clinically distinct proteome clusters.** PAM50 basal-like subtype is clinically approximated by TNBC IHC status; a group with heterogeneous biology unresponsive to anti-endocrine or anti-Her2 therapies. We addressed if our proteomic data could subclassify TNBC cases in a more clinically informative way. We analyzed 88 IHC-defined TNBC cases (profiled by RNA-based PAM50 as: 61 basal-like, 22 Her2-Enriched, and 5 luminal B), all in the 08–13 cohort (Fig. 1b). Unsupervised classification using the consensus clustering algorithm on the 25% most highly variable proteins ($n = 1055$) identified four robust clusters (Fig. 5a, b, Supplementary Data 3a, Supplementary Fig. 11a–c) with prognostically distinct differences in RFS and OS (Fig. 5c). TNBC-Cluster-1 consisted mostly of basal-like cases and had the most favorable survival compared to other clusters (Fig. 5c). This cluster was characterized by immune-response, antigen processing and presentation, and type I and II IFN signaling processes (Fig. 5d, Supplementary Data 3b, c). TNBC-Cluster-1 cases had high TIL content and high TAP1 and HLA-DQA1 expression (Fig. 5a); thus termed basal-immune hot. TNBC-Cluster-2 was mostly basal-like cases, had intermediate survival, and was enriched for ECM, blood coagulation, and humoral immune-response processes (Fig. 5c, d, Supplementary Data 3b, d). This cluster displayed significantly low expression of CLDN3 and the differentiated luminal cell surface markers MUC1 and EPCAM[32]; thus termed mesenchymal. TNBC-Cluster-3 was mostly Her2-Enriched cases by PAM50, had intermediate survival, and was enriched for lipid metabolism, catabolic, and oxidation-reduction processes (Fig. 5c, d, Supplementary Data 3b, e). This cluster had high expression of luminal cytokeratins 7, 8, 18, 19 and prolactin-induced protein (PIP)[33]; thus named luminal. TNBC-Cluster-4 were all basal-like cases, exhibited the poorest survival, was enriched for DNA replication and cell cycle proteins, had few

immune-related peptides (Fig. 5c, d, Supplementary Data 3b, f), and minimal TILs (Fig. 5a); thus termed basal-immune cold. When overlaying the TNBC proteome clusters with the 08–13 clusters, all TNBC-Cluster-1 cases were members of the immune hot Cluster-3. In contrast, the majority of cases in TNBC-Cluster-2 and TNBC-Cluster-4 fall into 08–13 Cluster-2, while most TNBC-Cluster-3 cases were members of 08–13 Cluster-4 (Supplementary Data 3g).

The protein abundance of 35 protein/RNA pairs in common with a 80 gene RNA-based TNBC classifier[23] showed our TNBC clusters were highly correlated with the RNA subtypes; TNBC-Cluster-3 (luminal) and 'luminal-androgen receptor', TNBC-Cluster-2 (mesenchymal) and 'mesenchymal', TNBC-Cluster-4 (basal-immune cold) and 'basal-immune suppressed', and TNBC-Cluster-1 (basal-immune hot) and 'basal-immune activated' (Fig. 6a, Supplementary Fig. 11d). The existence of these TNBC proteome clusters and their biological features were validated when applying consensus clustering, with identical parameters, on the 935 proteins that overlap with the 1055 mostly highly variant proteins in our 08–13 TNBC subset ($n = 88$) on the proteomic data for a set of 28 TNBC cases included in the CPTAC breast cancer cohort by Krug et al.[15] (Supplementary Fig. 12).

**Identification of protein biomarkers associated with clinical outcomes in TNBC.** We identified characteristic proteome signatures with a reduced number of proteins as candidate prognostic biomarkers for TNBC. Cox proportional-hazards analysis on protein abundance and survival outcomes identified 85 proteins that were significantly associated with longer RFS (HR < 1, BHadj log-rank $p < 0.05$), including several immune-related proteins, and 18 proteins that were significantly associated with poor RFS (HR > 1, BHadj log-rank $p < 0.05$) (Fig. 6b, Supplementary Data 3h). Restricting our analysis to proteins that were also found to be specifically enriched in a TNBC cluster (log2FC > 0.20, BHadj $p < 0.01$), we mapped these prognostic protein candidates to the TNBC clusters (Supplementary Data 3i). Among the 85 proteins associated with favorable RFS, 44 were also significantly highly expressed in TNBC-Cluster-1, whereas only one was characteristic for TNBC-Cluster-3 and none for TNBC-Cluster-2 or -4 (Supplementary Data 3i). Among the 18 proteins associated with poor RFS, four were characteristic for TNBC-Cluster-4 that displayed the worst survival, while none were characteristic for other TNBC clusters (Supplementary Data 3i).

**Proteomic signatures capture the biologic heterogeneity in luminal breast cancers.** Cases in the 86–92 cohort included patients classified as ER + using the dextran-coated charcoal ligand binding assay (confirmed by IHC). Consensus clustering of 110 qualified cases (Fig. 1c) identified three main clusters; 86–92-

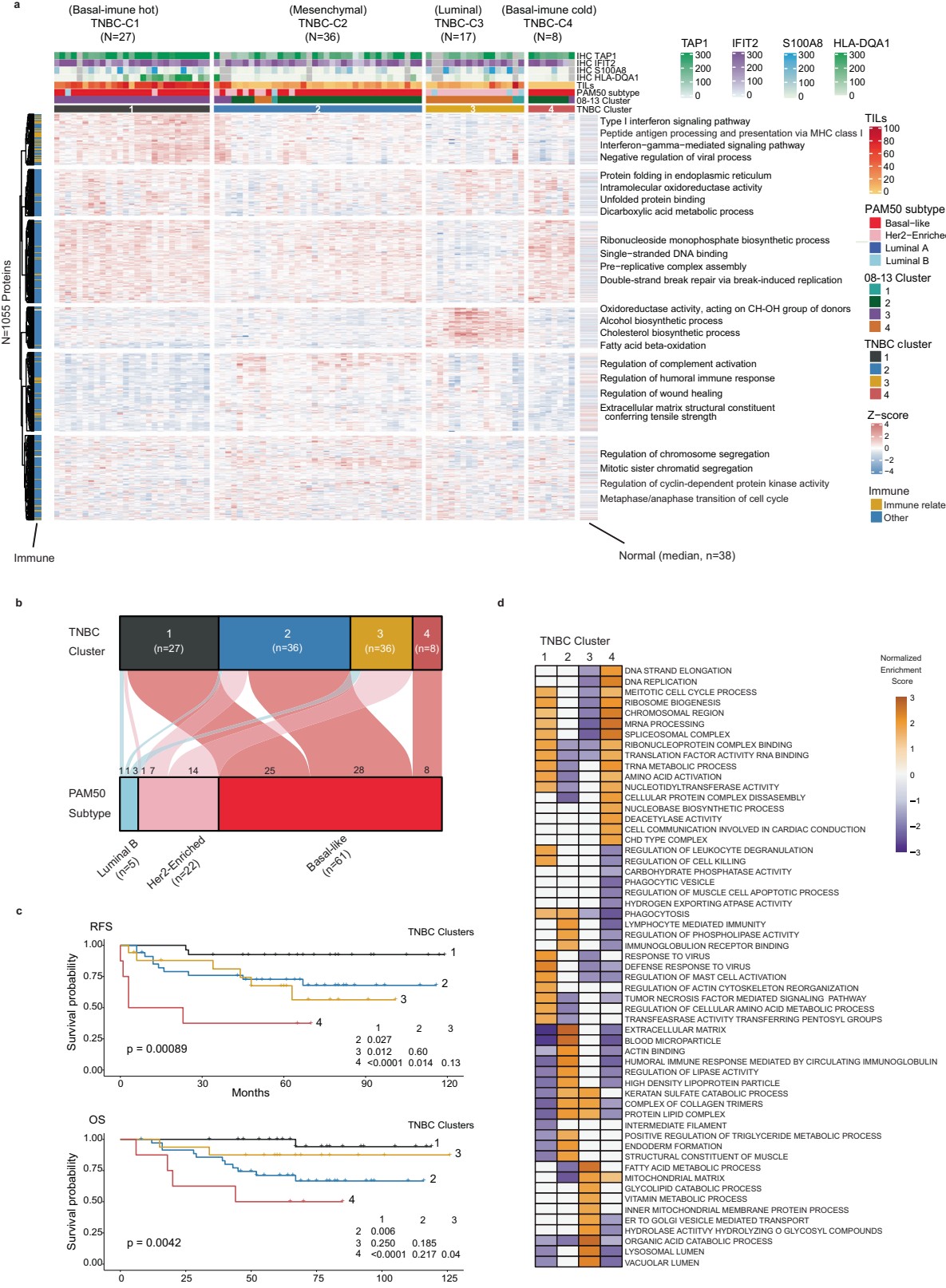

Cluster-1 (n = 41), 86–92-Cluster-2 (n = 30), and 86–92-Cluster-3 (n = 35), while 4 cases did not aggregate with the main clusters (Fig. 7a; Supplementary Data 4a, Supplementary Fig. 13a–c). 86–92-Cluster-1 and -2 were mainly luminal A cases, while

86–92-Cluster-3 was predominantly luminal B by PAM50 data. 86–92-Cluster-1 had elevated RNA processing and splicing processes, but was depleted for immune-related proteins involved in antigen processing and presentation, and type I and type II IFN

Fig. 5 Proteomic analysis reveals four clinically important subtypes in TNBC within the 08–13 cohort. a Consensus clustering of 88 IHC-defined TNBC cases, based on the relative abundance of 1055 most variant proteins. Immune related is defined based on the protein function as involved in immune-response biological process and for each protein cluster, the most representative terms displayed on the heatmap were selected based on g:profiler[85] enrichment analysis. b Alluvial plot shows the distribution of PAM50 subtypes across the TNBC clusters. c Kaplan Meier plots for RFS and OS across the TNBC clusters. d GSEA of selected significant biological processes between the TNBC clusters. Results are derived from normalized enrichment scores for most enriched pathways for each cluster compared to others with adjusted $p < 0.05$. IHC immunohistochemistry, RFS recurrence-free survival, OS overall survival. The enriched processes are listed in Supplementary Data 4. Source data are provided as a Source Data file.

signaling (Fig. 7b, Supplementary Data 4b, c, Supplementary Fig. 13d). The 86–92-Cluster-2 was enriched for stromal proteins and ECM components, including several collagen members (termed luminal A stromal-enriched) (Fig. 7b, Supplementary Data 4b, d, Supplementary Fig. 13d). 86–92-Cluster-3 had high expression of proteins for DNA replication, cell cycle, response to DNA damage, and immune-response, including antigen processing and presentation on MHC class I, and T cell signaling (Fig. 7b, Supplementary Data 4b, S4e, Supplementary Fig. 13d), and was depleted for ECM, blood coagulation, epithelial cell differentiation and response to estrogen and steroid hormones compared to 86–92-Cluster-1 and -2.

**Identification of protein biomarker candidates for survival outcome in luminal patients**. We sought to identify individual protein candidates that could be associated with tamoxifen benefit or resistance within a long-term follow-up of 10 years or more by associating the continuous increase of individual proteins and RFS. Multiple correction testing identified fatty acid-binding protein-7 (FABP7) as the only candidate biomarker associated with >10-year RFS on tamoxifen treatment (log-rank BHadj $p = 0.00004$) (Supplementary Data 4f, Supplementary Fig. 13e), consistent with previous literature[34]. High *FABP7* mRNA expression was significantly associated with favorable RFS in luminal A and luminal B subtypes in publicly available datasets[35] (log-rank $p < 0.01$) (Supplementary Fig. 13f). The clinicopathological characteristics of the 86–92 cohort showed significantly higher expression of the Ki67 proliferation marker in 86–92-Cluster-3 compared to the 86–92-Cluster-1 and -2 (Chi-square $p = 0.02$) (Supplementary Data 4g). However, no statistically significant differences were observed in RFS and OS (Supplementary Table 3).

## Discussion

In this study, the size of our cohort allowed us to comprehensively profile breast cancer surgical pathology specimens linked to clinical outcomes, detailed treatment, pathological factors, IHC, and PAM50 intrinsic subtype data.

Our standardized SP3-CTP discovery MS methodology[18–20] reproducibly captured in depth proteomic features of breast cancer using significantly smaller FFPE material amounts compared to previously reported FFPE MS methods[17,22], an amount readily available in most bio-repositories. Within this experiment, we further implemented isoDoping, a semi-targeted method of isobaric peptide doping, forcing the MS to also reproducibly detect a set of 706 peptides corresponding to 179 lower abundance proteins related to breast cancer. The majority of the doped peptides and 93% of their corresponding proteins were quantified in every sample, including 49/50 PAM50 proteins, demonstrating that the isoDoping strategy can reliably quantify proteins of interest within a global proteome profiling study without specialized assay development or a different analytical routine. The 4214 proteins quantified in every sample ($n = 342$) across a large-scale breast cancer project using minimal tissue demonstrates the

efficiency and high sensitivity of the SP3-CTP approach for FFPE cancer proteomics studies[36–39].

The analysis of the 08–13 set of 174 FFPE samples revealed four proteome groups with distinct clinical outcomes. Within the aggressive basal-like breast cancer subtype, two main proteome subsets had different characteristic protein signatures, differential immune responses and remarkably different survival. Importantly, these groups are currently classified as one group by RNA PAM50 (basal-like), illustrating the inherent heterogeneity that exists beyond current subtyping classifications. Our result is consistent with a proteomic profiling study of 2 basal-immune hot cases vs. 7 basal-immune cold cases[16]. However, our study on clinically annotated archived basal-like FFPE samples had 31 cases of basal-immune hot and 41 cases that could be classified as basal-immune cold, and are linked to outcome data. The division of basal-like cancers based on immune features was also evident in the four proteome subgroups for 88 TNBC cases with disparate survival outcomes, which aligned with RNA level heterogeneity of TNBC[23]. While TNBC subgroups at the DNA and RNA level[23,40,41] have identified basal-immune distinctions with prognostic value, current protein-based diagnostic tests cannot discriminate these TNBC subtypes for treatment decisions. Thus, our outcome-linked proteomic data could aid the development of protein biomarkers for clinical tests to distinguish TNBC/basal-like patients with favorable versus poor prognosis that may benefit from therapies beyond standard chemotherapies.

The immune hot cluster that displayed the most favorable survival represents a group of immunogenic breast cancers enriched for immune-related pathways, illustrating that their combined functions may form an effective anti-tumor immune-response[42,43]. Cases in the immune hot group were predominantly basal and Her2-Enriched subtypes and included all TNBC-Cluster-1 basal-immune hot cases. They were enriched in type I and type II IFN signaling molecules, the STAT1 transcription factor, and other components of tumor antigenicity including immunoproteasome subunits that generate neopeptides for MHC class I loading. IFN signaling induces the expression of MHC class I and enable dendritic cells to present cancer antigens on MHC class I and II, thus playing an essential role in the priming, activation, and tumor infiltration of effector T cells[42,44,45]. The display of cancer antigens via MHC class I could spur recognition by CD8 + cytotoxic T cells that produce high levels of IFNγ and STAT1, resulting in a stronger immune response and release of cancer antigens[46,47]. Our findings that associate high expression of these pathways and elevated CD8 immune infiltrates with improved survival is consistent with previous studies[48–51] and is expected as our 08–13 cohort was mostly treated with adjuvant anthracycline-taxane regimens which are known to induce immunogenic tumor cell death via release of cancer antigens and activation of antigen presenting cells sensitizing these tumors to the immune system[52]. While the 08–13 cohort patients were not treated with immunotherapy, our findings could impact immunotherapy trial results since some patients do not benefit from immune checkpoint blockade due to deficiencies in antigen presentation features that are not revealed by typical PD1/PDL1 clinical tests[46,53,54]. The finding that most

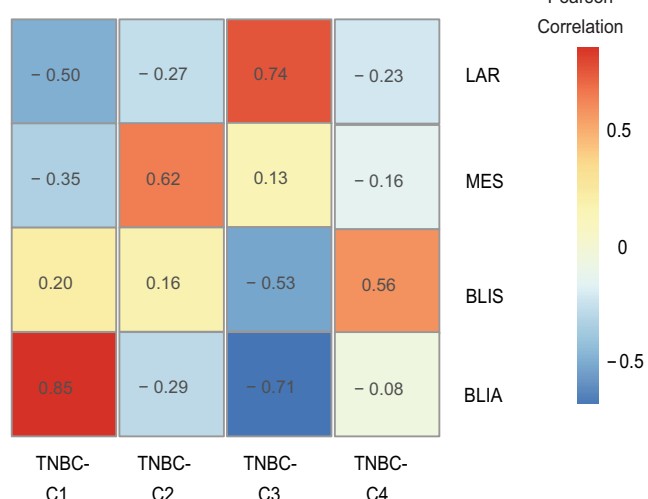

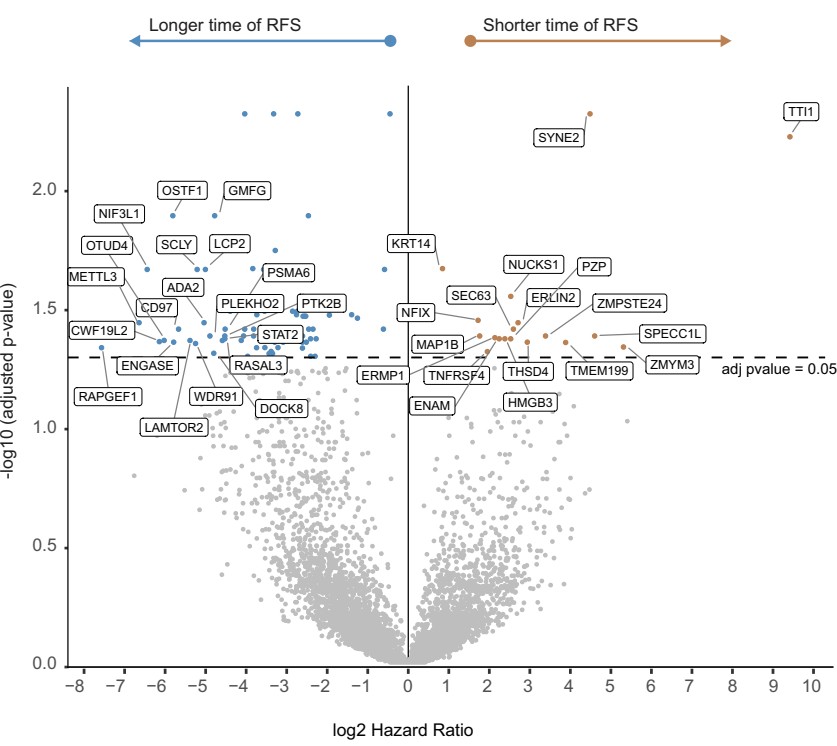

**Fig. 6 RNA-protein correlated stratification of biological subgroups and clinical outcomes in TNBC. a** Comparison between TNBC proteomic clusters with published RNA-based TNBC subgroups. BLIA basal-immune activated, BLIS basal-immune suppressed, MES mesenchymal, LAR luminal androgen receptor[23]. 35 cognate proteins identified from the 80 gene TNBC RNA classifier were used to generate the correlation heatmap based on the median expression of proteins for each TNBC subgroup. **b** Volcano plot showing proteins significantly associated with RFS in TNBC. Results are based on a Cox regression hazard model with a 2-sided log-rank *p*-value. Results were adjusted for multiple comparison using the Benjamini–Hochberg method. The *x*-axis is log2 hazard ratio (HR) and the *y*-axis is −log10 (*p*-value). Low (blue) and high (orange) HR's indicate proteins associated with longer and shorter survival, respectively. The horizontal and vertical lines indicate *p* < 0.05, and log2HR > 0 or <0, respectively. The proteins and HR's are listed in Supplementary Data 4. For visibility reasons only the top 20 proteins showing the lowest HR and highest HR with adjusted *p*-value < 0.05 were included in the plot. TNBC triple-negative breast cancer, RFS recurrence-free survival. Source data are provided as a Source Data file.

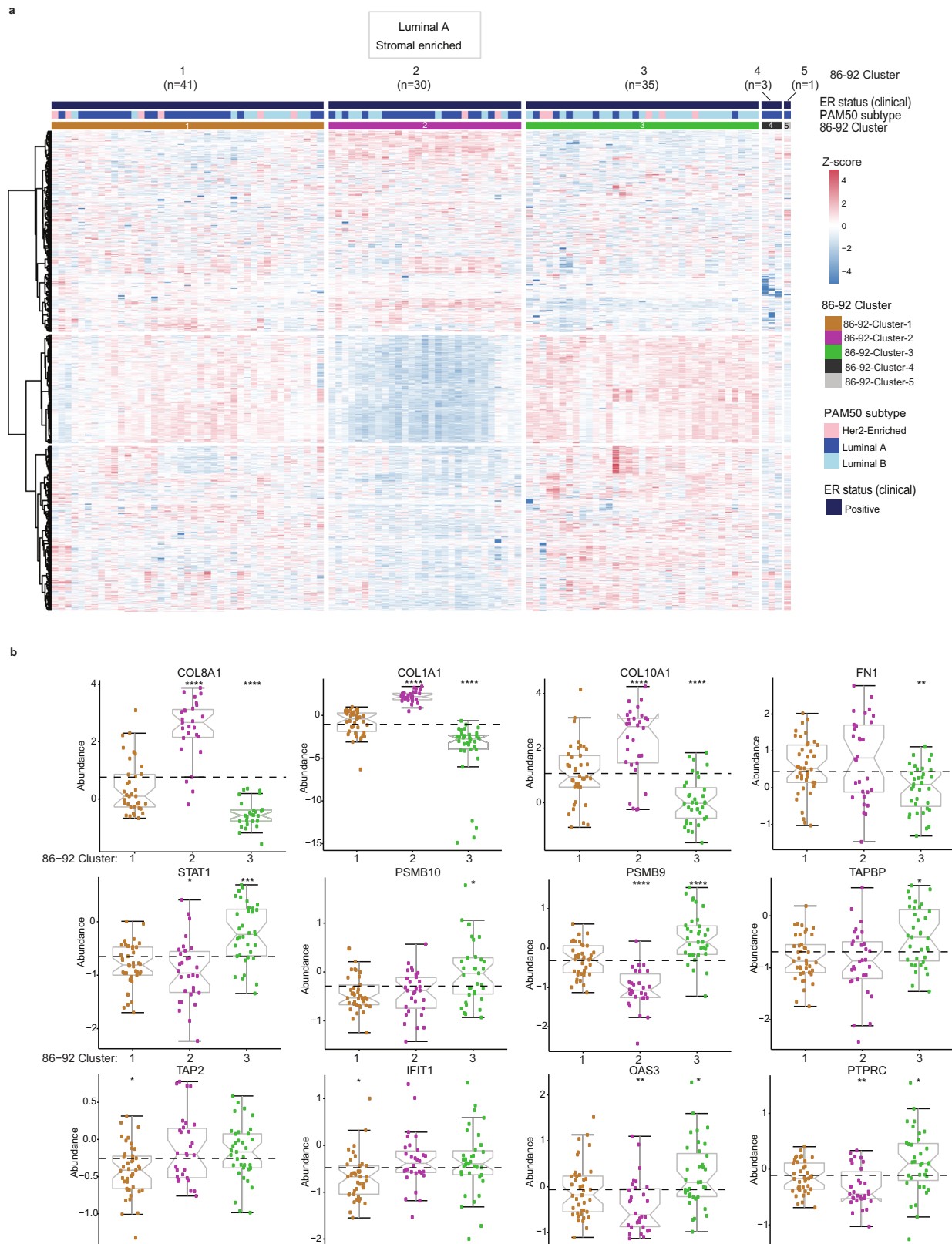

**Fig. 7 Proteomic signatures capture the biologic heterogeneity in luminal breast cancers in the 86–92 cohort. a** Consensus clustering of 110 evaluable cases, based on the relative abundance of 1054 most variant proteins. Four cases formed two separate clusters. **b** The expression levels of selected proteins in the 3 main clusters. Boxplots show the median (center bar), and the third and first quartiles (upper and lower edges, respectively) of protein expression. Each data point is one case. Boxplot whiskers range extends to the most extreme data point which is no more than 1.5 times the interquartile range from the box. Asterisks show the pairwise significance of the mean in each group against "all" as a reference: (*$p < 0.05$), (**$p < 0.01$), (***$p < 0.001$), (****$p < 0.0001$). Results are derived from a 2-sided $t$-test of the means of each cluster compared to all. Source data are provided as a Source Data file.

cases with high levels of the antigen presentation biomarkers TAP1 and HLA-DQA1 were in the immune hot cluster suggests that these two IHC proteins are candidates for biomarker panels that match patients to immune-modulating chemotherapy and immune checkpoint blockade. Other proteins elevated in the immune hot cluster with available IHC quality antibodies could also be used and developed as candidate biomarkers. Of note, the assessment of the validated markers by IHC was performed only on the carcinoma cells and using the H score that, in addition to positivity, takes into account staining intensity when reporting the IHC expression. These variables along with using a TMA format and the differences related to IHC as a multi-step antibody-based assay vs. MS contribute to the weak-moderate correlations observed with these biomarkers.

The 08–13-Cluster-2 group had the poorest survival and minimally expressed immune-related proteins, representing an immunologically ignorant or excluded group[55]. This group was enriched for the basal-like subtype, included most of the TNBC-Cluster-2 and -4 cases, and was enriched for ECM proteins associated with collagen organization and angiogenesis, potentially reflecting an immune-evasive, highly proliferative, and invasive tumor microenvironment[56]. Several stromal and ECM proteins in Cluster-2, such as FN1, act as a barrier that blocks immune cells from infiltrating the tumor and generating effective anti-tumor immunity[56]. Patients in Cluster-2 and TNBC-Cluster-2 and -4 might benefit from treatments targeting this immune barrier, such as angiogenic inhibitors or immune-augmentation strategies[55].

Our data show that Cluster-4, containing some Her2-Enriched, luminal A and B subtype cancers, was also enriched for ECM pathways and stromal content is consistent with CPTAC results on the presence of a proteome stromal-enriched cluster[14]. However, we show that stromal enriched breast cancers are heterogeneous and form two distinct groups (Cluster-4 and -2) that are both depleted for immune-related proteins, but differ in their biological subtype, the expression of DNA replication and DNA damage repair pathways, and clinical outcome. Cluster-4 Luminal/Her2 stromal enriched cases had significantly improved survival when compared to stromal-enriched cases with a basal-like subtype (Cluster-2), illustrating the opposing activities in the stroma compartment that can vary with tumor subtype. Our results are consistent with TCGA observations showing that, unlike basal-like cases, luminal tumors enriched for intratumoral stroma had improved prognosis[57].

The unique 08–13-Cluster-1 containing luminal B and Her2-Enriched cases had elevated lipid and fatty acid metabolic processes which agree with findings that these tumors types may rely more on uptake of exogenous fatty acids[58,59]. Supporting this, our basal-like containing Cluster-2 and -3 and the CPTAC basal-containing subtype[15] were depleted for lipid metabolism and oxidation processes.

Our findings demonstrate the heterogeneity that still exists within Her2-Enriched tumors whether classified by PAM50, IHC, ERBB2 levels, or the proteins typically found in amplicons encompassing ERRB2[15,60], where the Her2-Enriched tumors in the 08–13 cohort were distributed across all four proteome clusters. In Cluster-1, Her2-Enriched cases had higher ERBB2 expression compared to those in other clusters, suggesting that biomarkers defining Cluster-1 could be evaluated in anti-Her2 therapy trials. In contrast, some Her2-Enriched tumors in Cluster-4 had low expression of ERBB2 and high expression of ECM proteins and could have a different clinical impact, as stromal proteins may drive resistance to trastuzumab in Her2-Enriched tumors[61]. In particular, tumors with low ERBB2 levels in TNBC-Cluster-3 were classified as Her2-Enriched by RNA, despite being TNBC by validated clinical IHC assays, suggesting

that TNBC/Her2-Enriched subgroups warrant further evaluation to assess responsiveness to anti-Her strategies[62].

Our analysis of ER + cases with mature clinical data identified a stromal-enriched subset (86–92-Cluster-2) consistent with previous reports[57,63], which could help subclassify luminal breast cancer. However, our data characterize the luminal A stromal enriched cluster in a more comprehensive manner and identify protein candidates that are beyond those captured by the restricted number of proteins in the antibody-based RPPA assay. The luminal B enriched 86–92-Cluster-3 had high expression of Ki67 and immune-response proteins compared to the luminal A clusters concordant with recent reports[15,64].

The use of FFPE tissues and inconsistent fixation methods could have impacted this study, as some proteins might be adversely affected by formalin-induced cross-links and modifications that interfere with protein recovery and analysis[17,65]. While formalin-induced modifications were more prevalent in the 86–92 cohort, necessitating their separate analysis, those results were consistent with the known features of luminal A and B subtypes[57,64] and with CPTAC results on frozen tissue[15], supporting the robustness of our findings to sample handling. Since FFPE is commonly available from clinical laboratories, our findings demonstrate the high utility for using an LC-MS/MS approach in clinical trials where FFPE samples are routinely collected. It is also critical that proteomic FFPE studies control for fixation variables in study design, adjust for potential effects of sample collection, and minimize differences in processing (e.g. fixation times). However, the high number of samples analyzed in this study and the high proteome coverage achieved even for FFPE materials still described biologically and clinically relevant subtype heterogeneity in both the 86–92 and 08–13 cohorts.

Our study demonstrates that global proteomic analysis on standard FFPE specimens with linked outcome data characterizes the heterogeneity of breast cancer in a reliable and clinically applicable manner. The findings on immune distinctions, ECM, and lipid metabolism pathways are potentially clinically relevant as standard clinical tests do not yet interrogate this level of heterogeneity for breast cancer subtyping. Furthermore, this study identifies protein candidates for in-depth analysis of existing archived clinical trial FFPE specimens, providing a valuable resource to develop diagnostic and prognostic biomarkers in breast cancer.

## Methods

**Ethics approval and consent to participate**. This study was approved in accordance with the ethical standards of the institutional board of the University of British Columbia and BC Cancer (approval number: H17-01207). All patients had signed a written informed consent to allow the use of their tumor tissue for future study–related research purposes. The approval for the subsequent use of these previously assembled patient specimens for this proteomics study was obtained under a waiver of informed consent policy, per the Canadian Tri-Council Policy Statement: Ethical Conduct for Research Involving Humans-TCPS 2. Only de-identified, coded study ID numbers were provided, and no participant compensation is associated with this study.

**Patients samples and study datasets**. The current study included 300 archival FFPE tissues corresponding to primary tumor excision specimens of patients diagnosed with invasive breast cancer in the periods of 2008–2013 ($n = 178$; 08–13 cohort) and 1986–1992 ($n = 122$; 86–92 cohort). These specimens were assembled from different centers across the province of British Columbia and retained at Vancouver General Hospital, Canada. Cases were linked to well-annotated data including clinicopathological factors, and detailed treatment and clinical outcome information was available through BC Cancer's Breast Cancer Outcome Unit. Cases were derived from two published cohorts and selected specifically to include 75 cases each of the four major intrinsic PAM50 RNA profile-defined breast cancer subtypes: Basal-like, Her2-Enriched, Luminal B and Luminal A. Full details on the methods involved in PAM50 intrinsic subtyping for these two cohorts, including FFPE macrodissection, RNA extraction, cDNA synthesis, and qRT-PCR subtype predictions are as previously published[66–68].

*08–13 cohort:* Basal-like and Her2-Enriched PAM50 cases in the current study were mainly derived from a cohort of patients diagnosed with invasive breast during the period January 2008 to September 2013, originally selected to enrich for ER-negative and ER low breast as previously described[66]. Cases were assembled from five participating centers across British Columbia that maintain high reproducibility and proficiency for IHC testing under the Canadian Immunohistochemistry Quality Control program. The median follow-up of the original cohort was 5.6 years; cases were treated in accordance with contemporary guidelines[26].

The 08–13 cohort included 88 IHC-defined TNBC cases (PAM50: 61 basal-like, 22 Her2-Enriched, and 5 luminal B) and these were further analyzed as a separate cohort. For the analysis of TNBC cohort, we used a broader definition of TNBC to include cases defined as weakly positive for ER staining based on the Allred scoring system (ER Allred scores 3–5)[66,69–71].

*86–92 cohort:* To account for the long follow-up required to obtain sufficient events for outcome analyses, the majority of luminal PAM50 cases in the current study were derived from patients diagnosed with invasive breast cancer in the period January 1986 to September 1992[67]. At the time the samples were originally acquired, many high-risk hormone receptor positive patients were treated with adjuvant tamoxifen and no chemotherapy in adherence to provincial guidelines recommended at that time[72,73]. Thus, the majority of cases in the 86–92 cohort were postmenopausal with high risk features including lymph node positive disease, high grade and lymphovascular invasion (LVI) with a recurrence rate of 50% within 10 years. These treatment decisions were based upon different clinicopathological factors including positive ER levels as determined by dextran-coated charcoal ligand binding assay, a technique that required fresh-frozen tissue and thus the specimens had been frozen prior to formalin fixation. The median follow-up for cases in this cohort was 12.6 years and clinical outcomes were periodically updated by the British Columbia Breast Cancer outcome unit as previously published[72].

*Normal cohort:* A set of normal FFPE samples were sourced from 38 healthy women who were referred to plastic surgery at Vancouver General Hospital and UBC hospital for reduction mammoplasties in the period January 2015 to September 2017. The median age of women at time of surgery was 42 years old and ranged between 19 and 61 years old.

*IHC validation cohort:* A tissue microarray for an independent set of 176 breast cancer cases was used to validate observations on the 08–13 cohort for the key protein IHC biomarkers. This validation cohort was sourced from breast cancer women who were referred to the BC Cancer between the years 2005–2009 and had similar clinicopathological characteristics to the 08–13 cohort and was analyzed for IHC biomarker association with clinical outcomes. The median follow-up for the IHC validation cohort was 10 years and patients were treated in accordance with contemporary guidelines[26]. Characteristics of this cohort appear in Supplementary Table 2.

### SP3-clinical tissue LC-MS/MS-based proteomics

*Tissue sample acquisition and preparation.* Hematoxylin and eosin slides for the corresponding archival FFPE breast cancer specimens were reviewed by a pathologist who circled areas containing viable invasive breast carcinoma. Depending on the tumor surface area, 1–6 unstained tissue sections of 10-μm-thickness were cut, mounted on corresponding unstained slides and used for macro-dissecting the tumor tissue by removing the non-tumor tissue outside the circled area. When the tumor surface area measured ≥100 mm², a single 10 μm slide mounted tissue section was used as an input to ensure sufficient tumor content for protein extraction, whereas 3 slides were used as an input when the tumor surface measured 30–100 mm² and 6 slides when the tumor surface area measured 4–30 mm². Overall, the 1–6 macro-dissected 10 μm sections per case were submitted in 1.5 mL Eppendorf tubes for protein analyses.

*Preparation of tissue samples for SP3 processing.* FFPE tumor samples were spun down for 1 min at 20,000 *g* and deparaffinized by adding 800 μL xylene (Sigma, cat# 534056) twice. Xylene supernatant was removed, sections were air dried for 10 min in a fumehood, and then stored at −20 °C prior to lysis. Deparaffinized tumor tissue samples were homogenized in a lysis buffer containing 500 mM Tris-Cl pH8, 2% (wt/vol) SDS (Bio-Rad, cat# 1610302), 1% (vol/vol) Triton X-100 (Sigma, cat# T8787), 1% (vol/vol) NP-40 (Merck Millipore, cat# 492016), 5 mM EDTA (Thermo Scientific, cat# 15575020), 50 mM NaCl (Sigma, cat# S7653), 10 mM TCEP (Sigma, cat# C4706) and 40 mM CAA (Sigma, cat# C0267). 100 μL lysis buffer was added to each sample with 1–2 sections and 200 μL to samples of 3–6 tissue sections[24]. Samples were incubated for 90 min at 95 °C in a Thermo-Mixer (Eppendorf) with mixing at 1000 rpm. After incubation, tubes were allowed to cool down for 10 min at room temperature.

*SP3 processing, protein clean-up and digestion.* Protein samples were processed per the SP3 protocol as previously published[18,20,24]. A combination of two different types of SP3 paramagnetic beads (GE Healthcare, cat# 45152105050250 and cat# 65152105050250), both with a hydrophilic surface (Sera-Mag Speed Beads, GE Life Sciences), were prepared by taking 10 μL of each bead stock per sample to be processed and combined in a single tube. Beads were rinsed with water and reconstituted in 10 μL of water per sample. 12 μL of the rinsed SP3 beads were then added to a fresh 1.5 mL tube for each sample. Tissue samples were centrifuged at

20,000 *g* for 5 min, and supernatants were recovered to the 1.5 mL tubes with the SP3 beads and mixed by pipetting.

Protein binding was induced by adding ethanol (Sigma, cat# E7023) to a final concentration of 50% v/v. Tubes were incubated at room temperature for 10 min with mixing at 1000 rpm in a ThermoMixer to allow the binding to occur. Tubes were then placed on a magnetic rack, the supernatant was discarded and the beads were rinsed off-magnet twice with 200 μL of 80% ethanol, discarding the supernatant each time. Beads were resuspended in 100 μL of 200 mM HEPES pH 8 (Sigma, cat# H3375) containing trypsin/Lys-C mix (Promega, cat# V5071) at a ratio of 1:50 (μg/μg) protein to enzyme concentration and incubated for at least 14-h (overnight) at 37 °C in a ThermoMixer with mixing at 1000 rpm. The eluted peptides were recovered by clearing the supernatant with a magnetic rack and the supernatant containing the peptides was transferred to 1.5 mL tubes and stored at −20 °C until TMT labeling.

*Tandem mass tag labeling of peptides and design of 11-plex batches.* 11-plex TMT labeling kits (Thermo Scientific, cat# A37725) were used for TMT labeling of the peptide solutions derived from the SP3 digests. Each sample was labeled with 1 of the 11 specific isobaric TMTs. These labels have the same mass, but differ in the number of $^{12}C/^{13}C$ and $^{14}N/^{15}N$ isotopes in the mass reporter allowing for multiplex analysis[74]. Each 11-plex batch was designed to include a normal reduction mammoplasty sample (TMT[11]-126), 2 samples of each of the tumor PAM50 subtypes (luminal A (TMT[11]-127N, TMT[11]-127C), luminal B (TMT[11]-128N, TMT[11]-128C), basal-like (TMT[11]-129N, TMT[11]-129C), Her2-Enriched (TMT[11]-130N, TMT[11]-130C)), a SuperMix control consisting of 13 cancer cell lines (TMT[11]-131N), and a pooled internal standard (PIS) made up of aliquots from the tumor and normal samples plus isoDoping peptides (see below) (TMT[11]-131C). Each 5 mg TMT reagent was reconstituted in 500 μL of acetonitrile (Sigma, cat# 34851) at a final concentration of 10 μg/μL. Labeling reactions were carried out at room temperature in two volumetrically equal steps that included the addition of 10 μL of 10 μg/μL TMT reagent to each sample tube and incubation for 30 min. Reactions were then quenched through the addition of 15 μL of 1 M glycine (Sigma, cat# G8898). Labeled peptides were concentrated on a SpeedVac centrifuge (Thermo Scientific) to remove excess acetonitrile, combined in a single sample for each 11-plex, acidified to 1% (v/v) TFA (Sigma, cat# T6508), and cleaned up with C18 SepPak (50 mg, Waters, cat# WAT054960) prior to HPLC fractionation.

*SuperMix cell lines composition and preparation.* The composition and the preparation of the SuperMix including cell line lysis, protein reduction, alkylation, clean-up and digestion was performed as previously published[75]. The SuperMix was composed of 13 cancer cell lines that represent a collection of different cancer origins: primitive neuronal (HEK 293T) (ATCC, CRL-3216), cervical adenocarcinoma (HeLa) (ATCC, CCL-2), osteosarcoma (U2OS) (ATCC, HTB-96), colorectal carcinoma (HCT-116) (ATCC, CCL-247), chronic myelogenous leukemia (K562) (ATCC, CCL-243), lung carcinoma (A549) (ATCC, CCL-185), hepatocellular carcinoma (HepG2) (ATCC, HB-8065), TNBC (Hs578t) (ATCC, HTB-126), ovarian cancer (TOV-21G) (ATCC, CRL-11730), ductal epithelioid carcinoma (PANC-1) (ATCC, CRL-1469), prostate adenocarcinoma (PC3) (ATCC, CRL-1435), acute T-cell leukemia (Jurkat) (ATCC, TIB-152) and melanoma (SK-MEL-2) (ATCC, HTB-67).

Cell pellets for the different cell lines were combined, lysed, reduced and alkylated with a buffer composed of 50 mM HEPES pH 8.5, 4 M guanidine hydrochloride (Sigma, cat# G4505-500G), 10 mM TCEP (Sigma, cat# C4706), 40 mM CAA (Sigma, cat# C0267), with the addition of 1X complete protease inhibitor—EDTA free (Sigma, cat# 4693132001) and 1X phosphatase inhibitor (Sigma, cat# 4906845001). The pellets were pipette mixed and kept on ice. Lysis mixtures were transferred to 2 mL FastPrep-24 Lysing D Matrix tubes (MP Biomedicals, cat# 116913050). Lysis mixtures were vortexed on the FastPrep-24 instrument (MP Biomedicals, 6 M/s, 45 s, 1 cycle) twice with a rest of 30 s between cycles. Tubes were centrifuged at 20,000 *g* for 1 min, and the supernatant was recovered to a low-bind 1.5 mL tube. Resultant lysates were incubated at 95 °C for 15 min in the thermomixer with mixing at 1200 rpm. The lysate was aliquoted at 100 μL and stored at −80 °C prior to digestion.

Prior to digestion, lysates were thawed, split into two aliquots of 50 μL and diluted to 500 μL with 450 μL of 0.2 M HEPES, pH 8. 2 μg of trypsin/Lys-C mix (Promega, cat# V5071) was added to each aliquot and incubated for at least 16 h at 37 °C with mixing at 1200 rpm. After incubation, the tubes were centrifuged at 20,000 *g* for 1 min and the supernatants were recovered.

*Design of synthetic peptides and isoDoping library.* Due to stochastic sampling in data-dependent acquisition used in the SP3-CTP method, peptides of interest may be detected only in a subset of samples due to their low abundance relative to other peptides in the samples. We compiled a list of biologically important proteins from published gene signatures of interest in breast cancer that might be translated to biologically important and clinically relevant protein targets. These included the PAM50 gene signature[4], proteins used in TCGA to characterize breast cancer subtypes using RPPA[8], and additional clinically relevant proteins (Supplementary Data 1b). A pool of 706 synthetic peptides corresponding to 179 biologically important proteins were spiked into the PIS channel of each 11-plex set to boost the MS1 response and ensure that the combined MS1 signal for the ions of these peptides is above the selection threshold for MS2.

The isoDoping method utilizes the same additive MS1 properties of the isobaric TMT tags used to enhance the untargeted detection of low abundance proteins in single cells in the SCoPE-MS method[76] or to enhance the detection of untargeted phosphoproteins in the iBASIL method[77]. However, our isoDoping method is designed to include the quantification of specific proteins within a global proteome profiling method that could be readily implemented in standard TMT-based proteome profiling experiments.

Using this isobaric peptide doping (isoDoping) method, we doped for 3–5 proteotypic peptides/protein to ensure that we obtained signal for at least 2–3 peptides/protein for accurate protein quantification. The set of synthetic peptides was selected to fulfill the following criteria: (i) unique for the protein, and (ii) peptides should be between 6 and 20 amino acids long and have physiochemical properties amenable to MS detection. All the 706 isoDoping peptides were combined into a single mixture, and a volume corresponding to ~4.26 pmol of each peptide was spiked into the PIS sample pool (made up of aliquots from all tumor and normal samples) in each 11-plex and labeled together with that channel.

*High-pH reversed-phase fractionation.* Samples were fractionated using high pH C18 reverse phase high-performance LC (Agilent 1100) to decrease the complexity of each sample injected to the MS, as previously published[20] and as described below.

Fractionation was performed on a Kinetix EVO-C18 column (2.1 × 150 mm, 1.7 μm core shell, 100 Å, Phenomenex, cat# 00F-4725-AN). Columns were heated to 50 °C using HotSleeve column ovens (Analytical Sales and Services). Elution was performed at a flow rate of 0.25 mL per minute using a gradient of mobile phase A (20 mM ammonium bicarbonate, Sigma, cat# 09830) and B (acetonitrile, Sigma, cat# 34851). The gradient profile was 5% B for 5 min, followed by an increase to 8% B over 3 min and a second linear gradient to 30% B over 27 min. The %B was increased to 40% over 10 min then increased to 80% B over 1 min and held for 5 min to wash the column. The mobile phase concentration was decreased to 5% B over 1 min then held at 5% B for 18 min to re-equilibrate the column. Fractions were collected every minute from 5 to 53 min resulting in 48 individual fractions that were concatenated into 12 final fractions (fraction 1 = A1, B1, C1, D1; fraction 2 = A2, B2, C2, D2, etc.). Fractions were then dried in a SpeedVac and prepared for MS analysis by reconstituting the samples in 20 μL of 0.1% formic acid (Sigma, cat# 33015).

*Mass spectrometry data acquisition.* Analysis of labeled peptides was performed on an Orbitrap Fusion Tribrid MS platform equipped with an Easy-nLC1000 (Thermo Scientific) as previously described by Hughes et al.[20,78] and described below using control software version 3.1.2412.17 in data-dependent mode with MS2 scan. Data were collected for all 38 samples of a single fraction in a randomized order on the MS to reduce batch effects over the course of data acquisition. Columns used for trapping and analytical separations were packed with C18 (Reprosil-Pur, Dr. Maisch, 3 μm particle size, cat# r13.aq) and fritted (formamide and Kasil, 1:3 ratio heated for 15 min at 60–90 °C) in-house in 100 μm i.d., 360 μm o.d. polyimide coated fused silica capillary (Molex).

The trapping column was equilibrated with 10 μL of mobile phase A (0.1% formic acid in HPLC water) and the analytical column with 3 μL of mobile phase A. 2 μL of each fraction was loaded on the trapping column. Trapping was carried out on a 1–2 cm column for a total volume of 15 μL of mobile phase A at a pressure of 400 bar. After trapping, gradient elution of peptides was performed on a 20 cm analytical column heated to 45 °C using AgileSLEEVE column ovens (Analytical Sales & Services).

Elution was performed using a water-acetonitrile gradient at a flow rate of 450 nL/min. Elution was performed with a gradient from 3 to 7% mobile phase B (acetonitrile and 0.1% formic acid) over 2 min, 7–25% B over 94 min, and to 40% B over 17 min. The percent mobile phase B was increased to 80% over 1 min and the column was washed at 80% B for 6 min. MS data were acquired on the Orbitrap Fusion (control software version 3.1.2412.17) in data-dependent mode. 2.4 kV was applied to the nanoelectrospray source to generate ions; ion transfer tube temperature was 325 °C. MS1 scans were acquired in the Orbitrap at a resolution of 120,000 over a mass range of 400–1200 $m/z$ with an RF lens setting of 60%, an automatic gain control (AGC) target value of $4 \times 10^5$, and a maximum ion injection time of 120 ms.

For MS2 scans, monoisotopic precursor selection was set to peptides, charge state filtering was limited to 2–4, and undetermined charge states were included. Dynamic exclusion of selected masses was enabled after 1 observation for 15 s with a tolerance of 20 ppm.

High energy collision-induced dissociation fragmentation was performed with a quadrupole isolation window of 1.4 $m/z$ and a collision energy of 40%. Data were acquired in the Orbitrap in the normal scan range with a resolution of 50,000, a fixed first mass of 120 $m/z$, an AGC target of $1.2 \times 10^5$, and a max injection time of 86 ms with 1 microscan in centroid mode.

## Bioinformatics and statistical analysis

*Mass spectrometry data analysis.* For analysis, Proteome Discoverer Software (ver. 2.4) was used. Spectra were searched using Sequest HT against a combined UniProt Human proteome (03/08/2018) database appended to a list of common contaminants (20,387 sequences). Identification parameters in the Sequest HT were

specified as trypsin enzyme, two missed cleavages allowed, minimum peptide length of 6, precursor mass tolerance of 10 ppm, and a fragment mass tolerance of 0.05 Daltons. Oxidation of methionine, methylation of lysine, and TMT at lysine were set as variable modifications. Carbamidomethylation of cysteine and TMT at peptide N-terminus were set as fixed modifications. Peptide spectrum match (PSM) identification FDR was calculated using Percolator by searching the results against a decoy sequence set and only PSMs with FDR < 1% were retained in the analysis. The resulting protein set was filtered out for contaminant and decoys. Proteins identified by 2 or more peptides, of which at least one was unique, and with an FDR < 0.05 were retained for downstream analysis. Only proteins quantified across all 38 plexes were included in the dimensionality reduction and clustering analyses while proteins quantified in at least 75% of the samples were retained for differential expression and GSEA analyses.

*Peptide and protein quantification.* Reporter abundance quantification was reported as signal to noise ratio (S/N). All PSM reporter ion values were corrected for isotopic impurities provided by the manufacturer.

In order to remove samples with low signal, we set a minimum filter (2e06) on total S/N and removed 14 samples that did not pass this quality control threshold from the analysis.

For protein quantification, only unique PSMs with an abundance for the PIS channel >10 were used. Abundances for isodoped peptides were corrected based on the ratio of the median abundance of all isodoped peptides over endogenous peptides quantified in isodoped proteins. In order to correct for loading differences, all channel total abundance was scaled to 1e08 and PSMs abundances were divided by the relative PIS value in each TMT plex. For each protein, the median ratio of the 5 most abundant PSMs, determined by average reporter S/N, was used as relative abundance.

*Differential expression and gene set enrichment analysis.* In order to detect which proteins were differentially expressed between conditions, peptide-level expression-change averaging (PECA)[79] analysis was performed at the peptide level, using no normalization and modified t-statistic as parameters. While PECA can achieve improved or comparable overall performance with other differential expression methods[80], it can obtain very low p-values for proteins with a high number of quantified peptides.

GSEA was performed with the R package fgsea[81] (minsize = 2, maxsize = 500, nperm = 10000), using the GO term signature ('c5.all.v6.0') derived from the Molecular Signature Database (MSigDB) and REACTOME pathways from ('c2.cp.reactome.v6.0')[82].

*Consensus clustering.* The consensus clustering function from ConsensusClusterPlus R package[27,83] with K-means algorithm was applied on log2 relative abundance data from the 25% most highly variable (median absolute deviation) proteins in each dataset. Multiple iterations (5000) of K-means clustering, with K range = 2–12, were performed using the consensus clustering algorithm. The number of final clusters used was determined based on inspection of consensus matrix and examining the change in consensus CDF area with delta plots.

Clustering results were visualized by ComplexHeatmap r package[84].

For each protein cluster, the most representative terms were selected and presented on heatmaps based on g:profiler[85] enrichment analysis with the following parameters: organism = "hsapiens", ordered_query = FALSE, multi_query = FALSE, significant = TRUE, exclude_iea = TRUE, measure_underrepresentation = FALSE, evcodes = TRUE, user_threshold = 0.05, correction_method = "g_SCS", domain_scope = "annotated", custom_bg = NULL, numeric_ns = "", sources = NULL, term_size <150 and source in GO:MF, GO:BP or REACTOME'.

**Immunohistochemistry and scoring.** A series of three tissue microarrays previously constructed from the corresponding FFPE blocks used in the study were stained for key protein biomarkers selected for verification by IHC. These tissue microarrays were constructed using duplicate 0.6-mm tissue cores; serial 4 μm sections from each tissue microarray were stained for S100A8, TAP1, IFIT2, HLA-DQA1 and CD8 according to the Discovery XT semi-automated immunostainer protocol (Ventana medical Systems Inc. Tucson, AZ USA). The following antibodies were used: anti-S100A8 monoclonal mouse primary antibody (clone 749916, dilution 1:1000, R&D Systems, cat# MAB4570); anti-TAP1 polyclonal rabbit primary antibody (dilution 1:250, Proteintech, cat# 11114-1-AP); anti-IFIT2 polyclonal rabbit primary antibody (dilution 1:1000, Abcam, cat# ab113112); anti-HLA-DQA1 monoclonal rabbit primary antibody (clone [EPR7300], dilution 1:500, Abcam, cat# ab128959); anti-CD8 monoclonal mouse primary antibody (clone [C8/144B], dilution 1:50, Dako, cat# M7103).

Slides underwent antigen retrieval with standard Cell Conditioning 1 (Ventana Medical Systems) followed by 60 min of primary antibody incubation with heat, and detected using a DAB Map Detection Kit (Ventana Medical Systems). Slides were then incubated with a secondary antibody (Ventana universal secondary antibody) for an additional 32 min. Scoring of the S100A8, TAP1, IFIT2 and HLA-DQA1 IHC biomarkers were reported using the H scoring system (intensity × positivity) for the cytoplasmic staining observed in the invasive breast tumor cells. Intensity scores were reported as (0: none, 1: weak, 2: moderate, 3: strong) and the positivity proportion

scores were reported as (1–100%) for each core. The averaged H.score between the duplicate cores per case was used for the scoring of the protein expression by IHC. H.score was analyzed as a continuous variable against the different cluster groups.

For analysis of these IHC biomarkers as categorical variables, H-scores were dichotomized using cut-points optimized for best Cox model fit (assessed by Akaike Information Criterion; internally validated on 500 bootstrap samples). S100A8 was binarized into high (H-score ≥ 10) vs. low expression (H-score < 10). IFIT2 was scored as high (H-score ≥ 202) vs. low (H-score < 202); TAP1 was scored as high (H-score ≥ 290) vs. low (H-score < 290); and HLA-DQA1 was scored as high (H-score ≥ 10) vs. low (H-score < 10). For the analysis of CD8, we scored CD8+ TILs in the intratumoral compartment using established, analytically validated IHC staining and interpretation assay[86,87]. A subsequent validation of the IHC biomarkers was performed on an independent set of 176 breast cancer cases with similar clinicopathological characteristics to the 08–13 cohort (Supplementary Table 2).

All biomarkers were independently scored by pathologists blinded to clinical outcome data. All slides were scanned digitally using a Bliss System (Olympus America, Lombard, IL, USA).

**IHC subtyping**. Breast cancer cases were assigned into different IHC subtypes following guidelines adopted by the 2011 St Gallen Consensus panel and as previously described[88,89]. Luminal A: ER+ and/or PR+, Her2−, and ki67 < 14%; luminal B: ER+ and/or PR+, Her2−, and ki67 ≥ 14% or ER+ and/or PR+, any ki67 and Her2+; Her2+: ER−, PR−, Her2+; triple-negative: ER−, PR−, Her2−; and core basal: ER−, PR−, Her2− and [EGFR+ or CK5+].

**Survival analysis**. For the univariate and multivariate survival analyses, hazard ratios were derived using Cox regression models and stratified log-rank tests with the endpoints of RFS and OS. RFS was defined as the time interval between the date of diagnosis of invasive breast cancer and detection of any disease recurrence (local, regional or distant). OS was defined as the time interval between the date of diagnosis of invasive breast cancer and death from any cause. Kaplan–Meier curves and forest plots were used to display survival outcomes according to group categories including subtype, cluster and protein expression status. Multivariate analysis was adjusted for pathological tumor size T1 vs. (T2 or T3), nodal status negative vs. positive, grade (1 or 2) vs. 3, age at diagnosis ≥50 years vs. <50 years, LVI negative vs. positive, hormone and Her2 receptor status.

The association of single protein expression with RFS and OS was also assessed as a continuous variable in the Cox regression models and stratified log-rank tests. All tests were performed 2-sided at a significance level of 0.05. Analyses including multiple comparisons were adjusted for multiple testing using the Benjamini–Hochberg (BH) FDR method. Statistical survival analyses were performed using R statistical software using the "survminer" and "survival" packages.

Survival analysis for *FABP7* mRNA expression was performed using the previously established KMplotter analysis platform curated from 35 Gene Expression Omnibus datasets[35] and accessed using (https://kmplot.com/analysis/). Kaplan–Meier survival curves were generated by partitioning cases according to the median mRNA expression.

**Reporting summary**. Further information on research design is available in the Nature Research Reporting Summary linked to this article.

## Data availability
The mass spectrometry proteomics data have been deposited to the ProteomeXchange Consortium via the PRIDE[90] partner repository with the dataset identifier/accession code PXD024322 ("PXD024322 [https://www.ebi.ac.uk/pride/]"). Mass spectrometry data were searched against the UniProt Human proteome (03/08/2018 release, 20387 sequences) database. Gene set enrichment analysis was performed using annotated signatures: GO term signature ('c5.all.v6.0') as described in Subramanian et al.[82] (available online)—unique identifier: https://www.gsea-msigdb.org/gsea/msigdb/index.jsp and REACTOME pathways ('c2.cp.reactome.v6.0') as described in Subramanian et al.[82] (available online)—unique identifier: https://www.gsea-msigdb.org/gsea/msigdb/index.jsp. An anonymized data file containing characteristics of the study datasets, proteome clusters, and protein scores, used and analyzed in this study can be found in Supplementary Data 1. Images from immunohistochemistry slides of tissue microarrays used in the study coded as 11-012 and 14-004 are available online for public access via the website of Genetic Pathology Evaluation Center—unique identifier: http://www.gpec.ubc.ca/prot. Clinical data for the patients included in this study are not publicly available per policy to protect patient privacy. Clinical data access including de-identified individual patient characteristics and survival outcomes can be made available for qualified researchers on a request that does not include revelation of identifiable patient information through the Genetic Pathology Evaluation Centre and Breast Cancer Outcome Unit of BC Cancer, upon completion of a Data Transfer Agreement and confirmation of ethical approval. This clinical information would include the patient characteristic variables as presented in Supplementary Data 2h, 4g. Requests or queries should be directed to the corresponding author. Queries for data access will be answered within a time frame required to ensure high quality assessment and coordination of the

proposed collaborative work and a first response can be provided within ~2 weeks. This study involved the collection and analysis of data from multiple publicly available datasets. The CPTAC publicly available breast cancer dataset used in this study are available in the Supplementary Information of Krug et al.[15] (available online)—unique identifier: https://doi.org/10.1016/j.cell.2020.10.036. The OSLO2 publicly available breast cancer dataset used in this study are available in the Supplementary Data of Johansson et al.[16] (available online)—unique identifier: https://www.nature.com/articles/s41467-019-09018-y#Sec15. Survival analysis for *FABP7* mRNA expression was performed using the previously established KMplotter analysis platform[35] (available online)—unique identifiers: (https://kmplot.com/analysis/) and (10.1007/s10549-009-0674-9). The remaining data are available within the article, supplementary data or as deposited at PRIDE[90]. Source data are provided with this paper.

## Code availability
R code used for proteomics data processing and analysis is available at GitHub through the following link https://github.com/glnegri/brca and the corresponding DOI is as follows: https://doi.org/10.5281/zenodo.5873584[91].

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

## Acknowledgements

We thank the following for their contributions: Dr. Basile Tessier-Cloutier for assistance in sample collection and pathology review; Dr. Alona Nakonechnaya for overall project coordination and management; Dr. Grace S.W. Cheng for assistance in isoDoping library design; Christine Chow and Angela Cheng for their assistance in sample preparation; Samuel Leung for pathological data access management. This study was supported by funds from the Canadian Cancer Society (grant #319465, G.B.M.) and (grant #705463, T.O.N). K.A. is supported by the Vanier Canada Graduate Scholarship-Canadian Institutes of Health Research.

## Author contributions

Study conception and design: G.B.M., S.K.L.C., T.O.N., C.S.H., K.A.; performed experiment or data collection: K.A., S.E.S.M., S.C., C.S.H., X.Q.W., D.G.; computation and statistical analysis: G.L.N., K.A.; data interpretation: K.A., G.L.N., G.B.M., S.K.L.C., T.O.N., C.B.G.; writing—original drafts: K.A., S.E.S.M., G.L.N.; writing—review and editing: all; supervision: G.B.M., S.K.L.C., T.O.N., C.B.G.; funding acquisition: G.B.M., S.K.L.C., T.O.N.

## Competing interests

S.K.L.C. reports receiving consulting fees from Novartis Pharma, Pfizer, Hoffman LaRoche, Merck, AstraZeneca, Eli Lilly. T.O.N. played a role in the development of the PAM50 gene expression classifier, which has been licensed to Veracyte Technologies. The other authors declare no competing interests.
