## [Peer Review File · Nature Communications]

Proteomic analysis of archival breast cancer clinical specimens identifies biological subtypes with distinct survival outcomesReviewers' Comments:

Reviewer #1:

Remarks to the Author:

Asleh et al have performed quantitative proteomics of 300 breast tumors from formalin fixed paraffin embedded (FFPE) material. The general idea and its potential value to the community of this work is great.

The main Merits of the paper are: 1) the acquisition of proteomics data from FFPE samples across a large number of breast cancer samples with clinical follow up that can serve as a resource, 2) directly linking protein based sample groups with immune infiltration to improved outcome, 3) suggestion of potential biomarkers for tumor groups 4) identification of 4 TNBC groups as previously suggested at the RNA level and linking the immune infiltrated subgroup to good outcome, 5) identification of 3 ER positive tumor groups with a stromal enriched group.

Limitations regarding merits above:

To function as a resource the data needs to be judged as robust.

To evaluate protein quantitative robustness, the number of peptides used for quantification per protein needs to be available and visualized. Now it is lacking from the supplementary data table with all ratios. A panel can also be added to figure 2 to show nr of psms/protein used for quantification.

The supermix is present in all TMT sets and should represent how well quantifications can be reproduced between TMT sets. Can the supermix data be use for robustness evaluation between the sets? For example a heatmap for overview, variation of supermix in relation to the breast samples and particular sets with deviation on supermix-sample.

An overview clustering of the 2 cohorts with replicates would also be useful to judge how the whole dataset behaves. Does the technical replicates cluster together?

The data should also confirm with previous knowledge, as ER, PR, HER2, MKI67 levels in different PAM50 subtypes, and this would be good to show in a supplementary figure.

Proteomics have previously identified immune infiltration in breast cancer subgroups without directly linking them to outcome (Krug 2020 Cell, Johansson et al 2019 Nat Comm). Tumor-infiltrating lymphocytes (TILs) have also been linked to better outcome in breast cancer subtypes (Dieci 2021 Cells). The strength of this study is the direct link between proteomics data with "immune hot" tumors and outcome.

Relation to published data

In general, anchoring the novel findings further, e.g. by validation of findings in other breast proteomics data sets would be valuable to show the usefulness of the data as a resource and strengthen the findings. There is several decent datasets published now on breast cancer proteome so this should be done.

The 4 TNBC groups are correlating to their suggested RNA based groups. To strengthen the finding of 4 TNBC subtypes, can they be identified also at the protein level, for example in Krug 2020 Cell data? How generalizable are the 3 ER positive tumor subgroups identified in the manuscript? The authors cite Krug 2020 Cell in the discussion as consistent with the stromal-enriched subtype. But to my knowledge, the data in the Krug paper don't show a separate luminal A subgroup enriched for stroma. Dennison 2016 CCR, however show a stromal subtype of ER positive tumors that are or mixed subtype but enriched in Luminal A with a favorable clinical outcome. Are the same proteins (in RPPA and your MS data) deterministic of the stromal subgroup?

IHC validation of S100A8, TAP1, IFIT2, HLA-DQA1 and CD8 as suggested biomarkers of immune infiltration and better outcome are done on the same cohort as the proteomics. To consolidate the findings, validation in an independent cohort would be valuable. Also, what is correlation between the MS data and the IHC validated markers? Are the MS protein levels also related to outcome?

Additional comments:

From introduction: "This method can query large FFPE material cohorts linked to outcome data, enabling comprehensive quantification of protein expression from lower input quantities of routinely-

available patient specimens, and employs a more highly efficient workflow than other MS-based methods for protein profiling of clinical FFPE tissues^{21,22}. "

Based on the data, the MS workflow seems efficient, but there is really no data to comparing all other methods to support your claim of "more highly efficient workflow than other MS-based methods...."? Many of the large MS proteomics groups have published their versions of FFPE sample preparation methods. See for example Coscia 2020 Modern Pathology, Griesser 2020 MCP, Marchione 2020 JPR, Zhu 2019 Molecular Oncology.

In the abstract and in figure one, 300 samples are mentioned as included in the study. The number is correct but it is bit misleading since it's divided up in 2 cohorts. The overview in Figure 1A is not useful since this collection of samples are not used together later on in the paper. The overview presented in fig S1A are much more useful since it gives an overview of the samples used together in each of the later analyses. Also the number of samples drop after QC and removal of replicates. To make it clearer for the reader I suggest you make a combination figure of fig S1A and S2H with the tumor characteristics and the numbers that make up each cohort used in the downstream analysis. Also include the info of how the TNBC cohort was made. This took time to figure out and with a figure outlining the 2 cohorts, it would be much clearer from the beginning for the readability of the entire paper. To make it even clearer one could add what type of analysis / aim you have with each cohort. There is also normal samples for which it is unclear of their purpose/how they are used. Did not find any comparison to the normal samples in the text?

PAM50 is defined both by RNA and by surrogate IHC markers in the manuscript. However, it is unclear when each definition is used in the manuscript, which makes it confusing to read at times.

The authors use a new method denoted isodoping, with the aim to increase the overlap of identifications between TMT sets. The dynamic range in the orbitrap is max 3 orders of magnitude and the practical with TMT is closer to 2 orders of magnitude. To the pool of samples, 4.26 pmol of each peptide is added as isodoping. What is the evidence that you have not added 2 orders of magnitude of your spike in peptide compared to the endogenous levels? Adding spike in peptide amounts in excess of 2 orders of magnitude would make the other TMT channels hover around background and lose quantitative accuracy. How is it checked that this don't affect the quantification used? Could the same scenario happen for the SuperMix channel?

The isodoping is presented in fig1. This to me indicates that it is one of the main concepts in the paper since if it comes in the first main figure. However, this is a technicality which the authors say that they are preparing a manuscript for and could be moved to supplementary.

It is also unclear how the isodoping peptides were selected. Usually peptides are selected due to their good ionization capabilities which could explain much of the results in fig 1b?

Figure 1C is unclear to me. How do you reach 74 isodoping dependent proteins? Can you update the figure legend or make a new clearer figure?

From results: "The cases in the 08-13 cohort were treated in accordance with contemporary guidelines and contained cases from all four PAM50 subtypes, including all 75 basal-like cases (Supplementary Figs. S1a, S2h, Supplementary Data S1d)."

Which contemporary guidelines are you referring to?

LVI, lymphovascular invasion is mentioned. Don't find it in materials & methods.

When the tumor groups are defined they are given numbers. However, when they are first introduced in figures (for example 2b & c, 5a, 7a) they are not in numerical order. It would maybe be much easier to follow if the clusters are renumbered in numerical order in the first figure where they appear.

The identification and quantification of 4214 proteins across all samples is a good result for MS

analysis of FFPE samples. But could some of the results be explained by not reaching deep enough into the proteome, considering that there should be around 14000 proteins in a tissue according to ProteinAtlas. Could this be a reason for the grouping of Luminal A tumors with Her2 in fig 2b? In the Krug et al Cell 2020 paper their tumor grouping almost exclusively only mix luminal A and Bs. No HER2 based on 7679 proteins quantified across all 122 samples. Can the lum A mixing with Her2 be reproduced with the same proteins? Or is this an effect of FFPE?

In fig S3a, you refer to biological replicates. How is biological replicates defined in clinical samples? For the technical replicates, it would have been better if they were spread out in different TMT sets.

The PAM50 subtypes have got standard color code. See TCGA 2012 Nature or Krug et al Cell 2020. To avoid confusion I strongly recommended to use the same color code.

In general the authors make a good job in describing their findings. But to make it easier to follow I would suggest to add ER, PR and HER2 status to fig 2C. For example in the text it says: "Most cases in Cluster-2 and -3 were associated with ER, PR and Her2 negativity by IHC clinical tests, high proliferation index (Ki67), and the "core basal" phenotype (defined as ER-, PR-, Her2- and [EGFR+ or CK5+])²⁹ (Supplementary Table 1)." Adding the clinicopathological markers to the heatmap in fig 2c would make it easy to see this in addition to the table. But this is a matter of taste and you can ignore if you like.

In fig 2c, 5a, there is a column called immune with 2 categories, Immune related and Other. How are they defined? Also, for the protein groups there are enrichments, how were the enrichments done? Specify in fig text how the terms were selected, representative/ cutoff?

Fig 2a, is this using all or the most varying proteins? In 2b it does not say that the grouping is based on consensus clustering.

In figure S2b-c, the authors show number of peptides per protein. Bit unclear to what it refers to when mentioning peptide? Is that unique peptides? Nr of peptides per protein, is that the per set or total across all TMT sets or mean/median?

The number of unique peptides per protein, nr of psm per protein and nr of psms/protein for TMT quantification is missing from the supplementary table with all MS data. Please add this, since it is important when it comes to judging the quantitative robustness. Having said that, must give all the credits for clear clinical information and that the authors include it in the same document so it is easy to access!

In figure 3e, the y-axis says abundance. Is this log2 ratio to the pool of samples? ESR1 is high in cluster 2 which is one of the basal enriched clusters, which is surprising. Could this be due to isodoping or poor quantification? KRT18 and FOXA1 on the other hand behave as expected.

PECA is used for calculating p-values. Wonder if that inflates the p-values and makes them smaller just because you have a lot of peptides per protein?

<https://pubs.acs.org/doi/10.1021/acs.jproteome.5b00363>

Full credit for uploading the immunohistochemistry slides to <http://www.gpec.ubc.ca/prot>. But why limit to representative images. For the dataset to be useful, all images needs to be available. In addition, there should be an easy way to download all data for image analysis.

To make the data analysis part transparent and reproducible, analysis code should be uploaded to Github or similar repository.

Orbitrap MS2 data was matched with 0.5 Daltons tolerance. This is a very large window that is usually

used for iontrap data. For orbitrap MS2, the tolerance should be around 0.02 Dalton to reduce the risk of miss assigning transitions. Since you are also using methylation of lysine as a variable modification, this in combination with a large tolerance will increase your FDR. You should research at least parts of the data and compare the results to your present results to determine if all data needs to be researched. In relation to this, what is the protein FDR of the dataset, q-value, pep value for each protein?

From results: FFPE samples were macro-dissected from 3-6 sections to obtain >80% tumor content and analyzed using the SP3-CTP multiplex MS proteomics protocol²⁴ (Supplementary Fig. S1b). Should it not be ref 19 instead of 24?

Reviewer #2:

Remarks to the Author:

In this study, the authors carry out mass-spec proteomic profiling of 300 FFPE breast cancer surgical specimens. The specimens are separated into two cohorts based on batch effects. The 08-13 cohort included 75 basal-like, 62 Her2-Enriched, 30 luminal B, and 11 luminal A PAM50 defined cases. The 86-92 cohort provided the long-term outcome data required for luminal cases and included 64 luminal A, 45 luminal B, and 13 Her2-Enriched PAM50 cases. The 08-13 cohort was used for subtype discovery, both across all tumors and within the TNBC subset. ER+ subtypes examined in the 08-13 cohort were examined in the 86-92 cohort.

Specific comments:

1. Batch effects were found between the 08-13 and 86-92 cohorts, likely due to differences in collection techniques, pre-analytical handling, and fixation procedures. Could the authors try to harmonize the two datasets using Combat (<https://rdrr.io/bioc/sva/man/ComBat.html>)? In practice, Combat is very good at removing batch effect differences. Data from different platforms (RPPA, RNA-seq, DNA methylation) have been successfully processed with Combat, and the method is independent of nature of the batch effect. The PAM50 subtype could be used as the experimental group. There would be advantages in having one harmonized dataset of 300 samples. It seems worth a try. As currently written, the Abstract suggests that there is one dataset that was analyzed, rather than two separate cohorts.

2. Page 8: "Cluster-1 (n=34) consisted mostly of luminal B and Her2-Enriched PAM50 cases. Clusters-2 (n=50) was enriched for basal-like subtype, included few Her2-Enriched, but had no luminal cases. Cluster-3 (n=47) was primarily basal-like cases but included Her2-Enriched cases. Cluster-4 (n=43) was mostly Her2-Enriched but included luminal A and luminal B cases." It seems that actual numbers to reflect the noted associations would be helpful here, e.g. exactly how many basal-like cases and Her2 cases were in Cluster-3, and was Cluster-2 SIGNIFICANTLY enriched for basal-like.

3. In general, where the word "significantly" appears in the main text, it would be good to include a p-value and associated test to support the claim. The figures referred to likely include the test, but reflecting this in the main text as well would be helpful to the reader. For example, page 11: "The immune hot cluster also had significantly higher CD8+ TILs in the intratumoral compartment compared to other clusters (Fig. 4a)." by what p-value and test?

4. Wherever a p-value appears in the main text, the test used to derive that p-value should also be indicated. For example, page 12: "The subgroups with a high expression for only one of these biomarkers were characterized with intermediate RFS (Supplementary Fig. S5b). 70% (21/30) of the cases classified as (TAP1 high/HLA-DQA1 high) were in Cluster-3, while 90% (76/84) of (TAP1 low/HLA-DQA1 low) cases were in other clusters (p-value<0.00001) (Supplementary Table 1)." What test was used here (we can save the reader from having to go the Table for the answer)?

5. Page 15: "Multiple correction testing identified fatty acid-binding protein-7 (FABP7) as a candidate biomarker most significantly associated with >10-year RFS on tamoxifen treatment..." Was this the only protein that was significant? Were other proteins significant and using what statistical test and cutoff?

6. Discussion, page 16. Many journals are uncomfortable with the phrase "(manuscript in preparation)." It seems that the method indicated should be described in sufficient detail in the Methods, if it isn't already.

7. In addition to making the raw data available on ProteomeXchange, it would be most helpful to include the processed proteinXsample tables as Supplementary Data with the published paper. CPTAC has done a similar thing with their past publications.

8. For boxplots in the figures, please define the ranges involved.

Reviewer #3:

Remarks to the Author:

The authors present their previously described highly sensitive MS-based methodology termed "Single-Pot, Solid-Phase enhanced, Sample Preparation"-Clinical Tissue Proteomics (SP3-CTP). This technology has been shown to capture known and novel features in FFPE tumor samples. The authors have previously shown that this method can be applied on large FFPE material cohorts linked to outcome data. Comprehensive quantification of protein expression can be achieved even from lower input quantities of patient specimens such as small biopsies. Here it would have been useful to know how small?

In this paper they have applied the method to 300 well-characterized archival FFPE breast cancer specimens in terms of clinical outcome, IHC, and PAM50 RNA-based intrinsic subtypes. The authors demonstrate that at the protein level one can identify groups characterized by high expression of immune-response proteins and favorable clinical outcomes.

Does this paper bring a sufficient novelty? While it is true that "classifications do not always guide therapeutic choices, due to the extensive heterogeneity that still characterizes breast cancers" can this be solved by adding one more, at the level of proteomics?

Q1. How does this extension to 300 cases add to what we know from Johansson et al NatComm, 2019?

Q2. How does the heterogeneity described here match what is known from RNA based classification (basal also divided in several immune clusters)

Q3. If the authors were to make biomarkers based on protein as they suggest, which ones would they choose?

An introduction of 5 pages and large number of references (81) makes it into a difficult read. This paper, as rigorously performed and described, would benefit from some clarity and simplification, just highlighting the results that move the field forward.

RESPONSE TO REVIEWER COMMENTS

Reviewer #1, expert in proteomics (Remarks to the Author):

Asleh et al have performed quantitative proteomics of 300 breast tumors from formalin fixed paraffin embedded (FFPE) material. The general idea and its potential value to the community of this work is great.

The main Merits of the paper are: 1) the acquisition of proteomics data from FFPE samples across a large number of breast cancer samples with clinical follow up that can serve as a resource, 2) directly linking protein based sample groups with immune infiltration to improved outcome, 3) suggestion of potential biomarkers for tumor groups 4) identification of 4 TNBC groups as previously suggested at the RNA level and linking the immune infiltrated subgroup to good outcome, 5) identification of 3 ER positive tumor groups with a stromal enriched group.

We appreciate the reviewer's view that the work will provide important value to both the fields of breast cancer and of proteomic analysis of patient samples in general.

Limitations regarding merits above:

1. To function as a resource the data needs to be judged as robust.

To evaluate protein quantitative robustness, the number of peptides used for quantification per protein needs to be available and visualized. Now it is lacking from the supplementary data table with all ratios. A panel can also be added to figure 2 to show nr of psms/protein used for quantification.

We thank the reviewer for this point. We have added the total number of peptides for each protein, number of unique peptides per protein and number of PSMs used in quantification per protein to the Supplementary Data S1c. We have also added the data on the peptide abundance per protein (now appears as new Supplementary Data S1d) and PSMs per protein in Supplementary Figure S2d.

Supplementary Figure S2

2. The supermix is present in all TMT sets and should represent how well quantifications can be reproduced between TMT sets. Can the supermix data be used for robustness evaluation between the sets? For example a heatmap for overview, variation of supermix in relation to the breast samples and particular sets with deviation on supermix-sample.

We thank the reviewer for this point. The 38 SuperMix replicates included in our experiment showed a high correlation across the 38 plexes. Unsupervised clustering of our data for all samples including breast tumors, normals, and SuperMix show the SuperMix samples clustered together and are clearly separated from the breast tumor and normal samples. The correlation between the SuperMix samples was the highest when compared to the breast tumor and normal samples, supporting the robustness of the evaluation of SuperMix samples across the sets (appears as new Supplementary Fig. S4c). Pairwise correlation between the 38 SuperMix replicates (ranged between 0.68-0.81, median 0.75) was significantly higher than the pairwise correlation across the 38 normals (ranged between 0.53- 0.85, median 0.71). These findings are shown in new Supplementary Fig. S4d.

The following information has been added to the results section page #8:

“An overview clustering of all the samples included in our study showed that the 38 SuperMix replicates had the highest correlation across the 38 plexes (range 0.68-0.81) when compared to the breast tumors and normal samples, supporting the robust quantification of SuperMix samples across the different sets (Supplementary Figs. S4c-S4d).”

Supplementary Figure S4

3. An overview clustering of the 2 cohorts with replicates would also be useful to judge how the whole dataset behaves. Does the technical replicates cluster together?

Per the reviewer's request, we generated a heatmap showing the overview clustering for all the samples, as also requested in the previous comment. As now shown in Supplementary Fig. S4c, the three technical replicates indeed have clustered adjacent to each other (T_rep 5, T_rep 6, T_rep 7). Regarding the biological replicates, 2 of 3 replicates clustered adjacent to each other while the 3rd biological replicates clustered very closely together, a variance in line with expectations for intratumoral regional sampling. The normal samples clearly separated from tumor samples and showed an overall correlation of 0.70. An overall correlation of 0.5-0.6 was observed for the different breast tumor clusters and these included a mix of samples from both 08-13 and 86-92 cohorts.

This information has been added to the results section page #8:

“All the technical replicates and 2 out of 3 biological replicates clustered adjacent to each other, while the 3rd biological replicates clustered very closely together, a variance in line with expectations for intra-tumoral regional sampling (Supplementary Fig. S4c).”

4. The data should also confirm with previous knowledge, as ER, PR, HER2, MKI67 levels in different PAM50 subtypes, and this would be good to show in a supplementary figure.

HER2 (ERBB2) and MKI67 expression levels across the different PAM50 subtypes are found in Supplementary Fig. S6d. ER (ESR1) and PR (PGR) expression levels across the different PAM50 subtypes are now also included in Supplementary Fig. S6d.

5. Proteomics have previously identified immune infiltration in breast cancer subgroups without directly linking them to outcome (Krug 2020 *Cell*, Johansson et al 2019 *Nat Comm*). Tumor-infiltrating lymphocytes (TILs) have also been linked to better outcome in breast cancer subtypes (Dieci 2021 *Cells*). The strength of this study is the direct link between proteomics data with “immune hot” tumors and outcome.

Relation to published data

In general, anchoring the novel findings further, e.g. by validation of findings in other breast proteomics data sets would be valuable to show the usefulness of the data as a resource and strengthen the findings. There is several decent datasets published now on breast cancer proteome so this should be done.

Per the reviewer's request, we performed a validation of our findings on previous proteomic datasets published by Krug et al. *Cell* 2020 (CPTAC) and Johansson et al. *Nat. Commun* 2019 (OSLO2).

Validation using the Krug et al. 2020 CPTAC breast tumor cohort: In order to compare our results with available published datasets, we performed consensus clustering with the same parameters used in our cohort on the CPTAC *Cell* 2020 cohort, using 939 proteins from the CPTAC data that overlap with the 1054 mostly highly-variant proteins of the 08-13 cohort. This analysis identified four main proteome clusters that highly resembled the original CPTAC NMF clusters of “LumA-I”, “LumB-I”, “Basal-I”, “HER2-I”. Two of these were almost entirely similar to the original NMF clusters of “Basal-I”, and “LumA-I”. Another cluster highly resembled NMF “LumB-I” and consistent with Krug et al consisted of 54% luminal A cases (compared to 55% luminal A cases assigned as “LumB-I” in the original NMF CPTAC clusters

by Krug et al). Similar to the original NMF CPTAC clustering composition, the NMF CPTAC “HER2-I” cluster identified had a mix of Her2-Enriched, luminal A and luminal B breast cancers. Of note, the original Krug et al 2020 study of 122 breast tumors included a majority of luminal A PAM50 subtype (n=57, 47%), followed by basal-like (n=29, 24%), luminal B (n=17, 14%) and Her2-Enriched (n=13, 11%) when compared to the composition of our 08-13 cohort consisting of a higher number of basal-like (n=73, 42%) and Her2-Enriched (n=62, 36%) cases, but few luminal A (n=11, 6%). Despite this, our analysis further reproduced the existence of subsets enriched for immune response pathways at the proteome level within the basal-like and Her2-Enriched subtypes not captured by CPTAC analysis. Consistent with our analysis on the 08-13 cohort, stromal pathways were enriched in luminal A tumors and lipid metabolism was enriched within luminal B and Her2-Enriched tumors. A description of these findings is displayed in Supplementary Fig. S9a.

Validation using the Johansson et al 2019 “OSLO2 breast cancer landscape cohort”:

Validating our findings on the 36 cases of the 4 main subtypes (9 for each PAM50 type) on “OSLO2 landscape cohort”, we performed consensus clustering with the same parameters used in our analysis, using 775 proteins from the OSLO2 data that overlap with the 1054 mostly highly-variant proteins of the 08-13 cohort. This analysis identified 4 clusters that highly resembled the main consensus core tumor clusters (CoTCs) and their biological functions as reported in Johansson et al. These clusters consisted of CoTC1 (basal-like immune cold), CoTC2 (basal-like immune hot), CoTC3 with few CoTC6 cases (luminal A-enriched) and CoTC6 (luminal B and Her2-Enriched). Importantly, the immune distinctions within the basal-like subtype were entirely reproduced using our highly variant proteins showing that the two basal-like samples of OSL.3EB and OSL.449 (CoTC2) were consistently classified as “basal immune hot cluster” when compared to other basal cases characterized as “basal immune cold”. These findings are displayed in Supplementary Fig. S9b.

The results section page #14 has been updated to include our comparison analysis using the Krug et al 2020 and Johansson et al 2019 proteomics datasets, as a new section entitled “**Comparison with previous breast cancer proteomics studies**”.

Supplementary Figure S9

6. The 4 TNBC groups are correlating to their suggested RNA based groups. To strengthen the finding of 4 TNBC subtypes, can they be identified also at the protein level, for example in Krug 2020 *Cell* data?

We validated our TNBC proteome clusters using the 935 proteins that overlap with the 1055 mostly highly-variant proteins in our 08-13 TNBC (n=88) subset on a set of 28 TNBC cases included in the CPTAC breast cancer cohort by Krug et al. Our analysis reproduced the existence of the four main proteome TNBC subgroups and the biological features of ‘luminal-androgen receptor’, ‘mesenchymal’, ‘basal-immune suppressed’, and ‘basal-immune activated’ as now shown in Supplementary Fig. S11.

The results section page #16 has been updated to include this information:

“The existence of these TNBC proteome clusters and their biological features were validated when applying consensus clustering, with identical parameters, on 935 proteins overlapping with the 1055 mostly highly-variant proteins of the 08-13 TNBC subset on the proteomic data for a set of 28 TNBC cases included in the CPTAC breast cancer cohort by Krug et al (Supplementary Fig. S11).”

Supplementary Figure S11

7. How generalizable are the 3 ER positive tumor subgroups identified in the manuscript? The authors cite Krug 2020 *Cell* in the discussion as consistent with the stromal-enriched subtype. But to my knowledge, the data in the Krug paper don't show a separate luminal A subgroup enriched for stroma. Dennison 2016 *CCR*, however show a stromal subtype of ER positive tumors that are or mixed subtype but enriched in Luminal A with a favorable clinical outcome. Are the same proteins (in RPPA and your MS data) deterministic of the stromal subgroup?

We agree with the reviewer that Krug 2020 *Cell* did not identify a separate stromal enriched subtype as a unique cluster by mass spectrometry, but described a subset of luminal A tumors as stromal-enriched since these tumors were classified originally in TCGA 2012 based on RPPA data as "reactive". In the subsequent *Nature* 2016 CPTAC proteomics profiling breast cancer publication, the proteomic cluster that was highly correlated with the "reactive" RPPA cluster was referred to as stromal-enriched.

The Dennison 2016 *CCR* basically tried to characterize the biological and clinical features of the stromal enriched tumors as a whole (i.e. reactive tumors) identified in the TCGA based on RPPA data. The majority of these tumors were found to be classified as luminal A by PAM50, and among the luminal A as a group those that had high stromal protein expression displayed favorable clinical outcomes.

Comparing the proteins in our MS data that are in common with the RPPA proteins (n=30) used to classify the "stromal-enriched" vs. the "ER positive cancer derived" subtypes in Dennison 2016 *CCR*, we found 5 proteins in the RPPA "stromal-enriched" Dennison 2016 *CCR* that were also characteristic for our luminal A stromal enriched proteomics cluster ($\log_2FC > 0.20$, adjusted p-value < 0.05). These were fibronectin, annexin, collagen VI, caveolin, and MYH11.

While our data correlate with those results, the RPPA data only cover a small percentage of the proteome that was quantified in our experiment; thus, our data characterize the luminal A stromal enriched cluster in a more comprehensive manner and identify protein candidates that are beyond those captured by the restricted number of proteins in the antibody-based RPPA assay.

The discussion page #22 has been updated to highlight this information.

"Our analysis of ER+ cases with mature clinical data identified a stromal-enriched subset (86-92-Cluster-2) consistent with previous reports^{56,62}, which could help sub-classify luminal breast cancer. However, our data characterize the luminal A stromal enriched cluster in a more comprehensive manner and identify protein candidates that are beyond those captured by the restricted number of proteins in the antibody-based RPPA assay".

8. IHC validation of S100A8, TAP1, IFIT2, HLA-DQA1 and CD8 as suggested biomarkers of immune infiltration and better outcome are done on the same cohort as the proteomics. To consolidate the findings, validation in an independent cohort would be valuable. Also, what is correlation between the MS data and the IHC validated markers? Are the MS protein levels also related to outcome?

First part of the reviewer's comment: Per the reviewer's request, we have now performed a validation of these IHC biomarkers on an independent set of 176 breast cancer cases with similar clinicopathological characteristics to the 08-13 cohort. Our analysis confirmed that high expression of HLA-DQA1 as a single biomarker had a significantly better survival (log-rank

p=0.02) and a similar trend was seen with high TAP1 as a single biomarker (log-rank p=0.09). The findings further confirmed that tumors with IHC expression for both TAP1 and HLA-DQA1 showed the most favorable survival, while the subgroup with low expression for both had the worst RFS (log-rank p=0.05) (Supplementary Fig. S8).

Supplementary Figure S8

The results section page #13 has been updated with this information:

“We subsequently confirmed our observations on an independent, clinically similar set of 176 breast cancer cases and showed that high expression of HLA-DQA1 as a single biomarker had a significantly better survival (log-rank $p=0.02$) and a trend was seen for high TAP1 as a single biomarker (log-rank $p=0.09$). These data also confirmed that tumors with high IHC expression for both TAP1 and HLA-DQA1 showed the most favorable survival, while the subgroup with low expression for both had the worst RFS (log-rank $p=0.05$) (Supplementary Table 3; Supplementary Fig. S8).

The Supplementary methods in the Supplementary Information file page #25 and Supplementary Table 3 include information on the characteristics of this IHC validation cohort:

“IHC validation cohort: A tissue microarray for an independent set of 176 breast cancer cases was used to validate observations on the 08-13 cohort for the key protein IHC biomarkers. This validation cohort had clinicopathological characteristics similar to the 08-13 cohort and was analyzed for IHC biomarker association with clinical outcomes. The median follow-up for the IHC validation cohort was 10 years and cases were treated in accordance with contemporary guidelines”. Characteristics of this cohort appear in the new updated Supplementary Table 3.

Supplementary Table 3

Characteristic	IHC Validation cohort (n=176)
Age at diagnosis (median)	53 years
Tumor size (median)	2 cm
Tumor grade	
1, 2	44 (25%)
3	127 (72%)
Missing	5 (3%)
Nodal status	
Negative	105 (60%)
Positive	66 (37%)
Missing	5 (3%)
IHC subtype	
Luminal ([ER+ or PR+])	69 (39%)
ER-, PR-, HER2+	32 (18%)
ER-, PR-, HER2-	71 (40%)
Missing	4 (3%)
Disease specific death	
No	134 (76%)
Yes	35 (20%)
Missing	7 (4%)
CD8 iTILs	
<1%	42 (24%)
≥1%	129 (73%)
Missing	5 (3%)
TAP1/HLA-DQA1 IHC groups	
TAP1 high /HLA-DQA1 high	35 (20%)

TAP1 low /HLA-DQA1 high	22 (13%)
TAP1 high /HLA-DQA1 low	50 (28%)
TAP1 low /HLA-DQA1 low	65 (37%)
Missing	4 (2%)

Second part of the reviewer’s comment: The Spearman correlation between the MS data and the H score for the IHC validated markers was found to be 0.51 for TAP1 and S100A8, 0.31 for HLA-DQA1, and 0.11 for IFIT2 as shown in the figure below. Of note, the assessment of the validated markers by IHC was performed on the carcinoma cells.

Third part of the reviewer’s comment: The selection of the biomarkers for IHC validation was based on biology rather than clinical outcomes. In response to the reviewer’s comment, we performed a Cox proportional-hazards analysis on the protein abundance (in MS data) and recurrence free survival for the protein candidates we assessed by IHC. MS protein levels are significantly correlated with improved outcome for TAP1 and IFIT2, while a trend is shown for HLA-DQA1 and S100A8 as follows:

Protein	Survival analysis for RFS HR (95% CI), P-value	Adjusted P-value
TAP1	0.34 (0.18-0.65), 0.001	0.04
HLA-DQA1	0.87 (0.69-1.10), 0.24	0.71
S100A8	0.87 (0.73-1.06), 0.16	0.62
IFIT2	0.38 (0.18-0.80), 0.01	0.19

Additional comments:

9. From introduction: “This method can query large FFPE material cohorts linked to outcome data, enabling comprehensive quantification of protein expression from lower input quantities of routinely-available patient specimens, and employs a more highly efficient workflow than other MS-based methods for protein profiling of clinical FFPE tissues^{21,22}. “
Based on the data, the MS workflow seems efficient, but there is really no data to comparing all other methods to support your claim of “more highly efficient workflow than other MS-based methods...”? Many of the large MS proteomics groups have published their versions of FFPE sample preparation methods. See for example Coscia 2020 Modern Pathology, Griesser 2020 MCP, Marchione 2020 JPR, Zhu 2019 Molecular Oncology.

We thank the reviewer for this comment. We have updated this sentence in the introduction page #5 accordingly:

“This method can be used to query large FFPE material cohorts linked to outcome data, enabling comprehensive quantification of protein expression from lower input quantities of routinely-available patient specimens, and employings a ~~more~~ highly efficient workflow than other MS-based methods for protein profiling of clinical FFPE tissues^{21,22}.

10. In the abstract and in figure one, 300 samples are mentioned as included in the study. The number is correct but it is bit misleading since it's divided up in 2 cohorts. The overview in Figure 1A is not useful since this collection of samples are not used together later on in the paper. The overview presented in fig S1A are much more useful since it gives an overview of the samples used together in each of the later analyses. Also the number of samples drop after QC and removal of replicates. To make it clearer for the reader I suggest you make a combination figure of fig S1A and S2H with the tumor characteristics and the numbers that make up each cohort used in the downstream analysis. Also include the info of how the TNBC cohort was made. This took time to figure out and with a figure outlining the 2 cohorts, it would be much clearer from the beginning for the readability of the entire paper. To make it even clearer one could add what type of analysis / aim you have with each cohort. There is also normal samples for which it is unclear of their purpose/how they are used. Did not find any comparison to the normal samples in the text?

Per the reviewer's recommendation we have moved the original Figures S1A and S2H to Figure 1. Now they appear as Fig. 1b and Fig. 1c.

Given that normals were sourced from independent reduction mammoplasties, they are very biologically different from tumors and thus they are not helpful in the subtyping or performing direct comparisons with tumor samples. The normals were included in the UMAP plots where they form a clearly separated cluster from tumors, added to the heatmaps (Figures 2c and 5a) as a reference to illustrate that proteins and pathways of interest for the proteome clusters were not high in normals, and as a visual comparator for the expression of key breast cancer associated proteins in Supplementary Figure S6d.

In addition, when we picked specific proteins of interest for validation in IHC, we used candidates that were not highly expressed in normals. We updated the text to include this specific information in page #12.

*“We selected four that were among the top differentially-expressed proteins between the immune hot cluster vs. others (Supplementary Data S2c), had available antibodies applicable to FFPE, and had a practical scoring methodology on carcinoma cells: TAP1 (MHC class I), HLA-DQA1 (MHC class II), IFIT2 (type I interferon signaling) and S100A8 by IHC (Figs. 4b-4c). **In addition, these proteins were not highly expressed in the normal reduction mammoplasty samples**”.*

11. PAM50 is defined both by RNA and by surrogate IHC markers in the manuscript. However, it is unclear when each definition is used in the manuscript, which makes it confusing to read at times.

PAM50 per definition only refers to RNA not IHC as PAM50 is a RNA-based assay. There is no definition of PAM50 by IHC in the manuscript. We have however now added the word “RNA-based” before the word PAM50 in the section that included IHC data for further clarity.

page #15: “We analyzed 88 IHC defined TNBC cases (profiled by RNA-based PAM50 as: 61 basal-like, 22 Her2-Enriched, and 5 luminal B), all in the 08-13 cohort (Fig. 1b)

12. The authors use a new method denoted isodoping, with the aim to increase the overlap of identifications between TMT sets. The dynamic range in the orbitrap is max 3 orders of magnitude and the practical with TMT is closer to 2 orders of magnitude. To the pool of samples, 4.26 pmol of each peptide is added as isodoping. What is the evidence that you have not added 2 orders of magnitude of your spike in peptide compared to the endogenous levels? Adding spike in peptide amounts in excess of 2 orders of magnitude would make the other TMT channels hover around background and lose quantitative accuracy. How is it checked that this don't affect the quantification used? Could the same scenario happen for the SuperMix channel?

The reviewers make an astute point that issues with the dynamic quantification range can arise when implementing TMT. As shown in Supplementary Fig. S2f, when we compared the average S/N ratio, before normalization, across different sample types we detected an average difference of 3.7x between SuperMix and tumor samples, with all SuperMix samples showing an average S/N comparable to the tumor samples with higher signal.

In Supplementary Fig. S2g, it is displayed that there is only a 3.2x difference between the average abundance of isoDoped peptides and endogenous peptides for isodoped proteins in the PIS+isoDoping channel. When comparing the average S/N of the isoDoping peptides in the tumor samples and the spiked in channel we detected an 8.6x difference, below the suggested limit of 20x (Cheung TK et al. “Defining the carrier proteome limit for single-cell proteomics” *Nature Methods*, 2021).

13. The isodoping is presented in fig1. This to me indicates that it is one of the main concepts in the paper since if it comes in the first main figure. However, this is a technicality which the authors say that they are preparing a manuscript for and could be moved to supplementary.

Per the reviewer's recommendation we have moved the isodoping performance to Supplementary Figure S2.

Supplementary Figure S2

14. It is also unclear how the isodoping peptides were selected. Usually peptides are selected due to their good ionization capabilities which could explain much of the results in fig 1b?

Figure 1C is unclear to me. How do you reach 74 isodoping dependent proteins? Can you update the figure legend or make a new clearer figure?

As elaborated on in the methods section, the set of synthetic peptides was selected to fulfill the following criteria: (i) include unique peptides for the protein, and (ii) peptides should be between 6 and 20 amino acids long and/or (iii) have physiochemical properties amenable to MS detection. Our isoDoping methodology has been updated and improved in following subsequent experiments for which we have a manuscript under review and can be made available upon request once it is in pre-print. We have also removed Figure 1C from the manuscript.

15. From results: “The cases in the 08-13 cohort were treated in accordance with contemporary guidelines and contained cases from all four PAM50 subtypes, including all 75 basal-like cases (Supplementary Figs. S1a, S2h, Supplementary Data S1d).”

Which contemporary guidelines are you referring to?

The contemporary guidelines refer to the updated recent guidelines recommended to treat breast cancer commonly used in practice. A reference (Cardoso F et al. Early breast cancer: ESMO Clinical Practice Guidelines for diagnosis, treatment and follow-up. *Annals of Oncology* 2019) has now been added to support this statement.

16. LVI, lymphovascular invasion is mentioned. Don't find it in materials & methods.

Lymphovascular invasion is found in the methods as part of the survival analysis section. The acronym (LVI) has been added to page #34 as well.

17. When the tumor groups are defined they are given numbers. However, when they are first introduced in figures (for example 2b & c, 5a, 7a) they are not in numerical order. It would maybe be much easier to follow if the clusters are renumbered in numerical order in the first figure where they appear.

The assignment of numbers of the clusters in figures 2b,2c, 5a and 7a is not random and were not manually chosen, but derived from the consensus clustering algorithm we used. The numbers assigned for each cluster are based on the consensus clustering algorithm output and determined in an unsupervised manner by the ConsensusClusterPlus function. If we were to manually change the numbers in figure 2b to be in a numerical order, we would need to force changing the figure itself to follow that order. This will consequently result in changing the numerical order of the clusters in figure 2c again and the reader would not be able to match the cluster names with the consensus matrices plots present in Supplementary figures S5, S10 and S12. This is described in the consensus clustering algorithm of the ConsensusClusterPlus package where it makes cluster number decisions based on the purity of members in the clusters {Wilkerson MD; ConsensusClusterPlus: a class discovery tool with confidence assessments and item tracking. *Bioinformatics* 2010}.

18. The identification and quantification of 4214 proteins across all samples is a good result for MS analysis of FFPE samples. But could some of the results be explained by not reaching deep enough into the proteome, considering that there should be around 14000 proteins in a tissue according to ProteinAtlas. Could this be a reason for the grouping of Luminal A tumors with Her2 in fig 2b? In the Krug et al *Cell* 2020 paper their tumor grouping almost exclusively only mix luminal A and Bs. No HER2 based on 7679 proteins quantified across all 122 samples. Can the lum A mixing with Her2 be reproduced with the same proteins? Or is this an effect of FFPE?

The composition of our 08-13 cohort is different from CPTAC as our cohort included only 11 luminal A cases compared to 73 basal-like, 62 Her2-Enriched, and 28 luminal B. The 4 clusters displayed in Fig 2b were the best to segregate this cohort by consensus clustering and thus with only 11 cases, luminal A tumors were not found as a unique cluster, but grouped with clusters 1 and 4 that included luminal B and Her2-Enriched in Fig 2b. In these clusters 1 and 4, luminal B and Her2-Enriched were often intermixed which is a commonly known phenomenon in breast cancer subtyping {Prat, A. et al. Molecular features and survival outcomes of the intrinsic subtypes within HER2-positive breast cancer. *JNCI* 2014} and is consistent with the proteomics breast cancer data in {Johansson et al. *Nat Comm* 2019}.

The cluster membership of our cohort compared to the CPTAC breast cancer cohort was dependent on a different combination of cases and in turn our analysis of the 86-92 with more luminal A cases was more powered showing distinctions of two subgroups within the luminal A subtype including a unique luminal A “stromal enriched” cluster, and a cluster that was more a mix of luminal A and B. Thus, overall our results are driven by the biology and the composition of our 08-13 cohort rather than an artifact or a technical limitation.

19. In fig S3a, you refer to biological replicates. How is biological replicates defined in clinical samples? For the technical replicates, it would have been better if they were spread out in different TMT sets.

The biological replicates refer to different specimens taken from the same patients. We acknowledge that technical replicates were in the same TMT set.

We have added the definition of biological replicates to the text on page #7-8:

“High reproducibility was observed between the biological replicates (referring to different specimens taken from the same patient) (mean $r=0.71$) and the technical replicates (mean $r=0.88$) (Supplementary Figs. S4a-S4b)”.

20. The PAM50 subtypes have got standard color code. See TCGA 2012 *Nature* or Krug et al *Cell* 2020. To avoid confusion I strongly recommended to use the same color code.

As per the reviewers’ request to make it easier for a reader to compare our results with recent breast cancer ‘omic studies we changed the colors to match the color code used in Johansson et al and Krug et al.

21. In general the authors make a good job in describing their findings. But to make it easier to follow I would suggest to add ER, PR and HER2 status to fig 2C. For example in the text it says: “Most cases in Cluster-2 and -3 were associated with ER, PR and Her2 negativity by IHC

clinical tests, high proliferation index (Ki67), and the “core basal” phenotype (defined as ER-, PR-, Her2- and [EGFR+ or CK5+])²⁹ (Supplementary Table 1).” Adding the clinicopathological markers to the heatmap in fig 2c would make it easy to see this in addition to the table. But this is a matter of taste and you can ignore if you like.

We thank the reviewer for this suggestion. Supplementary Table 1, Supplementary Table 2, Supplementary Data S1e and the “results” section describe and elaborate on the correlation between these clinicopathological variables and clusters. Figure 2c is already rich in information and different types of analysis and so we feel that the main emphasis for readers should be the PAM50 subtype membership in each proteome cluster.

22. In fig 2c, 5a, there is a column called immune with 2 categories, Immune related and Other. How are they defined? Also, for the protein groups there are enrichments, how were the enrichments done? Specify in fig text how the terms were selected, representative/ cutoff?

Immune related proteins were defined based on their protein function involvement in immune-response biological processes. Proteins belonging to any of these gene ontology (GO) categories were labeled as Immune:

```
"GO_DEFENSE_RESPONSE_TO_VIRUS", "GO_RESPONSE_TO_VIRUS",  
"GO_RESPONSE_TO_TYPE_I_INTERFERON",  
"GO_CELLULAR_RESPONSE_TO_INTERFERON_GAMMA",  
"GO_RESPONSE_TO_INTERFERON_GAMMA",  
"GO_REGULATION_OF_INNATE_IMMUNE_RESPONSE",  
"GO_CYTOKINE_MEDIATED_SIGNALING_PATHWAY",  
"GO_ANTIGEN_RECEPTOR_MEDIATED_SIGNALING_PATHWAY",  
"GO_IMMUNE_EFFECTOR_PROCESS",  
"GO_ACTIVATION_OF_INNATE_IMMUNE_RESPONSE",  
"GO_ANTIGEN_PROCESSING_AND_PRESENTATION_OF_PEPTIDE_ANTIGEN_VIA_MHC_CLASS_I",  
"GO_FC_EPSILON_RECEPTOR_SIGNALING_PATHWAY",  
"GO_POSITIVE_REGULATION_OF_INNATE_IMMUNE_RESPONSE"
```

For each protein cluster, the most representative terms were selected based on gprofiler enrichment analysis with the following parameters: organism = "hsapiens", ordered_query = FALSE, multi_query = FALSE, significant = TRUE, exclude_ia = TRUE, measure_underrepresentation = FALSE, evcodes = TRUE, user_threshold = 0.05, correction_method = "g_SCS", domain_scope = "annotated", custom_bg = NULL, numeric_ns = "", sources = NULL, term_size < 150 and source in GO:MF, GO:BP or REACTOME'

Raudvere, U., Kolberg, L., Kuzmin, I., Arak, T., Adler, P., Peterson, H., & Vilo, J. (2019). Reference: g:Profiler: a web server for functional enrichment analysis and conversions of gene lists (2019 update). *Nucleic Acids Research*, 47(W1), W191–W198. <https://doi.org/10.1093/nar/gkz369>.

The legends of figures 2c and 5a were updated to include this information.

“Immune related is defined based on the protein function as involved in immune-response biological process and for each protein cluster, the most representative terms displayed on the heatmap were selected based on g:profiler⁴ enrichment analysis”.

The methods section page #31 was updated to include information on the terms selected from the enrichment analysis.

For each protein cluster, the most representative terms were selected and presented on heatmaps based on g:profiler⁷⁷ enrichment analysis with the following parameters: organism = "hsapiens", ordered_query = FALSE, multi_query = FALSE, significant = TRUE, exclude_iea = TRUE, measure_underrepresentation = FALSE, evcodes = TRUE, user_threshold = 0.05, correction_method = "g_SCS", domain_scope = "annotated", custom_bg = NULL, numeric_ns = "", sources = NULL, term_size < 150 and source in GO:MF, GO:BP or REACTOME'.

23. Fig 2a, is this using all or the most varying proteins? In 2b it does not say that the grouping is based on consensus clustering.

UMAP in Fig 2a is based on using all proteins quantified in every sample (4214). The figure legend has been updated accordingly.

The legend of Fig. 2b has been updated to show that the grouping of the different clusters is based on consensus clustering.

(a) Uniform Manifold Approximation and Projection of the 08-13 cohort for the basal-like, luminal A, luminal B, and Her2-Enriched PAM50 subtypes based on all proteins quantified in every samples (4214).

(b) Alluvial plot shows the relationship between PAM50 subtypes and the four proteomic consensus clusters in the 08-13 cohort.

24. In figure S2b-c, the authors show number of peptides per protein. Bit unclear to what it refers to when mentioning peptide? Is that unique peptides? Nr of peptides per protein, is that the per set or total across all TMT sets or mean/median?

It refers to the total number of peptides identified per protein across all TMT sets. The legends for these figures have been updated accordingly.

(a) Percentage of the total number of proteins detected in different number of samples.

(b and c) Number and percentages of proteins identified according to total number of peptides per protein. Yellow bars in the histogram show the number of proteins identified by different numbers of peptides per protein. Blue dots show the percentage of total proteins identified per minimal number of peptides per protein.

25. The number of unique peptides per protein, nr of psm per protein and nr of psm/protein for TMT quantification is missing from the supplementary table with all MS data. Please add this, since it is important when it comes to judging the quantitative robustness. Having said that, must

give all the credits for clear clinical information and that the authors include it in the same document so it is easy to access!

We thank the reviewer for this point. As also requested in the reviewer's comment #1, we have added the total number of peptides for each protein, number of unique peptides per protein and number of PSMs used for quantification per protein to the Supplementary Data S1c.

26. In figure 3e, the y-axis says abundance. Is this log₂ ratio to the pool of samples? ESR1 is high in cluster 2 which is one of the basal enriched clusters, which is surprising. Could this be due to isodoping or poor quantification? KRT18 and FOXA1 on the other hand behave as expected.

Protein abundance shown is based on a log₂ ratio for PSM abundances divided by the relative PIS value in each TMT plex. Then for each protein, the median ratio of the 5 most abundant PSMs was used as relative abundance. This is explained in the methods section page #30 and has been added to the legend of Fig. 3e.

“Protein abundance values are based on log₂ ratio for PSMs abundances divided by the relative PIS value in each TMT plex. For each protein, the median ratio of the 5 most abundant PSMs was used as relative abundance”.

The abundance for ESR1 was significantly lower in Cluster-3 than the mean against “all” while ESR1 was non-significantly high in Cluster-2. This could be due to challenges in quantifying ESR1 as endogenous peptides for this protein were only detected in less than 10% of the samples. Using isoDoping, 3 isoDoping peptides for ESR1 were detected in the majority of samples and thus challenges in ESR1 quantification might explain the non-significantly higher levels observed for Cluster-2.

27. PECA is used for calculating p-values. Wonder if that inflates the p-values and makes them smaller just because you have a lot of peptides per protein?

<https://pubs.acs.org/doi/10.1021/acs.jproteome.5b00363>

PECA method leverages the number of peptides per protein to assign higher confidence to proteins with higher peptide coverage. While we agree that this method tends to drive the p-value of certain proteins with a particularly high number of peptides, we find it useful to separate proteins with a small number of peptides since these are the ones with lower confidence in quantification levels. We directly compared PECA performance to another differential expression algorithm (DEqMS, Zhu, Y., Orre, L. M., Zhou Tran, Y., Mermelekas, G., Johansson, H. J., Malyutina, A., Anders, S., & Lehtiö, J. (2020). DEqMS: A Method for Accurate Variance Estimation in Differential Protein Expression Analysis. *Molecular & Cellular Proteomics*, 19(6), 1047–1057. <https://doi.org/10.1074/mcp.tir119.001646>) on the first differential expression contrast (Cluster1 vs Cluster2-3-4). We found that the two methods give comparable results in terms of calling differentially expressed (DE) proteins (adjusted p-value < 0.05). We found an overall agreement by DE status on 86% of the proteins: 6% of the proteins differentially expressed in PECA and not in DEqMS, 9% of proteins differentially expressed in DEqMS and

not PECA, 11% consistently identified as DE in both methods, and 75% consistently identified as not differentially expressed.

While several differential expression analysis methods are routinely used in the proteomics field and their evaluation over multiple types of data and experiments would be of great interest, we believe that a technical evaluation of PECA and/or comparison with other methods are beyond the scope of this paper.

28. Full credit for uploading the immunohistochemistry slides to <http://www.gpec.ubc.ca/prot>. But why limit to representative images. For the dataset to be useful, all images needs to be available. In addition, there should be an easy way to download all data for image analysis.

Our IT team at the Genetic Pathology Evaluation Centre has diligently uploaded all images to <http://www.gpec.ubc.ca/prot>. This information has been updated under the section of “Data availability”, page #34, in the methods.

“Images from immunohistochemistry slides of tissue microarrays used in the study coded as “11-012” and “14-004” are available for public access via the website of Genetic Pathology Evaluation Center (<http://www.gpec.ubc.ca/prot>).

Data image analysis and clinical outcome data for the cases used in this study can be made available through the Genetic Pathology Evaluation Centre and Breast Cancer Outcomes Unit of BC Cancer Centre, upon completion of a Data Transfer Agreement and confirmation of ethical approval for qualified researchers”.

29. To make the data analysis part transparent and reproducible, analysis code should be uploaded to Github or similar repository.

“Code Availability” section has been added to the methods after the “Data Availability” section as requested. Code used for proteomics data analysis is available at GitHub <https://github.com/glnegri/brca>.

30. Orbitrap MS2 data was matched with 0.5 Daltons tolerance. This is a very large window that is usually used for iontrap data. For orbitrap MS2, the tolerance should be around 0.02 Dalton to reduce the risk of miss assigning transitions. Since you are also using methylation of lysine as a variable modification, this in combination with a large tolerance will increase your FDR. You should research at least parts of the data and compare the results to your present results to determine if all data needs to be researched. In relation to this, what is the protein FDR of the dataset, q-value, pep value for each protein?

We thank the reviewer for this point. This was actually a typographical error in the text; the data were in fact searched with 0.05 Da tolerance. We have updated the “methods” page #29 accordingly. The full parameters used for the Proteome Discoverer search, together with the results output are available at the PRIDE repository with the dataset identifier PXD024322.

31. From results: FFPE samples were macro-dissected from 3-6 sections to obtain >80% tumor content and analyzed using the SP3-CTP multiplex MS proteomics protocol²⁴ (Supplementary

Fig. S1b).

Should it not be ref 19 instead of 24?

Reference #24 {Hughes, C.S., *et al.* Single-pot, solid-phase-enhanced sample preparation for proteomics experiments. *Nat Protoc* **14**, 68-85 (2019)} is a more detailed and up-to-date protocol for the methods used in this study when compared to Reference #19 {Hughes, C.S., *et al.* Quantitative Profiling of Single Formalin Fixed Tumour Sections: proteomics for translational research. *Sci Rep* **6**, 34949 (2016)}. Given that Reference #19 included work done on FFPE (in ovarian cancer) we now include both references #19 and #24 to support our statement.

Reviewer #2, expert in bioinformatics and subtype classification (Remarks to the Author):

In this study, the authors carry out mass-spec proteomic profiling of 300 FFPE breast cancer surgical specimens. The specimens are separated into two cohorts based on batch effects. The 08-13 cohort included 75 basal-like, 62 Her2-Enriched, 30 luminal B, and 11 luminal A PAM50 defined cases. The 86-92 cohort provided the long-term outcome data required for luminal cases and included 64 luminal A, 45 luminal B, and 13 Her2-Enriched PAM50 cases. The 08-13 cohort was used for subtype discovery, both across all tumors and within the TNBC subset. ER+ subtypes examined in the 08-13 cohort were examined in the 86-92 cohort.

Specific comments:

1. Batch effects were found between the 08-13 and 86-92 cohorts, likely due to differences in collection techniques, pre-analytical handling, and fixation procedures. Could the authors try to harmonize the two datasets using Combat (<https://rdrr.io/bioc/sva/man/ComBat.html>)? In practice, Combat is very good at removing batch effect differences. Data from different platforms (RPPA, RNA-seq, DNA methylation) have been successfully processed with Combat, and the method is independent of nature of the batch effect. The PAM50 subtype could be used as the experimental group. There would be advantages in having one harmonized dataset of 300 samples. It seems worth a try. As currently written, the Abstract suggests that there is one dataset that was analyzed, rather than two separate cohorts.

As has been shown before, ComBat can lead to overestimating ratios and many in the field believe should be avoided. (Methods that remove batch effects while retaining group differences may lead to exaggerated confidence in downstream analyses <https://academic.oup.com/biostatistics/article/17/1/29/1744261>), especially considering that the batch effect observed in our study is mostly driven by missed identification of peptides cleaved at lysines and not by artifacts on quantification, as shown in figures S3c and S3d. Furthermore, some of the subtypes are completely (basal-like) or almost completely (luminal A) confounded with the ‘cohort’ batch effect. While Combat will always transform the data to minimize batch differences, we believe that for the reasons above, its application in this dataset would lead to serious artifacts in the data.

We would also like to note that the decision to include cases from the 86-92 cohort in our study design was based on clinical and translational considerations. In order for analysis to be meaningful for luminal cases, a long enough follow-up was necessary to obtain sufficient events for outcome analyses. Thus, the majority of luminal PAM50 cases were derived from patients diagnosed with invasive breast cancer in the period January 1986 to September 1992. Forcing the

two cohorts to be lumped together for subtyping does not allow obtaining clinically-relevant results for the subtypes found, and could compromise any clinical relevant observations.

We have updated the abstract to highlight that for the 300 cases included there were 2 datasets analyzed rather than one.

“We performed comprehensive proteomic profiling of 300 FFPE breast cancer surgical specimens, 75 of each PAM50 subtype, from patients diagnosed in 2008-2013 (n=178) and 1986-1992 (n=122) with linked clinical outcomes”.

2. Page 8: "Cluster-1 (n=34) consisted mostly of luminal B and Her2-Enriched PAM50 cases. Clusters-2 (n=50) was enriched for basal-like subtype, included few Her2-Enriched, but had no luminal cases. Cluster-3 (n=47) was primarily basal-like cases but included Her2-Enriched cases. Cluster-4 (n=43) was mostly Her2-Enriched but included luminal A and luminal B cases." It seems that actual numbers to reflect the noted associations would be helpful here, e.g. exactly how many basal-like cases and Her2 cases were in Cluster-3, and was Cluster-2 SIGNIFICANTLY enriched for basal-like.

Cluster-2 is enriched for basal-like (pval<1.16e-11, Fisher's test), Cluster-3 is enriched for basal-like (pval<1.3e-4, Fisher's test), Cluster-4 is enriched for Her2-Enriched (pval<1.9e-4, Fisher's test).

The numbers reflecting the breakdown for each PAM50 subtype within each proteome cluster as they appear in Fig. 2b have also been added to the text, page #9.

“Cluster-1 (n=34) consisted mostly of luminal B (n=18) and Her2-Enriched (n=13) PAM50 cases. Clusters-2 (n=50) was significantly enriched for basal-like subtype (n=41), included few Her2-Enriched, but had no luminal cases (p-value<1.16e-11, Fisher's test). Cluster-3 (n=47) was primarily basal-like cases (n=31) but included Her2-Enriched cases (n=14) (p-value<1.3e-4, Fisher's test). Cluster-4 (n=43) was mostly Her2-Enriched (n=26) but included luminal A (n=8) and luminal B (n=8) cases (p-value<1.9e-4, Fisher's test)”.

3. In general, where the word "significantly" appears in the main text, it would be good to include a p-value and associated test to support the claim. The figures referred to likely include the test, but reflecting this in the main text as well would be helpful to the reader. For example, page 11: "The immune hot cluster also had significantly higher CD8+ TILs in the intratumoral compartment compared to other clusters (Fig. 4a)." by what p-value and test?

The p-values and tests are now updated across the text where the word “significantly” appears.

4. Wherever a p-value appears in the main text, the test used to derive that p-value should also be indicated. For example, page 12: "The subgroups with a high expression for only one of these biomarkers were characterized with intermediate RFS (Supplementary Fig. S5b). 70% (21/30) of the cases classified as (TAP1 high/HLA-DQA1 high) were in Cluster-3, while 90% (76/84) of (TAP1 low/HLA-DQA1 low) cases were in other clusters (p-value<0.00001) (Supplementary Table 1)." What test was used here (we can save the reader from having to go the Table for the answer)?

The test used was the Chi-square test. The text in page #13 has been updated to include this information.

“70% (21/30) of the cases classified as (TAP1 high/HLA-DQA1 high) were in Cluster-3, while 90% (76/84) of (TAP1 low/HLA-DQA1 low) cases were in other clusters (Chi-square p-value<0.00001) (Supplementary Table 1)”.

5. Page 15: "Multiple correction testing identified fatty acid-binding protein-7 (FABP7) as a candidate biomarker most significantly associated with >10-year RFS on tamoxifen treatment..." Was this the only protein that was significant? Were other proteins significant and using what statistical test and cutoff?

The association between the continuous increase in each individual protein identified in the cohort 86-92 and the endpoint of 10-years RFS was tested using a Cox regression model and stratified log-rank test. This analysis is displayed in Supplementary Data S4f. Only protein biomarkers that had a significant log-rank p-value <0.05 when adjusted for multiplicity testing by the Benjamini-Hochberg test were selected. **Only** FABP7 protein was found to meet these criteria as displayed in Supplementary Data S4f.

The relevant text for the 86-92 analysis page #18 has been updated to include this information. *“Multiple correction testing identified fatty acid-binding protein-7 (FABP7) as the only candidate biomarker associated with >10-year RFS on tamoxifen treatment (log-rank BHadj p=0.00004) (Supplementary Data S4f, Supplementary Fig. S12e)”.*

6. Discussion, page 16. Many journals are uncomfortable with the phrase "(manuscript in preparation)." It seems that the method indicated should be described in sufficient detail in the Methods, if it isn't already.

We believe that the methods regarding the isoDoping methodology are now described in sufficient detail in the methods section of this manuscript for the reader to be able to reproduce the experiment as was intended. While we are currently preparing an even more detailed and comprehensive description of the general isoDoping strategy for a separate primary methodology-oriented publication, to avoid confusion we have deleted the mention of a "manuscript in preparation." From pages #6 and #18.

7. In addition to making the raw data available on ProteomeXchange, it would be most helpful to include the processed proteinXsample tables as Supplementary Data with the published paper. CPTAC has done a similar thing with their past publications.

The proteinXsample data are included in the original Supplementary Data S1c. As requested by reviewer #1, we have also added the peptides identified across the cohort to the Supplementary data S1 along with the total number of unique peptides per protein and number of PSMs used in quantification per protein (Supplementary Data S1c-S1d).

8. For boxplots in the figures, please define the ranges involved.

Boxplot whiskers range extends to the most extreme data point which is no more than 1.5 times the interquartile range from the box. This definition has been added to the legends of Fig. 3e and

Fig. 7b.

Reviewer #3, expert in breast cancer subtypes (Remarks to the Author):

1. The authors present their previously described highly sensitive MS-based methodology termed “Single-Pot, Solid-Phase enhanced, Sample Preparation”-Clinical Tissue Proteomics (SP3-CTP). This technology has been shown to capture known and novel features in FFPE tumor samples. The authors have previously shown that this method can be applied on large FFPE material cohorts linked to outcome data. Comprehensive quantification of protein expression can be achieved even from lower input quantities of patient specimens such as small biopsies. Here is would have been useful to know how small?

This is described in the methods section and supplementary Figure S1a. One to six unstained 10 μ m tissue sections were cut for each sample to obtain an aggregate total area of \sim 1cm x 1cm x 10 μ m, with >80% tumor content.

2. In this paper they have applied the method to 300 well-characterized archival FFPE breast cancer specimens in terms of clinical outcome, IHC, and PAM50 RNA-based intrinsic subtypes. The authors demonstrate that at the protein level one can identify groups characterized by high expression of immune-response proteins and favorable clinical outcomes. Does this paper bring a sufficient novelty? While it is true that “classifications do not always guide therapeutic choices, due to the extensive heterogeneity that still characterizes breast cancers” can this be solved by adding one more, at the level of proteomics?

As described in the introduction, we performed the current study because genomic classifications of breast cancer are inherently limited as clinical decisions are generally based on the protein level. The underlying technology’s application to FFPE breast cancer material is novel. To the extent that some of the findings overlap with genomic classifications, our study still provides an important verification at the protein level, where most drugs act.

Q1. How do this extension to 300 cases add to what we know from Johansson et al *Nat Comm*, 2019?

As highlighted in the introduction, Johansson et al. *Nat Comm* 2019 only profiled 9 tumor samples from each of the four main breast cancer PAM50 subtypes, a set which also lacked clinical outcome associations and was insufficient to characterize the biological heterogeneity of breast cancers in relation to clinical behavior and treatment response. In addition, their work required fresh-frozen tissues that are not routinely available from patients, unlike the FFPE clinical specimens we were able to use that can be accessed in larger numbers allowing meaningfully powered linkages to clinical outcomes.

Q2. How does the heterogeneity described here match what is known from RNA based classification (basal also divided in several immune clusters)

The PAM50 subtypes used in this study are an RNA-based classification and the associations of each proteome cluster membership with each PAM50 subtype are described in detail in the

manuscript. Within the basal-like RNA-based subtype, there are two distinct proteomic groups that differ in immune response. In the results section, we describe how the heterogeneity of triple negative breast cancer relates to what is known from RNA-based classifications by comparing our findings with those by Burstein M et al. *CCR* 2015, showing that our triple negative clusters were highly correlated with their corresponding RNA subtypes of ‘luminal-androgen receptor’, ‘mesenchymal’, ‘basal-immune suppressed’ and ‘basal-immune activated’.

Q3. If the authors were to make biomarkers based on protein as they suggest, which ones would they chose?

TAP1 and HLA-DQA1, as described in detail in the results and discussion sections. These choices are further supported by the supplementary validation work done in response to reviewer #1, comment #8 as described above (based on the data shown in Supplementary Figures S7 and the new figure S8). We do note that TAP1 and HLA-DQA1 were chosen, in part, because of the availability of quality IHC grade antibodies; it remains possible that other proteins may perform better on IHC-based tests when quality antibodies are available. Indeed, this is one of the prime utilities of our results for the breast cancer community, to spur additional biomarker research using our data.

The discussion page #21 has been updated with this information.

“Other proteins elevated in the immune hot cluster with available quality antibodies could also be used and developed as candidate biomarkers”.

Q4. An introduction of 5 pages and large number of references (81) makes it into a difficult read. This paper as rigorously performed and described, would benefit from some clarity and simplification, just highlighting the results that move the field forward.

The original work was written in a way that fits the requirements of *Nature Communications*. The introduction here is 2.5 pages double spaced rather than 5 pages as pointed out by the reviewer and the authors hold that this is adequate to succinctly review the pertinent literature, making it hard to remove any essential information from the introduction. As this research sits at a crossroads of breast cancer, bioinformatics, and analytical chemistry the authors believe it is important to provide key background information for scientists from a breadth of related and interested fields to fully appreciate the work. 84 references are merely supporting information for the interested reader to pursue, a number that complies with the *Nature Communications* guidelines (and we are aware of several detailed and comprehensive publications in *Nature Communications* that have a similar or even higher number of references used to properly cover the scientific data presented).

Reviewers' Comments:

Reviewer #1:
See attached

RESPONSE TO REVIEWER COMMENTS

Reviewer #1, expert in proteomics (Remarks to the Author):

Asleh et al have performed quantitative proteomics of 300 breast tumors from formalin fixed paraffin embedded (FFPE) material. The general idea and its potential value to the community of this work is great.

The main Merits of the paper are: 1) the acquisition of proteomics data from FFPE samples across a large number of breast cancer samples with clinical follow up that can serve as a resource, 2) directly linking protein based sample groups with immune infiltration to improved outcome, 3) suggestion of potential biomarkers for tumor groups 4) identification of 4 TNBC groups as previously suggested at the RNA level and linking the immune infiltrated subgroup to good outcome, 5) identification of 3 ER positive tumor groups with a stromal enriched group.

We appreciate the reviewer's view that the work will provide important value to both the fields of breast cancer and of proteomic analysis of patient samples in general.

Limitations regarding merits above:

1. To function as a resource the data needs to be judged as robust.

To evaluate protein quantitative robustness, the number of peptides used for quantification per protein needs to be available and visualized. Now it is lacking from the supplementary data table with all ratios. A panel can also be added to figure 2 to show nr of psms/protein used for quantification.

We thank the reviewer for this point. We have added the total number of peptides for each protein, number of unique peptides per protein and number of PSMs used in quantification per protein to the Supplementary Data S1c. We have also added the data on the peptide abundance per protein (now appears as new Supplementary Data S1d) and PSMs per protein in Supplementary Figure S2d.

Good! However, you need to fix the x-axis. Now it reads: **Average number of PSMs x TMT plex**

Fig text: Average number of quantified PSMs per protein, across the full cohort – is that for the subset with quantification across all or including all proteins?

In suppl data S1C the column header says: set_1_number_PSMs – that is nr of psms used for quantification I presume? When you add this information, it would also be informative to add the nr of unique peptides/protein per set. Also, protein scores and q-values are missing from the table. Add a column to easily select the proteins that you have used in your data analysis.

Supplementary Figure S2

2. The supermix is present in all TMT sets and should represent how well quantifications can be reproduced between TMT sets. Can the supermix data be used for robustness evaluation between the sets? For example a heatmap for overview, variation of supermix in relation to the breast samples and particular sets with deviation on supermix-sample.

We thank the reviewer for this point. The 38 SuperMix replicates included in our experiment showed a high correlation across the 38 plexes. Unsupervised clustering of our data for all samples including breast tumors, normals, and SuperMix show the SuperMix samples clustered together and are clearly separated from the breast tumor and normal samples. The correlation between the SuperMix samples was the highest when compared to the breast tumor and normal samples, supporting the robustness of the evaluation of SuperMix samples across the sets (appears as new Supplementary Fig. S4c). Pairwise correlation between the 38 SuperMix replicates (ranged between 0.68-0.81, median 0.75) was significantly higher than the pairwise correlation across the 38 normals (ranged between 0.53- 0.85, median 0.71). These findings are shown in new Supplementary Fig. S4d.

The following information has been added to the results section page #8:

“An overview clustering of all the samples included in our study showed that the 38 SuperMix replicates had the highest correlation across the 38 plexes (range 0.68-0.81) when compared to the breast tumors and normal samples, supporting the robust quantification of SuperMix samples across the different sets (Supplementary Figs. S4c-S4d).”

The small difference in correlation between the Supermix, that should be exactly the same sample in all TMT sets, and the normal samples, which are biologically different are surprising. The supermix should represent technical variation and in this case are very close to the biological variation. The large number of proteins used in the sample to sample correlation analysis will provide a relatively high correlation, which limits this analysis.

To be able to support the claim of the dataset as resource, the reader needs to be able to better understand the technical variation in the dataset. For example, you could calculate coefficient of variation for each protein based on the supermix and plot that. Also, you have IHC data for some proteins as ESR1, PGR etc, how these measurements correlate to the proteome data would be useful for judging the quality of the data.

Supplementary Figure S4

3. An overview clustering of the 2 cohorts with replicates would also be useful to judge how the whole dataset behaves. Does the technical replicates cluster together?

Per the reviewer's request, we generated a heatmap showing the overview clustering for all the samples, as also requested in the previous comment. As now shown in Supplementary Fig. S4c, the three technical replicates indeed have clustered adjacent to each other (T_rep 5, T_rep 6, T_rep 7). Regarding the biological replicates, 2 of 3 replicates clustered adjacent to each other while the 3rd biological replicates clustered very closely together, a variance in line with expectations for intratumoral regional sampling. The normal samples clearly separated from tumor samples and showed an overall correlation of 0.70. An overall correlation of 0.5-0.6 was observed for the different breast tumor clusters and these included a mix of samples from both 08-13 and 86-92 cohorts.

This information has been added to the results section page #8:

"All the technical replicates and 2 out of 3 biological replicates clustered adjacent to each other, while the 3rd biological replicates clustered very closely together, a variance in line with expectations for intra-tumoral regional sampling (Supplementary Fig. S4c).

Ok

4. The data should also confirm with previous knowledge, as ER, PR, HER2, MKI67 levels in different PAM50 subtypes, and this would be good to show in a supplementary figure.

HER2 (ERBB2) and MKI67 expression levels across the different PAM50 subtypes are found in Supplementary Fig. S6d. ER (ESR1) and PR (PGR) expression levels across the different PAM50 subtypes are now also included in Supplementary Fig. S6d.

Ok, see my comment to question 2.

5. Proteomics have previously identified immune infiltration in breast cancer subgroups without directly linking them to outcome (Krug 2020 *Cell*, Johansson et al 2019 *Nat Comm*). Tumor-infiltrating lymphocytes (TILs) have also been linked to better outcome in breast cancer subtypes (Dieci 2021 *Cells*). The strength of this study is the direct link between proteomics data with "immune hot" tumors and outcome.

Relation to published data

In general, anchoring the novel findings further, e.g. by validation of findings in other breast proteomics data sets would be valuable to show the usefulness of the data as a resource and strengthen the findings. There is several decent datasets published now on breast cancer proteome so this should be done.

Per the reviewer's request, we performed a validation of our findings on previous proteomic datasets published by Krug et al. *Cell* 2020 (CPTAC) and Johansson et al. *Nat. Commun* 2019 (OSLO2).

Validation using the Krug et al. 2020 CPTAC breast tumor cohort: In order to compare our results with available published datasets, we performed consensus clustering with the same parameters used in our cohort on the CPTAC *Cell* 2020 cohort, using 939 proteins from the CPTAC data that overlap with the 1054 mostly highly-variant proteins of the 08-13 cohort. This

analysis identified four main proteome clusters that highly resembled the original CPTAC NMF clusters of “LumA-I”, “LumB-I”, “Basal-I”, “HER2-I”. Two of these were almost entirely similar to the original NMF clusters of “Basal-I”, and “LumA-I”. Another cluster highly resembled NMF “LumB-I” and consistent with Krug et al consisted of 54% luminal A cases (compared to 55% luminal A cases assigned as “LumB-I” in the original NMF CPTAC clusters by Krug et al). Similar to the original NMF CPTAC clustering composition, the NMF CPTAC “HER2-I” cluster identified had a mix of Her2-Enriched, luminal A and luminal B breast cancers. Of note, the original Krug et al 2020 study of 122 breast tumors included a majority of luminal A PAM50 subtype (n=57, 47%), followed by basal-like (n=29, 24%), luminal B (n=17, 14%) and Her2-Enriched (n=13, 11%) when compared to the composition of our 08-13 cohort consisting of a higher number of basal-like (n=73, 42%) and Her2-Enriched (n=62, 36%) cases, but few luminal A (n=11, 6%). Despite this, our analysis further reproduced the existence of subsets enriched for immune response pathways at the proteome level within the basal-like and Her2-Enriched subtypes not captured by CPTAC analysis. Consistent with our analysis on the 08-13 cohort, stromal pathways were enriched in luminal A tumors and lipid metabolism was enriched within luminal B and Her2-Enriched tumors. A description of these findings is displayed in Supplementary Fig. S9a.

In the results section you write: Our analysis reproduced the existence of subsets enriched for immune response pathways at the.... These subsets are within your clusters. They don’t come out as defined clusters. You need to make that clear. It looks though as it should be possible to separate out immune enriched samples.

Validation using the Johansson et al 2019 “OSLO2 breast cancer landscape cohort”:

Validating our findings on the 36 cases of the 4 main subtypes (9 for each PAM50 type) on “OSLO2 landscape cohort”, we performed consensus clustering with the same parameters used in our analysis, using 775 proteins from the OSLO2 data that overlap with the 1054 mostly highly-variant proteins of the 08-13 cohort. This analysis identified 4 clusters that highly resembled the main consensus core tumor clusters (CoTCs) and their biological functions as reported in Johansson et al. These clusters consisted of CoTC1 (basal-like immune cold), CoTC2 (basal-like immune hot), CoTC3 with few CoTC6 cases (luminal A-enriched) and CoTC6 (luminal B and Her2-Enriched). Importantly, the immune distinctions within the basal-like subtype were entirely reproduced using our highly variant proteins showing that the two basal-like samples of OSL.3EB and OSL.449 (CoTC2) were consistently classified as “basal immune hot cluster” when compared to other basal cases characterized as “basal immune cold”. These findings are displayed in Supplementary Fig. S9b.

The results section page #14 has been updated to include our comparison analysis using the Krug et al 2020 and Johansson et al 2019 proteomics datasets, as a new section entitled “**Comparison with previous breast cancer proteomics studies**”.

The number of immune hot samples are a little bite low, but in the other hand supports your findings.

Supplementary Figure S9

6. The 4 TNBC groups are correlating to their suggested RNA based groups. To strengthen the finding of 4 TNBC subtypes, can they be identified also at the protein level, for example in Krug 2020 *Cell* data?

We validated our TNBC proteome clusters using the 935 proteins that overlap with the 1055 mostly highly-variant proteins in our 08-13 TNBC (n=88) subset on a set of 28 TNBC cases included in the CPTAC breast cancer cohort by Krug et al. Our analysis reproduced the existence of the four main proteome TNBC subgroups and the biological features of ‘luminal-androgen receptor’, ‘mesenchymal’, ‘basal-immune suppressed’, and ‘basal-immune activated’ as now shown in Supplementary Fig. S11.

The results section page #16 has been updated to include this information:

“The existence of these TNBC proteome clusters and their biological features were validated when applying consensus clustering, with identical parameters, on 935 proteins overlapping with the 1055 mostly highly-variant proteins of the 08-13 TNBC subset on the proteomic data for a set of 28 TNBC cases included in the CPTAC breast cancer cohort by Krug et al’ (Supplementary Fig. S11).

Supplementary Figure S11

Ok, good!

7. How generalizable are the 3 ER positive tumor subgroups identified in the manuscript? The authors cite Krug 2020 *Cell* in the discussion as consistent with the stromal-enriched subtype. But to my knowledge, the data in the Krug paper don't show a separate luminal A subgroup enriched for stroma. Dennison 2016 *CCR*, however show a stromal subtype of ER positive tumors that are or mixed subtype but enriched in Luminal A with a favorable clinical outcome. Are the same proteins (in RPPA and your MS data) deterministic of the stromal subgroup?

We agree with the reviewer that Krug 2020 *Cell* did not identify a separate stromal enriched subtype as a unique cluster by mass spectrometry, but described a subset of luminal A tumors as stromal-enriched since these tumors were classified originally in TCGA 2012 based on RPPA data as "reactive". In the subsequent *Nature* 2016 CPTAC proteomics profiling breast cancer publication, the proteomic cluster that was highly correlated with the "reactive" RPPA cluster was referred to as stromal-enriched.

The Dennison 2016 *CCR* basically tried to characterize the biological and clinical features of the stromal enriched tumors as a whole (i.e. reactive tumors) identified in the TCGA based on RPPA data. The majority of these tumors were found to be classified as luminal A by PAM50, and among the luminal A as a group those that had high stromal protein expression displayed favorable clinical outcomes.

Comparing the proteins in our MS data that are in common with the RPPA proteins (n=30) used to classify the “stromal-enriched” vs. the “ER positive cancer derived” subtypes in Dennison 2016 *CCR*, we found 5 proteins in the RPPA “stromal-enriched” Dennison 2016 *CCR* that were also characteristic for our luminal A stromal enriched proteomics cluster ($\log_2FC > 0.20$, adjusted p-value < 0.05). These were fibronectin, annexin, collagen VI, caveolin, and MYH11.

While our data correlate with those results, the RPPA data only cover a small percentage of the proteome that was quantified in our experiment; thus, our data characterize the luminal A stromal enriched cluster in a more comprehensive manner and identify protein candidates that are beyond those captured by the restricted number of proteins in the antibody-based RPPA assay.

The discussion page #22 has been updated to highlight this information.

“Our analysis of ER+ cases with mature clinical data identified a stromal-enriched subset (86-92-Cluster-2) consistent with previous reports^{56,62}, which could help sub-classify luminal breast cancer. However, our data characterize the luminal A stromal enriched cluster in a more comprehensive manner and identify protein candidates that are beyond those captured by the restricted number of proteins in the antibody-based RPPA assay”.

OK

8. IHC validation of S100A8, TAP1, IFIT2, HLA-DQA1 and CD8 as suggested biomarkers of immune infiltration and better outcome are done on the same cohort as the proteomics. To consolidate the findings, validation in an independent cohort would be valuable. Also, what is correlation between the MS data and the IHC validated markers? Are the MS protein levels also related to outcome?

First part of the reviewer’s comment: Per the reviewer’s request, we have now performed a validation of these IHC biomarkers on an independent set of 176 breast cancer cases with similar clinicopathological characteristics to the 08-13 cohort. Our analysis confirmed that high expression of HLA-DQA1 as a single biomarker had a significantly better survival (log-rank $p=0.02$) and a similar trend was seen with high TAP1 as a single biomarker (log-rank $p=0.09$). The findings further confirmed that tumors with IHC expression for both TAP1 and HLA-DQA1 showed the most favorable survival, while the subgroup with low expression for both had the worst RFS (log-rank $p=0.05$) (Supplementary Fig. S8).

Supplementary Figure S8

The results section page #13 has been updated with this information:

“We subsequently confirmed our observations on an independent, clinically similar set of 176 breast cancer cases and showed that high expression of HLA-DQA1 as a single biomarker had a significantly better survival (log-rank p=0.02) and a trend was seen for high TAP1 as a single biomarker (log-rank p=0.09). These data also confirmed that tumors with high IHC

expression for both TAP1 and HLA-DQA1 showed the most favorable survival, while the subgroup with low expression for both had the worst RFS (log-rank $p=0.05$) (Supplementary Table 3; Supplementary Fig. S8).

The Supplementary methods in the Supplementary Information file page #25 and Supplementary Table 3 include information on the characteristics of this IHC validation cohort:

“IHC validation cohort: A tissue microarray for an independent set of 176 breast cancer cases was used to validate observations on the 08-13 cohort for the key protein IHC biomarkers. This validation cohort had clinicopathological characteristics similar to the 08-13 cohort and was analyzed for IHC biomarker association with clinical outcomes. The median follow-up for the IHC validation cohort was 10 years and cases were treated in accordance with contemporary guidelines”. Characteristics of this cohort appear in the new updated Supplementary Table 3.

Supplementary Table 3

Characteristic	IHC Validation cohort (n=176)
Age at diagnosis (median)	53 years
Tumor size (median)	2 cm
Tumor grade	
1, 2	44 (25%)
3	127 (72%)
Missing	5 (3%)
Nodal status	
Negative	105 (60%)
Positive	66 (37%)
Missing	5 (3%)
IHC subtype	
Luminal ([ER+ or PR+])	69 (39%)
ER-, PR-, HER2+	32 (18%)
ER-, PR-, HER2-	71 (40%)
Missing	4 (3%)
Disease specific death	
No	134 (76%)
Yes	35 (20%)
Missing	7 (4%)
CD8 iTILs	
<1%	42 (24%)
≥1%	129 (73%)
Missing	5 (3%)
TAP1/HLA-DQA1 IHC groups	
TAP1 high /HLA-DQA1 high	35 (20%)
TAP1 low /HLA-DQA1 high	22 (13%)
TAP1 high /HLA-DQA1 low	50 (28%)
TAP1 low /HLA-DQA1 low	65 (37%)
Missing	4 (2%)

Good!

Second part of the reviewer’s comment: The Spearman correlation between the MS data and the H score for the IHC validated markers was found to be 0.51 for TAP1 and S100A8, 0.31 for HLA-DQA1, and 0.11 for IFIT2 as shown in the figure below. Of note, the assessment of the validated markers by IHC was performed on the carcinoma cells.

This data should also be in the paper together with the same kind of analysis for ESR1 and PGR. Why do you think the correlations are weak? For TAP1 and HLA-DQA1 that performs well together, what is the difference in signal that is picked up by IHC and MS? Both are prognostic but show weak correlations indicating different signal/information that they pick up.

Third part of the reviewer’s comment: The selection of the biomarkers for IHC validation was based on biology rather than clinical outcomes. In response to the reviewer’s comment, we performed a Cox proportional-hazards analysis on the protein abundance (in MS data) and recurrence free survival for the protein candidates we assessed by IHC. MS protein levels are significantly correlated with improved outcome for TAP1 and IFIT2, while a trend is shown for HLA-DQA1 and S100A8 as follows:

Protein	Survival analysis for RFS HR (95% CI), P-value	Adjusted P-value
TAP1	0.34 (0.18-0.65), 0.001	0.04
HLA-DQA1	0.87 (0.69-1.10), 0.24	0.71
S100A8	0.87 (0.73-1.06), 0.16	0.62
IFIT2	0.38 (0.18-0.80), 0.01	0.19

Additional comments:

9. From introduction: “This method can query large FFPE material cohorts linked to outcome data, enabling comprehensive quantification of protein expression from lower input quantities of routinely-available patient specimens, and employs a more highly efficient workflow than other MS-based methods for protein profiling of clinical FFPE tissues^{21,22}. “

Based on the data, the MS workflow seems efficient, but there is really no data to comparing all other methods to support your claim of “more highly efficient workflow than other MS-based

methods...”? Many of the large MS proteomics groups have published their versions of FFPE sample preparation methods. See for example Coscia 2020 Modern Pathology, Griesser 2020 MCP, Marchione 2020 JPR, Zhu 2019 Molecular Oncology.

We thank the reviewer for this comment. We have updated this sentence in the introduction page #5 accordingly:

“This method can be used to query large FFPE material cohorts linked to outcome data, enabling comprehensive quantification of protein expression from lower input quantities of routinely-available patient specimens, and employings a ~~more~~ highly efficient workflow than other MS-based methods for protein profiling of clinical FFPE tissues^{21,22}.

Ok

10. In the abstract and in figure one, 300 samples are mentioned as included in the study. The number is correct but it is bit misleading since it's divided up in 2 cohorts. The overview in Figure 1A is not useful since this collection of samples are not used together later on in the paper. The overview presented in fig S1A are much more useful since it gives an overview of the samples used together in each of the later analyses. Also the number of samples drop after QC and removal of replicates. To make it clearer for the reader I suggest you make a combination figure of fig S1A and S2H with the tumor characteristics and the numbers that make up each cohort used in the downstream analysis. Also include the info of how the TNBC cohort was made. This took time to figure out and with a figure outlining the 2 cohorts, it would be much clearer from the beginning for the readability of the entire paper. To make it even clearer one could add what type of analysis / aim you have with each cohort. There is also normal samples for which it is unclear of their purpose/how they are used. Did not find any comparison to the normal samples in the text?

Per the reviewer's recommendation we have moved the original Figures S1A and S2H to Figure 1. Now they appear as Fig. 1b and Fig. 1c.

Given that normals were sourced from independent reduction mammoplasties, they are very biologically different from tumors and thus they are not helpful in the subtyping or performing direct comparisons with tumor samples. The normals were included in the UMAP plots where they form a clearly separated cluster from tumors, added to the heatmaps (Figures 2c and 5a) as a reference to illustrate that proteins and pathways of interest for the proteome clusters were not high in normals, and as a visual comparator for the expression of key breast cancer associated proteins in Supplementary Figure S6d.

In addition, when we picked specific proteins of interest for validation in IHC, we used candidates that were not highly expressed in normals. We updated the text to include this specific information in page #12.

*“We selected four that were among the top differentially-expressed proteins between the immune hot cluster vs. others (Supplementary Data S2c), had available antibodies applicable to FFPE, and had a practical scoring methodology on carcinoma cells: TAP1 (MHC class I), HLA-DQAI (MHC class II), IFIT2 (type I interferon signaling) and S100A8 by IHC (Figs. 4b-4c). **In addition, these proteins were not highly expressed in the normal reduction mammoplasty samples**”.*

The authors have gone some way to make the paper clearer when it comes to the patient cohorts. However, the results section starts with: A cohort of 300 archival FFPE breast tumor primary tissues,.... All the samples are never used together as a cohort. So this sentence and fig 1A, B are misleading and need to be changed. You need to make it clear in the figure texts and abstract that you are analyzing 2 different cohorts, not one with 300 samples.

11. PAM50 is defined both by RNA and by surrogate IHC markers in the manuscript. However, it is unclear when each definition is used in the manuscript, which makes it confusing to read at times.

PAM50 per definition only refers to RNA not IHC as PAM50 is a RNA-based assay. There is no definition of PAM50 by IHC in the manuscript. We have however now added the word “RNA-based” before the word PAM50 in the section that included IHC data for further clarity.

page #15: “We analyzed 88 IHC defined TNBC cases (profiled by RNA-based PAM50 as: 61 basal-like, 22 Her2-Enriched, and 5 luminal B), all in the 08-13 cohort (Fig. 1b)

ok

12. The authors use a new method denoted isodoping, with the aim to increase the overlap of identifications between TMT sets. The dynamic range in the orbitrap is max 3 orders of magnitude and the practical with TMT is closer to 2 orders of magnitude. To the pool of samples, 4.26 pmol of each peptide is added as isodoping. What is the evidence that you have not added 2 orders of magnitude of your spike in peptide compared to the endogenous levels? Adding spike in peptide amounts in excess of 2 orders of magnitude would make the other TMT channels hover around background and lose quantitative accuracy. How is it checked that this don't affect the quantification used? Could the same scenario happen for the SuperMix channel?

The reviewers make an astute point that issues with the dynamic quantification range can arise when implementing TMT. As shown in Supplementary Fig. S2f, when we compared the average S/N ratio, before normalization, across different sample types we detected an average difference of 3.7x between SuperMix and tumor samples, with all SuperMix samples showing an average S/N comparable to the tumor samples with higher signal.

In Supplementary Fig. S2g, it is displayed that there is only a 3.2x difference between the average abundance of isoDoped peptides and endogenous peptides for isodoped proteins in the PIS+isoDoping channel. When comparing the average S/N of the isoDoping peptides in the tumor samples and the spiked in channel we detected an 8.6x difference, below the suggested limit of 20x (Cheung TK et al. “Defining the carrier proteome limit for single-cell proteomics” *Nature Methods*, 2021).

Ok, I would be curious to see how the TMT profiles compare between isodoped and not isodoped peptides from the same protein.

13. The isodoping is presented in fig1. This to me indicates that it is one of the main concepts in the paper since it comes in the first main figure. However, this is a technicality which the authors say that they are preparing a manuscript for and could be moved to supplementary.

Per the reviewer's recommendation we have moved the isodoping performance to Supplementary Figure S2.

Supplementary Figure S2

14. It is also unclear how the isodoping peptides were selected. Usually peptides are selected due to their good ionization capabilities which could explain much of the results in fig 1b? Figure 1C is unclear to me. How do you reach 74 isodoping dependent proteins? Can you update the figure legend or make a new clearer figure?

As elaborated on in the methods section, the set of synthetic peptides was selected to fulfill the following criteria: (i) include unique peptides for the protein, and (ii) peptides should be between 6 and 20 amino acids long and/or (iii) have physiochemical properties amenable to MS detection. Our isoDoping methodology has been updated and improved in following subsequent experiments for which we have a manuscript under review and can be made available upon request once it is in pre-print. We have also removed Figure 1C from the manuscript.

Ok

15. From results: “The cases in the 08-13 cohort were treated in accordance with contemporary guidelines and contained cases from all four PAM50 subtypes, including all 75 basal-like cases (Supplementary Figs. S1a, S2h, Supplementary Data S1d).”

Which contemporary guidelines are you referring to?

The contemporary guidelines refer to the updated recent guidelines recommended to treat breast cancer commonly used in practice. A reference (Cardoso F et al. Early breast cancer: ESMO Clinical Practice Guidelines for diagnosis, treatment and follow-up. *Annals of Oncology* 2019) has now been added to support this statement.

Good

16. LVI, lymphovascular invasion is mentioned. Don't find it in materials & methods.

Lymphovascular invasion is found in the methods as part of the survival analysis section. The acronym (LVI) has been added to page #34 as well.

Ok

17. When the tumor groups are defined they are given numbers. However, when they are first introduced in figures (for example 2b & c, 5a, 7a) they are not in numerical order. It would maybe be much easier to follow if the clusters are renumbered in numerical order in the first figure where they appear.

The assignment of numbers of the clusters in figures 2b,2c, 5a and 7a is not random and were not manually chosen, but derived from the consensus clustering algorithm we used. The numbers assigned for each cluster are based on the consensus clustering algorithm output and determined in an unsupervised manner by the ConsensusClusterPlus function. If we were to manually change the numbers in figure 2b to be in a numerical order, we would need to force changing the figure itself to follow that order. This will consequently result in changing the numerical order of the clusters in figure 2c again and the reader would not be able to match the cluster names with the consensus matrices plots present in Supplementary figures S5, S10 and S12. This is described in the consensus clustering algorithm of the ConsensusClusterPlus package where it makes

cluster number decisions based on the purity of members in the clusters {Wilkerson MD; ConsensusClusterPlus: a class discovery tool with confidence assessments and item tracking. *Bioinformatics* 2010}.

The lack of numerical order in multiple figures of clustering is confusing and makes the paper more difficult to read and understand. This will translate into fewer people understanding the paper and thus fewer citations etc..

If you want to make it easier for the reader, you can change the order of the clusters manually and just transfer that order between figures. It can all be done easily by a bioinformatician in the R-code.

18. The identification and quantification of 4214 proteins across all samples is a good result for MS analysis of FFPE samples. But could some of the results be explained by not reaching deep enough into the proteome, considering that there should be around 14000 proteins in a tissue according to ProteinAtlas. Could this be a reason for the grouping of Luminal A tumors with Her2 in fig 2b? In the Krug et al *Cell* 2020 paper their tumor grouping almost exclusively only mix luminal A and Bs. No HER2 based on 7679 proteins quantified across all 122 samples. Can the lum A mixing with Her2 be reproduced with the same proteins? Or is this an effect of FFPE?

The composition of our 08-13 cohort is different from CPTAC as our cohort included only 11 luminal A cases compared to 73 basal-like, 62 Her2-Enriched, and 28 luminal B. The 4 clusters displayed in Fig 2b were the best to segregate this cohort by consensus clustering and thus with only 11 cases, luminal A tumors were not found as a unique cluster, but grouped with clusters 1 and 4 that included luminal B and Her2-Enriched in Fig 2b. In these clusters 1 and 4, luminal B and Her2-Enriched were often intermixed which is a commonly known phenomenon in breast cancer subtyping {Prat, A. et al. Molecular features and survival outcomes of the intrinsic subtypes within HER2-positive breast cancer. *JNCI* 2014} and is consistent with the proteomics breast cancer data in {Johansson et al. *Nat Comm* 2019}.

The cluster membership of our cohort compared to the CPTAC breast cancer cohort was dependent on a different combination of cases and in turn our analysis of the 86-92 with more luminal A cases was more powered showing distinctions of two subgroups within the luminal A subtype including a unique luminal A “stromal enriched” cluster, and a cluster that was more a mix of luminal A and B. Thus, overall our results are driven by the biology and the composition of our 08-13 cohort rather than an artifact or a technical limitation.

Ok

19. In fig S3a, you refer to biological replicates. How is biological replicates defined in clinical samples? For the technical replicates, it would have been better if they were spread out in different TMT sets.

The biological replicates refer to different specimens taken from the same patients. We acknowledge that technical replicates were in the same TMT set.

We have added the definition of biological replicates to the text on page #7-8:

“High reproducibility was observed between the biological replicates (referring to different specimens taken from the same patient) (mean $r=0.71$) and the technical replicates (mean $r=0.88$) (Supplementary Figs. S4a-S4b)”.

Ok

20. The PAM50 subtypes have got standard color code. See TCGA 2012 *Nature* or Krug et al *Cell* 2020. To avoid confusion I strongly recommended to use the same color code.

As per the reviewers’ request to make it easier for a reader to compare our results with recent breast cancer ‘omic studies we changed the colors to match the color code used in Johansson et al and Krug et al.

21. In general the authors make a good job in describing their findings. But to make it easier to follow I would suggest to add ER, PR and HER2 status to fig 2C. For example in the text it says: “Most cases in Cluster-2 and -3 were associated with ER, PR and Her2 negativity by IHC clinical tests, high proliferation index (Ki67), and the “core basal” phenotype (defined as ER-, PR-, Her2- and [EGFR+ or CK5+])²⁹ (Supplementary Table 1).” Adding the clinicopathological markers to the heatmap in fig 2c would make it easy to see this in addition to the table. But this is a matter of taste and you can ignore if you like.

We thank the reviewer for this suggestion. Supplementary Table 1, Supplementary Table 2, Supplementary Data S1e and the “results” section describe and elaborate on the correlation between these clinicopathological variables and clusters. Figure 2c is already rich in information and different types of analysis and so we feel that the main emphasis for readers should be the PAM50 subtype membership in each proteome cluster.

22. In fig 2c, 5a, there is a column called immune with 2 categories, Immune related and Other. How are they defined? Also, for the protein groups there are enrichments, how were the enrichments done? Specify in fig text how the terms were selected, representative/ cutoff?

Immune related proteins were defined based on their protein function involvement in immune-response biological processes. Proteins belonging to any of these gene ontology (GO) categories were labeled as Immune:

"GO_DEFENSE_RESPONSE_TO_VIRUS", "GO_RESPONSE_TO_VIRUS",
"GO_RESPONSE_TO_TYPE_I_INTERFERON",
"GO_CELLULAR_RESPONSE_TO_INTERFERON_GAMMA",
"GO_RESPONSE_TO_INTERFERON_GAMMA",
"GO_REGULATION_OF_INNATE_IMMUNE_RESPONSE",
"GO_CYTOKINE_MEDIATED_SIGNALING_PATHWAY",
"GO_ANTIGEN_RECEPTOR_MEDIATED_SIGNALING_PATHWAY",
"GO_IMMUNE_EFFECTOR_PROCESS",
"GO_ACTIVATION_OF_INNATE_IMMUNE_RESPONSE",
"GO_ANTIGEN_PROCESSING_AND_PRESENTATION_OF_PEPTIDE_ANTIGEN_VIA_MHC_CLASS_I",
"GO_FC_EPSILON_RECEPTOR_SIGNALING_PATHWAY",
"GO_POSITIVE_REGULATION_OF_INNATE_IMMUNE_RESPONSE"

For each protein cluster, the most representative terms were selected based on gprofiler enrichment analysis with the following parameters: organism = "hsapiens", ordered_query = FALSE, multi_query = FALSE, significant = TRUE, exclude_ia = TRUE, measure_underrepresentation = FALSE, evcodes = TRUE, user_threshold = 0.05, correction_method = "g_SCS", domain_scope = "annotated", custom_bg = NULL, numeric_ns = "", sources = NULL, term_size < 150 and source in GO:MF, GO:BP or REACTOME'

Raudvere, U., Kolberg, L., Kuzmin, I., Arak, T., Adler, P., Peterson, H., & Vilo, J. (2019). Reference: g:Profiler: a web server for functional enrichment analysis and conversions of gene lists (2019 update). *Nucleic Acids Research*, 47(W1), W191–W198. <https://doi.org/10.1093/nar/gkz369>.

The legends of figures 2c and 5a were updated to include this information.

“Immune related is defined based on the protein function as involved in immune-response biological process and for each protein cluster, the most representative terms displayed on the heatmap were selected based on g:profiler⁴ enrichment analysis”.

The methods section page #31 was updated to include information on the terms selected from the enrichment analysis.

For each protein cluster, the most representative terms were selected and presented on heatmaps based on g:profiler⁷⁷ enrichment analysis with the following parameters: organism = "hsapiens", ordered_query = FALSE, multi_query = FALSE, significant = TRUE, exclude_ia = TRUE, measure_underrepresentation = FALSE, evcodes = TRUE, user_threshold = 0.05, correction_method = "g_SCS", domain_scope = "annotated", custom_bg = NULL, numeric_ns = "", sources = NULL, term_size < 150 and source in GO:MF, GO:BP or REACTOME'.

Ok, good

23. Fig 2a, is this using all or the most varying proteins? In 2b it does not say that the grouping is based on consensus clustering.

UMAP in Fig 2a is based on using all proteins quantified in every sample (4214). The figure legend has been updated accordingly.

The legend of Fig. 2b has been updated to show that the grouping of the different clusters is based on consensus clustering.

(a) Uniform Manifold Approximation and Projection of the 08-13 cohort for the basal-like, luminal A, luminal B, and Her2-Enriched PAM50 subtypes based on all proteins quantified in every samples (4214).

(b) Alluvial plot shows the relationship between PAM50 subtypes and the four proteomic consensus clusters in the 08-13 cohort.

24. In figure S2b-c, the authors show number of peptides per protein. Bit unclear to what it refers to when mentioning peptide? Is that unique peptides? Nr of peptides per protein, is that the per set or total across all TMT sets or mean/median?

It refers to the total number of peptides identified per protein across all TMT sets. The legends for these figures have been updated accordingly.

*(a) Percentage of the total number of proteins detected in different number of samples.
(b and c) Number and percentages of proteins identified according to total number of peptides per protein. Yellow bars in the histogram show the number of proteins identified by different numbers of peptides per protein. Blue dots show the percentage of total proteins identified per minimal number of peptides per protein.*

In my version of suppl info it reads: Number and percentages of proteins identified according to number of peptides per protein. Not total. Mean nr across TMT sets would be more informative, since some sets might have many peptides and some might have few peptides.

25. The number of unique peptides per protein, nr of psm per protein and nr of psms/protein for TMT quantification is missing from the supplementary table with all MS data. Please add this, since it is important when it comes to judging the quantitative robustness. Having said that, must give all the credits for clear clinical information and that the authors include it in the same document so it is easy to access!

We thank the reviewer for this point. As also requested in the reviewer's comment #1, we have added the total number of peptides for each protein, number of unique peptides per protein and number of PSMs used for quantification per protein to the Supplementary Data S1c.

26. In figure 3e, the y-axis says abundance. Is this log₂ ratio to the pool of samples? ESR1 is high in cluster 2 which is one of the basal enriched clusters, which is surprising. Could this be due to isodoping or poor quantification? KRT18 and FOXA1 on the other hand behave as expected.

Protein abundance shown is based on a log₂ ratio for PSM abundances divided by the relative PIS value in each TMT plex. Then for each protein, the median ratio of the 5 most abundant PSMs was used as relative abundance. This is explained in the methods section page #30 and has been added to the legend of Fig. 3e.

“Protein abundance values are based on log₂ ratio for PSMs abundances divided by the relative PIS value in each TMT plex. For each protein, the median ratio of the 5 most abundant PSMs was used as relative abundance”.

The abundance for ESR1 was significantly lower in Cluster-3 than the mean against “all” while ESR1 was non-significantly high in Cluster-2. This could be due to challenges in quantifying

ESR1 as endogenous peptides for this protein were only detected in less than 10% of the samples. Using isoDoping, 3 isoDoping peptides for ESR1 were detected in the majority of samples and thus challenges in ESR1 quantification might explain the non-significantly higher levels observed for Cluster-2.

What is the justification for limiting ratio calculation to top 5 most abundant PSMs? Should you not obtain a more robust median with more values (if available)? How is abundance in this case defined?

The unexpected behavior of ESR1 and PGR are concerning. How does the IHC data correlate to the proteomics data?

27. PECA is used for calculating p-values. Wonder if that inflates the p-values and makes them smaller just because you have a lot of peptides per protein?

<https://pubs.acs.org/doi/10.1021/acs.jproteome.5b00363>

PECA method leverages the number of peptides per protein to assign higher confidence to proteins with higher peptide coverage. While we agree that this method tends to drive the p-value of certain proteins with a particularly high number of peptides, we find it useful to separate proteins with a small number of peptides since these are the ones with lower confidence in quantification levels. We directly compared PECA performance to another differential expression algorithm (DEqMS, Zhu, Y., Orre, L. M., Zhou Tran, Y., Mermelekas, G., Johansson, H. J., Malyutina, A., Anders, S., & Lehtiö, J. (2020). DEqMS: A Method for Accurate Variance Estimation in Differential Protein Expression Analysis. *Molecular & Cellular Proteomics*, 19(6), 1047–1057. <https://doi.org/10.1074/mcp.tir119.001646>) on the first differential expression contrast (Cluster1 vs Cluster2-3-4). We found that the two methods give comparable results in terms of calling differentially expressed (DE) proteins (adjusted p-value < 0.05). We found an overall agreement by DE status on 86% of the proteins: 6% of the proteins differentially expressed in PECA and not in DEqMS, 9% of proteins differentially expressed in DEqMS and not PECA, 11% consistently identified as DE in both methods, and 75% consistently identified as not differentially expressed.

While several differential expression analysis methods are routinely used in the proteomics field and their evaluation over multiple types of data and experiments would be of great interest, we believe that a technical evaluation of PECA and/or comparison with other methods are beyond the scope of this paper.

Ok, this is a point that is good to include in the paper since the general breast cancer biologists reading the paper will not be aware of the inflated p-values and may draw wrong conclusions about the data.

28. Full credit for uploading the immunohistochemistry slides to <http://www.gpec.ubc.ca/prot>. But why limit to representative images. For the dataset to be useful, all images needs to be available. In addition, there should be an easy way to download all data for image analysis.

Our IT team at the Genetic Pathology Evaluation Centre has diligently uploaded all images to <http://www.gpec.ubc.ca/prot>. This information has been updated under the section of “Data availability”, page #34, in the methods.

“Images from immunohistochemistry slides of tissue microarrays used in the study coded as “11-012” and “14-004” are available for public access via the website of Genetic Pathology Evaluation Center (<http://www.gpec.ubc.ca/prot>).

Data image analysis and clinical outcome data for the cases used in this study can be made available through the Genetic Pathology Evaluation Centre and Breast Cancer Outcomes Unit of BC Cancer Centre, upon completion of a Data Transfer Agreement and confirmation of ethical approval for qualified researchers”.

Good

29. To make the data analysis part transparent and reproducible, analysis code should be uploaded to Github or similar repository.

“Code Availability” section has been added to the methods after the “Data Availability” section as requested. Code used for proteomics data analysis is available at GitHub <https://github.com/glnegri/brca>.

Good, but the code seems to only cover basic functions and processing. You need to add the code for consensus clustering and the figures. Also, make sure that the input data is readily available/pointed to.

30. Orbitrap MS2 data was matched with 0.5 Daltons tolerance. This is a very large window that is usually used for iontrap data. For orbitrap MS2, the tolerance should be around 0.02 Dalton to reduce the risk of miss assigning transitions. Since you are also using methylation of lysine as a variable modification, this in combination with a large tolerance will increase your FDR. You should research at least parts of the data and compare the results to your present results to determine if all data needs to be researched. In relation to this, what is the protein FDR of the dataset, q-value, pep value for each protein?

We thank the reviewer for this point. This was actually a typographical error in the text; the data were in fact searched with 0.05 Da tolerance. We have updated the “methods” page #29 accordingly. The full parameters used for the Proteome Discoverer search, together with the results output are available at the PRIDE repository with the dataset identifier PXD024322.
ok

31. From results: FFPE samples were macro-dissected from 3-6 sections to obtain >80% tumor content and analyzed using the SP3-CTP multiplex MS proteomics protocol²⁴ (Supplementary Fig. S1b).

Should it not be ref 19 instead of 24?

Reference #24 {Hughes, C.S., *et al.* Single-pot, solid-phase-enhanced sample preparation for proteomics experiments. *Nat Protoc* **14**, 68-85 (2019)} is a more detailed and up-to-date protocol for the methods used in this study when compared to Reference #19 {Hughes, C.S., *et al.*

Quantitative Profiling of Single Formalin Fixed Tumour Sections: proteomics for translational research. *Sci Rep* 6, 34949 (2016)}. Given that Reference #19 included work done on FFPE (in ovarian cancer) we now include both references #19 and #24 to support our statement.

Ok

New questions based on the updated manuscript

32. In the discussion the authors write: Furthermore, the 9088 total proteins identified is comparable to that achieved using fresh frozen materials (10,107 proteins) by the CPTAC breast cancer project¹⁵. The 4214 proteins quantified in every sample (n=342) across a large-scale breast cancer project using minimal tissue demonstrates the efficiency and high sensitivity of the SP3-CTP approach for FFPE cancer proteomics studies³⁶⁻³⁹.

The total number of identifications are easily achievable in MS based proteomics. The difficulties are to achieve good overlap in quantification across samples. To make it a fair comparison you should also include the nr of overlapping proteins with quantification. You have also used Johansson et al dataset and should add that also in the comparison.

33. In the discussion the authors write: Our result is consistent with a proteomic profiling study of 2 “basal immune hot” cases vs. 7 “basal immune cold” cases using >10mg of frozen tissue¹⁶. For max TMT labelling, 100 µg of peptides are used. This usually equates to 1-2 mg wet weight of tissue. Not >10 mg. where in paper 16 did you find this statement?

34. Many of the supplementary figures are a bite blurry, which needs to be fixed.

35. Check panel labeling in fig S5. The word robust is mentioned multiple times here and in the paper in general. What do the authors mean by robust? What is the criteria(s) that needs to be met to be called robust?

Reviewer #2, expert in bioinformatics and subtype classification (Remarks to the Author):

In this study, the authors carry out mass-spec proteomic profiling of 300 FFPE breast cancer surgical specimens. The specimens are separated into two cohorts based on batch effects. The 08-13 cohort included 75 basal-like, 62 Her2-Enriched, 30 luminal B, and 11 luminal A PAM50 defined cases. The 86-92 cohort provided the long-term outcome data required for luminal cases and included 64 luminal A, 45 luminal B, and 13 Her2-Enriched PAM50 cases. The 08-13 cohort was used for subtype discovery, both across all tumors and within the TNBC subset. ER+

subtypes examined in the 08-13 cohort were examined in the 86-92 cohort.

Specific comments:

1. Batch effects were found between the 08-13 and 86-92 cohorts, likely due to differences in collection techniques, pre-analytical handling, and fixation procedures. Could the authors try to harmonize the two datasets using Combat (<https://rdrr.io/bioc/sva/man/ComBat.html>)? In practice, Combat is very good at removing batch effect differences. Data from different platforms (RPPA, RNA-seq, DNA methylation) have been successfully processed with Combat, and the method is independent of nature of the batch effect. The PAM50 subtype could be used as the experimental group. There would be advantages in having one harmonized dataset of 300 samples. It seems worth a try. As currently written, the Abstract suggests that there is one dataset that was analyzed, rather than two separate cohorts.

As has been shown before, ComBat can lead to overestimating ratios and many in the field believe should be avoided. (Methods that remove batch effects while retaining group differences may lead to exaggerated confidence in downstream analyses <https://academic.oup.com/biostatistics/article/17/1/29/1744261>), especially considering that the batch effect observed in our study is mostly driven by missed identification of peptides cleaved at lysines and not by artifacts on quantification, as shown in figures S3c and S3d. Furthermore, some of the subtypes are completely (basal-like) or almost completely (luminal A) confounded with the ‘cohort’ batch effect. While Combat will always transform the data to minimize batch differences, we believe that for the reasons above, its application in this dataset would lead to serious artifacts in the data.

We would also like to note that the decision to include cases from the 86-92 cohort in our study design was based on clinical and translational considerations. In order for analysis to be meaningful for luminal cases, a long enough follow-up was necessary to obtain sufficient events for outcome analyses. Thus, the majority of luminal PAM50 cases were derived from patients diagnosed with invasive breast cancer in the period January 1986 to September 1992. Forcing the two cohorts to be lumped together for subtyping does not allow obtaining clinically-relevant results for the subtypes found, and could compromise any clinical relevant observations.

We have updated the abstract to highlight that for the 300 cases included there were 2 datasets analyzed rather than one.

“We performed comprehensive proteomic profiling of 300 FFPE breast cancer surgical specimens, 75 of each PAM50 subtype, from patients diagnosed in 2008-2013 (n=178) and 1986-1992 (n=122) with linked clinical outcomes”.

2. Page 8: "Cluster-1 (n=34) consisted mostly of luminal B and Her2-Enriched PAM50 cases. Clusters-2 (n=50) was enriched for basal-like subtype, included few Her2-Enriched, but had no luminal cases. Cluster-3 (n=47) was primarily basal-like cases but included Her2-Enriched cases. Cluster-4 (n=43) was mostly Her2-Enriched but included luminal A and luminal B cases." It seems that actual numbers to reflect the noted associations would be helpful here, e.g. exactly how many basal-like cases and Her2 cases were in Cluster-3, and was Cluster-2 SIGNIFICANTLY enriched for basal-like.

Cluster-2 is enriched for basal-like ($p\text{-value} < 1.16 \times 10^{-11}$, Fisher's test), Cluster-3 is enriched for basal-like ($p\text{-value} < 1.3 \times 10^{-4}$, Fisher's test), Cluster-4 is enriched for Her2-Enriched ($p\text{-value} < 1.9 \times 10^{-4}$, Fisher's test).

The numbers reflecting the breakdown for each PAM50 subtype within each proteome cluster as they appear in Fig. 2b have also been added to the text, page #9.

“Cluster-1 (n=34) consisted mostly of luminal B (n=18) and Her2-Enriched (n=13) PAM50 cases. Clusters-2 (n=50) was significantly enriched for basal-like subtype (n=41), included few Her2-Enriched, but had no luminal cases (p-value < 1.16e-11, Fisher's test). Cluster-3 (n=47) was primarily basal-like cases (n=31) but included Her2-Enriched cases (n=14) (p-value < 1.3e-4, Fisher's test). Cluster-4 (n=43) was mostly Her2-Enriched (n=26) but included luminal A (n=8) and luminal B (n=8) cases (p-value < 1.9e-4, Fisher's test)”.

3. In general, where the word "significantly" appears in the main text, it would be good to include a p-value and associated test to support the claim. The figures referred to likely include the test, but reflecting this in the main text as well would be helpful to the reader. For example, page 11: "The immune hot cluster also had significantly higher CD8+ TILs in the intratumoral compartment compared to other clusters (Fig. 4a)." by what p-value and test?

The p-values and tests are now updated across the text where the word “significantly” appears.

4. Wherever a p-value appears in the main text, the test used to derive that p-value should also be indicated. For example, page 12: "The subgroups with a high expression for only one of these biomarkers were characterized with intermediate RFS (Supplementary Fig. S5b). 70% (21/30) of the cases classified as (TAP1 high/HLA-DQA1 high) were in Cluster-3, while 90% (76/84) of (TAP1 low/HLA-DQA1 low) cases were in other clusters ($p\text{-value} < 0.00001$) (Supplementary Table 1)." What test was used here (we can save the reader from having to go the Table for the answer)?

The test used was the Chi-square test. The text in page #13 has been updated to include this information.

“70% (21/30) of the cases classified as (TAP1 high/HLA-DQA1 high) were in Cluster-3, while 90% (76/84) of (TAP1 low/HLA-DQA1 low) cases were in other clusters (Chi-square p-value < 0.00001) (Supplementary Table 1)”.

5. Page 15: "Multiple correction testing identified fatty acid-binding protein-7 (FABP7) as a candidate biomarker most significantly associated with >10-year RFS on tamoxifen treatment..." Was this the only protein that was significant? Were other proteins significant and using what statistical test and cutoff?

The association between the continuous increase in each individual protein identified in the cohort 86-92 and the endpoint of 10-years RFS was tested using a Cox regression model and stratified log-rank test. This analysis is displayed in Supplementary Data S4f. Only protein biomarkers that had a significant log-rank p-value < 0.05 when adjusted for multiplicity testing by the Benjamini-Hochberg test were selected. **Only** FABP7 protein was found to meet these criteria as displayed in Supplementary Data S4f.

The relevant text for the 86-92 analysis page #18 has been updated to include this information. *“Multiple correction testing identified fatty acid-binding protein-7 (FABP7) as the only candidate biomarker associated with >10-year RFS on tamoxifen treatment (log-rank BHadj $p=0.00004$) (Supplementary Data S4f, Supplementary Fig. S12e)”*.

6. Discussion, page 16. Many journals are uncomfortable with the phrase "(manuscript in preparation)." It seems that the method indicated should be described in sufficient detail in the Methods, if it isn't already.

We believe that the methods regarding the isoDoping methodology are now described in sufficient detail in the methods section of this manuscript for the reader to be able to reproduce the experiment as was intended. While we are currently preparing an even more detailed and comprehensive description of the general isoDoping strategy for a separate primary methodology-oriented publication, to avoid confusion we have deleted the mention of a "manuscript in preparation." From pages #6 and #18.

7. In addition to making the raw data available on ProteomeXchange, it would be most helpful to include the processed proteinXsample tables as Supplementary Data with the published paper. CPTAC has done a similar thing with their past publications.

The proteinXsample data are included in the original Supplementary Data S1c. As requested by reviewer #1, we have also added the peptides identified across the cohort to the Supplementary data S1 along with the total number of unique peptides per protein and number of PSMs used in quantification per protein (Supplementary Data S1c-S1d).

8. For boxplots in the figures, please define the ranges involved.

Boxplot whiskers range extends to the most extreme data point which is no more than 1.5 times the interquartile range from the box. This definition has been added to the legends of Fig. 3e and Fig. 7b.

Reviewer #3, expert in breast cancer subtypes (Remarks to the Author):

1. The authors present their previously described highly sensitive MS-based methodology termed "Single-Pot, Solid-Phase enhanced, Sample Preparation"-Clinical Tissue Proteomics (SP3-CTP). This technology has been shown to capture known and novel features in FFPE tumor samples. The authors have previously shown that this method can be applied on large FFPE material cohorts linked to outcome data. Comprehensive quantification of protein expression can be achieved even from lower input quantities of patient specimens such as small biopsies. Here it would have been useful to know how small?

This is described in the methods section and supplementary Figure S1a. One to six unstained 10 μ m tissue sections were cut for each sample to obtain an aggregate total area of \sim 1cm x 1cm x 10 μ m, with >80% tumor content.

2. In this paper they have applied the method to 300 well-characterized archival FFPE breast

cancer specimens in terms of clinical outcome, IHC, and PAM50 RNA-based intrinsic subtypes. The authors demonstrate that at the protein level one can identify groups characterized by high expression of immune-response proteins and favorable clinical outcomes.

Does this paper bring a sufficient novelty? While it is true that “classifications do not always guide therapeutic choices, due to the extensive heterogeneity that still characterizes breast cancers” can this be solved by adding one more, at the level of proteomics?

As described in the introduction, we performed the current study because genomic classifications of breast cancer are inherently limited as clinical decisions are generally based on the protein level. The underlying technology’s application to FFPE breast cancer material is novel. To the extent that some of the findings overlap with genomic classifications, our study still provides an important verification at the protein level, where most drugs act.

Q1. How do this extension to 300 cases add to what we know from Johansson et al Nat Comm, 2019?

As highlighted in the introduction, Johansson et al. *Nat Comm* 2019 only profiled 9 tumor samples from each of the four main breast cancer PAM50 subtypes, a set which also lacked clinical outcome associations and was insufficient to characterize the biological heterogeneity of breast cancers in relation to clinical behavior and treatment response. In addition, their work required fresh-frozen tissues that are not routinely available from patients, unlike the FFPE clinical specimens we were able to use that can be accessed in larger numbers allowing meaningfully powered linkages to clinical outcomes.

Q2. How does the heterogeneity described here match what is known from RNA based classification (basal also divided in several immune clusters)

The PAM50 subtypes used in this study are an RNA-based classification and the associations of each proteome cluster membership with each PAM50 subtype are described in detail in the manuscript. Within the basal-like RNA-based subtype, there are two distinct proteomic groups that differ in immune response. In the results section, we describe how the heterogeneity of triple negative breast cancer relates to what is known from RNA-based classifications by comparing our findings with those by Burstein M et al. *CCR* 2015, showing that our triple negative clusters were highly correlated with their corresponding RNA subtypes of ‘luminal-androgen receptor’, ‘mesenchymal’, ‘basal-immune suppressed’ and ‘basal-immune activated’.

Q3. If the authors were to make biomarkers based on protein as they suggest, which ones would they choose?

TAP1 and HLA-DQA1, as described in detail in the results and discussion sections. These choices are further supported by the supplementary validation work done in response to reviewer #1, comment #8 as described above (based on the data shown in Supplementary Figures S7 and the new figure S8). We do note that TAP1 and HLA-DQA1 were chosen, in part, because of the availability of quality IHC grade antibodies; it remains possible that other proteins may perform better on IHC-based tests when quality antibodies are available. Indeed, this is one of the prime

utilities of our results for the breast cancer community, to spur additional biomarker research using our data.

The discussion page #21 has been updated with this information.

“Other proteins elevated in the immune hot cluster with available quality antibodies could also be used and developed as candidate biomarkers”.

Q4. An introduction of 5 pages and large number of references (81) makes it into a difficult read. This paper as rigorously performed and described, would benefit from some clarity and simplification, just highlighting the results that move the field forward.

The original work was written in a way that fits the requirements of *Nature Communications*. The introduction here is 2.5 pages double spaced rather than 5 pages as pointed out by the reviewer and the authors hold that this is adequate to succinctly review the pertinent literature, making it hard to remove any essential information from the introduction. As this research sits at a crossroads of breast cancer, bioinformatics, and analytical chemistry the authors believe it is important to provide key background information for scientists from a breadth of related and interested fields to fully appreciate the work. 84 references are merely supporting information for the interested reader to pursue, a number that complies with the *Nature Communications* guidelines (and we are aware of several detailed and comprehensive publications in *Nature Communications* that have a similar or even higher number of references used to properly cover the scientific data presented).

Reviewer #2:

Remarks to the Author:

My previous comments have been addressed with the manuscript revision.

RESPONSE TO REVIEWER COMMENTS

Reviewer #1, expert in proteomics (Remarks to the Author):

Asleh et al have performed quantitative proteomics of 300 breast tumors from formalin fixed paraffin embedded (FFPE) material. The general idea and its potential value to the community of this work is great.

The main Merits of the paper are: 1) the acquisition of proteomics data from FFPE samples across a large number of breast cancer samples with clinical follow up that can serve as a resource, 2) directly linking protein based sample groups with immune infiltration to improved outcome, 3) suggestion of potential biomarkers for tumor groups 4) identification of 4 TNBC groups as previously suggested at the RNA level and linking the immune infiltrated subgroup to good outcome, 5) identification of 3 ER positive tumor groups with a stromal enriched group.

We appreciate the reviewer's view that the work will provide important value to both the fields of breast cancer and of proteomic analysis of patient samples in general.

Limitations regarding merits above:

1. To function as a resource the data needs to be judged as robust.

To evaluate protein quantitative robustness, the number of peptides used for quantification per protein needs to be available and visualized. Now it is lacking from the supplementary data table with all ratios. A panel can also be added to figure 2 to show nr of psms/protein used for quantification.

We thank the reviewer for this point. We have added the total number of peptides for each protein, number of unique peptides per protein and number of PSMs used in quantification per protein to the Supplementary Data S1c. We have also added the data on the peptide abundance per protein (now appears as new Supplementary Data S1d) and PSMs per protein in Supplementary Figure S2d.

Good! However, you need to fix the x-axis. Now it reads:

Fig text: Average number of quantified PSMs per protein, across the full cohort – is that for the subset with quantification across all or including all proteins?

In suppl data S1C the column header says: set_1_number_PSMs – that is nr of psms used for quantification I presume? When you add this information, it would also be informative to add the nr of unique peptides/protein per set. Also, protein scores and q-values are missing from the table. Add a column to easily select the proteins that you have used in your data analysis.

The x axis relates to the PSM data from the 4214 proteins quantified across all samples rather than all 9088 proteins quantified in total. We have updated the text of Figure S2d legends accordingly.

“(d) Average number of quantified PSMs per protein, across the full cohort (corresponding to the 4214 quantified across all samples)”.

Supplementary Data S1c shows the number of PSM per protein after filtering as described in the methods section, which were then used for quantification.

Per the reviewer's request, we added additional columns to Supplementary Data S1c with the PSMs #/protein per set, protein scores per set and q-values per set. In addition, we added a column to quickly identify the proteins used in the analysis (TRUE vs. FALSE).

Supplementary Figure S2

2. The supermix is present in all TMT sets and should represent how well quantifications can be reproduced between TMT sets. Can the supermix data be use for robustness evaluation between the sets? For example a heatmap for overview, variation of supermix in relation to the breast samples and particular sets with deviation on supermix-sample.

We thank the reviewer for this point. The 38 SuperMix replicates included in our experiment showed a high high correlation across the 38 plexes. Unsupervised clustering of our data for all samples including breast tumors, normals, and SuperMix show the SuperMix samples clustered together and are clearly separated from the breast tumor and normal samples. The correlation between the SuperMix samples was the highest when compared to the breast tumor and normal samples, supporting the robustness of the evaluation of SuperMix samples across the sets (appears as new Supplementary Fig. S4c). Pairwise correlation between the 38 SuperMix replicates (ranged between 0.68-0.81, median 0.75) was significantly higher than the pairwise correlation across the 38 normals (ranged between 0.53- 0.85, median 0.71). These findings are shown in new Supplementary Fig. S4d.

The following information has been added to the results section page #8:

“An overview clustering of all the samples included in our study showed that the 38 SuperMix replicates had the highest correlation across the 38 plexes (range 0.68-0.81) when compared to the breast tumors and normal samples (Supplementary Figs. S4c-S4d).

The small difference in correlation between the Supermix, that should be exactly the same sample in all TMT sets, and the normal samples, which are biologically different are surprising. The supermix should represent technical variation and in this case are very close to the biological variation. The large number of proteins used in the sample to sample correlation analysis will provide a relatively high correlation, which limits this analysis.

To be able to support the claim of the dataset as resource, the reader needs to be able to better understand the technical variation in the dataset. For example, you could calculate coefficient of variation for each protein based on the supermix and plot that.

Also, you have IHC data for some proteins as ESR1, PGR etc, how these measurements correlate to the proteome data would be useful for judging the quality of the data.

Indeed, as shown in the plot below, we expected a tighter correlation between SuperMix replicates since they should represent the technical variation across TMT plexes. However, we found that the SuperMix shows an average higher variation across the cohort compared to the one observed in the (biologically distinct) normal samples. We believe that this increase in variation is the result of the very different background matrix composition of the samples as the SuperMix includes 13 different cancer models cultured in vitro (as described in the methods) while the rest of the cohort consists of breast tissue, which was FFPE preserved. This under-representation (1/11th of the channels) of the SuperMix matrix makes it more likely to be affected from isolation interference and background noise leak from the breast tissue FFPE samples. For these reasons, while the SuperMix is an important reference standard that allows future comparisons with any cohorts that will include a SuperMix control in the design, it doesn't completely reflect the true technical variation in this cohort. However, a better representation of the technical variation can be estimated from the technical (n= 3) and biological (n= 3) tumor replicates that we included as part of the cohort (Supplementary Fig. S4a-b).

Supplementary Figure S4

We include the comparison of ER, PR and HER2 IHC results with the proteome data in Supplementary Fig. S7 along with the validated IHC markers mentioned in the reviewer's comment #8.

Supplementary Figure S7

a

b

Supplementary Fig. S7. Correlation between proteomic abundance scores vs. IHC for selected proteins.
 (a) Relative abundance of ESR, PGR and HER2 by Mass spectrometry according to their IHC categories.
 (b) Correlation of protein expression values for protein candidates by mass spectrometry vs. IHC. Scoring values of the S100A8, TAP1, IFIT2 and HLA-DQA1 IHC biomarkers were reported using the H scoring system (intensity x

positivity) for the cytoplasmic staining observed in the invasive breast tumor cells. Spearman correlations are shown on each panel. Abbreviations: IHC, immunohistochemistry.

The categories of ER, PR and Her2 are assigned per the available pathological data extracted from the patients' charts reporting hormone receptor status and HER2 as positive vs. negative. A highly significant association was observed between the proteomic relative abundance of HER2 and clinical HER2 status. A significant association between the proteomic relative abundance of ESR1 and ER IHC status was also observed. PGR relative abundance was overall higher in PR+ by IHC, but this result was not significant.

In addition, the results page #12 were updated accordingly to include these results:

"When testing the association between the MS data for ESR1, PGR, HER2 and their IHC categories, results were significant for HER2 ($p < 0.0001$) and ER ($p = 0.02$) IHC expression (Supplementary Fig. S7)".

As is also explained in detail in our response to comment #8, there are several reasons why different IHC biomarkers could differ in their association with the proteomic data. ER and PR assessment were performed per the current established guidelines that evaluate their **nuclear staining on carcinoma cells only, using pre-established clinically validated cutpoints to report results categorically as positive vs. negative**. In contrast, the MS relative abundance does not consider **this spatial information** when reporting the overall protein scores. The inference of the protein level in MS is based on peptide level quantification, while IHC is semi-quantitative with the inherent limitations of being an antibody-based assay with analytical and preanalytical issues that can affect the results.

3. An overview clustering of the 2 cohorts with replicates would also be useful to judge how the whole dataset behaves. Does the technical replicates cluster together?

Per the reviewer's request, we generated a heatmap showing the overview clustering for all the samples, as also requested in the previous comment. As now shown in Supplementary Fig. S4c, the three technical replicates indeed have clustered adjacent to each other (T_rep 5, T_rep 6, T_rep 7). Regarding the biological replicates, 2 of 3 replicates clustered adjacent to each other while the 3rd biological replicates clustered very closely together, a variance in line with expectations for intratumoral regional sampling. The normal samples clearly separated from tumor samples and showed an overall correlation of 0.70. An overall correlation of 0.5-0.6 was observed for the different breast tumor clusters and these included a mix of samples from both 08-13 and 86-92 cohorts.

This information has been added to the results section page #8:

"All the technical replicates and 2 out of 3 biological replicates clustered adjacent to each other, while the 3rd biological replicates clustered very closely together, a variance in line with expectations for intra-tumoral regional sampling (Supplementary Fig. S4c).

Ok

4. The data should also confirm with previous knowledge, as ER, PR, HER2, MKI67 levels in different PAM50 subtypes, and this would be good to show in a supplementary figure.

HER2 (ERBB2) and MKI67 expression levels across the different PAM50 subtypes are found in Supplementary Fig. S6d. ER (ESR1) and PR (PGR) expression levels across the different PAM50 subtypes are now also included in Supplementary Fig. S6d.

Ok, see my comment to question 2.

We include the comparison of ER, PR and HER2 IHC results with the proteome data as explained in the above response to comment #2.

5. Proteomics have previously identified immune infiltration in breast cancer subgroups without directly linking them to outcome (Krug 2020 Cell, Johansson et al 2019 Nat Comm). Tumorinfiltrating lymphocytes (TILs) have also been linked to better outcome in breast cancer subtypes (Dieci 2021 Cells). The strength of this study is the direct link between proteomics data with "immune hot" tumors and outcome.

Relation to published data

In general, anchoring the novel findings further, e.g. by validation of findings in other breast proteomics data sets would be valuable to show the usefulness of the data as a resource and strengthen the findings. There is several decent datasets published now on breast cancer proteome so this should be done.

Per the reviewer's request, we performed a validation of our findings on previous proteomic datasets published by Krug et al. Cell 2020 (CPTAC) and Johansson et al. Nat. Commun 2019 (OSLO2).

Validation using the Krug et al. 2020 CPTAC breast tumor cohort: In order to compare our results with available published datasets, we performed consensus clustering with the same parameters used in our cohort on the CPTAC Cell 2020 cohort, using the 939 proteins from the CPTAC data that overlap with the 1054 mostly highly-variant proteins of our 08-13 cohort. This analysis identified four main proteome clusters that highly resembled the original CPTAC NMF clusters of “LumA-I”, “LumB-I”, “Basal-I”, “HER2-I”. Two of these were almost entirely similar to the original NMF clusters of “Basal-I”, and “LumA-I”. Another cluster highly resembled NMF “LumB-I” and consistent with Krug et al consisted of 54% luminal A cases (compared to 55% luminal A cases assigned as “LumB-I” in the original NMF CPTAC clusters by Krug et al). Similar to the original NMF CPTAC clustering composition, the NMF CPTAC “HER2-I” cluster identified had a mix of Her2-Enriched, luminal A and luminal B breast cancers. Of note, the original Krug et al 2020 study of 122 breast tumors included a majority of luminal A PAM50 subtype (n=57, 47%), followed by basal-like (n=29, 24%), luminal B (n=17, 14%) and Her2-Enriched (n=13, 11%) when compared to the composition of our 08-13 cohort which consisted of a higher number of basal-like (n=73, 42%) and Her2-Enriched (n=62, 36%) cases, but few luminal A (n=11, 6%). Despite this, our analysis further reproduced the existence of subsets enriched for immune response pathways at the proteome level within the basal-like and Her2-Enriched subtypes not captured in the CPTAC analysis. Consistent with our analysis on the 08-13 cohort, stromal pathways were enriched in luminal A tumors and lipid metabolism was enriched within luminal B and Her2-Enriched tumors. A description of these findings is displayed in Supplementary Fig. S10a.

In the results section you write: Our analysis reproduced the existence of subsets enriched for immune response pathways at the.... These subsets are within your clusters. They don't come out as defined clusters. You need to make that clear. It looks though as it should be possible to separate out immune enriched samples.

We agree with the reviewer. Our analysis of the CPTAC breast tumor cohort did not demonstrate these as separate defined clusters, though it seemed possible to separate out some immune enriched samples that were classified as basal-like and Her2-Enriched. In contrast, the analysis of our 08-13 cohort revealed an “immune hot” cluster that was referred to as a defined and distinct cluster. These differences might be because of the reasons explained above in our original response regarding the composition of our 08-13 cohort, which includes a much higher number of basal-like and Her2-Enriched cases when compared to the CPTAC cohort.

Overall, our analysis on the CPTAC cohort illustrates that there is a fraction (subset) within the basal-like and the Her2-Enriched subtypes that are enriched for immune response pathways. For clarity, we have replaced the word “reproduced” with “demonstrated” in the sentence mentioned in the reviewer’s comment and updated this sentence in the results section page #14 and in the legend of Supplementary Figure S10a as below, highlighting that these were not captured as defined clusters in the CPTAC analysis.

“Our analysis demonstrated the existence of subsets enriched for immune response pathways at the proteome level and these included basal-like and Her2-Enriched subtypes. In contrast to the 08-13 cohort, these subsets were not captured as separate and defined clusters by CPTAC analysis”.

Validation using the Johansson et al 2019 “OSLO2 breast cancer landscape cohort”:

To validate our findings on the 36 cases of the 4 main subtypes (9 for each PAM50 type) in the “OSLO2 landscape cohort”, we performed consensus clustering with the same parameters used in our analysis, using the 775 proteins from the OSLO2 data that overlap with the 1054 mostly highly-variant proteins of our 08-13 cohort. This analysis identified 4 clusters that highly resembled the main consensus core tumor clusters (CoTCs) and their biological functions as reported in Johansson et al. These clusters consisted of CoTC1 (basal-like immune cold), CoTC2 (basal-like immune hot), CoTC3 with few CoTC6 cases (luminal A-enriched) and CoTC6 (luminal B and Her2-Enriched). Importantly, the immune distinctions within the basal-like subtype were entirely reproduced using our highly variant proteins showing that the two basal-like samples of OSL.3EB and OSL.449 (CoTC2) were consistently classified as “basal immune hot cluster” when compared to other basal cases characterized as “basal immune cold”. These findings are displayed in Supplementary Fig. S10b.

The results section page #14 has been updated to include our comparison analysis using the Krug et al 2020 and Johansson et al 2019 proteomics datasets, as a new section entitled “Comparison with previous breast cancer proteomics studies”.

The number of immune hot samples are a little bite low, but in the other hand supports your findings.

Indeed. We agree with the reviewer that the number used in Johansson et al. is extremely low when compared to our dataset and we highlight that in the comparison we make in discussion section pages #19-20. To date, the only proteomic published data preceding our current study which showed the existence of defined immune hot vs. immune cold clusters consisting of basal-like cases is Johansson et al, and thus despite its limitations serves as the best available proteomic dataset for comparison. It does support our findings as highlighted in the introduction page #4 and the results section page #15.

Supplementary Figure S10

a) Validation using the Krug et al. 2020 CPTAC breast tumor cohort

b) Validation using the Johansson et al 2019 OSLO2 breast cancer landscape cohort

6. The 4 TNBC groups are correlating to their suggested RNA based groups. To strengthen the finding of 4 TNBC subtypes, can they be identified also at the protein level, for example in Krug 2020 Cell data?

We validated our TNBC proteome clusters using the 935 proteins from Krug et al that overlap with the 1055 mostly highly-variant proteins in our 08-13 TNBC (n=88) subset on the set of 28 TNBC cases included in the CPTAC breast cancer cohort by Krug et al. Our analysis reproduced the existence of the four main proteome TNBC subgroups and the biological features of 'luminal-androgen receptor', 'mesenchymal', 'basal-immune suppressed', and 'basal-immune activated' as now shown in Supplementary Fig. S12.

The results section page #16 has been updated to include this information:

"The existence of these TNBC proteome clusters and their biological features were validated when applying consensus clustering, with identical parameters, on the 935 proteins overlapping with the 1055 mostly highly-variant proteins of our 08-13 TNBC subset on the proteomic data for a set of 28 TNBC cases included in the CPTAC breast cancer cohort by Krug et al (Supplementary Fig. S12).

Supplementary Figure S12

Ok, good!

7. How generalizable are the 3 ER positive tumor subgroups identified in the manuscript? The authors cite Krug 2020 Cell in the discussion as consistent with the stromal-enriched subtype. But to my knowledge, the data in the Krug paper don't show a separate luminal A subgroup

enriched for stroma. Dennison 2016 CCR, however show a stromal subtype of ER positive tumors that are or mixed subtype but enriched in Luminal A with a favorable clinical outcome. Are the same proteins (in RPPA and your MS data) deterministic of the stromal subgroup?

We agree with the reviewer that Krug 2020 Cell did not identify a separate stromal enriched subtype as a unique cluster by mass spectrometry, but described a subset of luminal A tumors as stromal-enriched since these tumors were classified originally as “reactive” in the TCGA 2012 RPPA data. In the subsequent Nature 2016 CPTAC proteomics profiling breast cancer publication, the proteomic cluster that was highly correlated with the “reactive” RPPA cluster was referred to as stromal-enriched.

The Dennison 2016 CCR study basically tried to characterize the biological and clinical features of the stromal enriched tumors as a whole (i.e. reactive tumors) identified in the TCGA based on RPPA data. The majority of these tumors were found to be classified as luminal A by PAM50, and among the luminal A as a group those that had high stromal protein expression displayed favorable clinical outcomes.

Comparing the proteins in our MS data that are in common with the RPPA proteins (n=30) used to classify the “stromal-enriched” vs. the “ER positive cancer derived” subtypes in Dennison 2016 CCR, we found 5 proteins in the RPPA “stromal-enriched” Dennison 2016 CCR that were also characteristic for our luminal A stromal enriched proteomics cluster (log2FC>0.20, adjusted p-value<0.05). These were fibronectin, annexin, collagen VI, caveolin, and MYH11.

While our data correlate with those results, the RPPA data only cover a small percentage of the proteome that was quantified in our experiment; thus, our data characterize the luminal A stromal enriched cluster in a more comprehensive manner and identify protein candidates that are beyond those captured by the restricted number of proteins in the antibody-based RPPA assay.

The discussion page #23 has been updated to highlight this information.

“Our analysis of ER+ cases with mature clinical data identified a stromal-enriched subset (86-92-Cluster-2) consistent with previous reports^{57,63}, which could help sub-classify luminal breast cancer. However, our data characterize the luminal A stromal enriched cluster in a more comprehensive manner and identify protein candidates that are beyond those captured by the restricted number of proteins in the antibody-based RPPA assay”.

OK

8. IHC validation of S100A8, TAP1, IFIT2, HLA-DQA1 and CD8 as suggested biomarkers of immune infiltration and better outcome are done on the same cohort as the proteomics. To consolidate the findings, validation in an independent cohort would be valuable. Also, what is correlation between the MS data and the IHC validated markers? Are the MS protein levels also related to outcome?

First part of the reviewer’s comment: Per the reviewer’s request, we have now performed a validation of these IHC biomarkers on an independent set of 176 breast cancer cases with similar clinicopathological characteristics to the 08-13 cohort. Our analysis confirmed that high expression of HLA-DQA1 as a single biomarker had a significantly better survival (log-rank p=0.02) and a similar trend was seen with high TAP1 as a single biomarker (log-rank p=0.09). The findings further confirmed that tumors with IHC expression for both TAP1 and HLA-DQA1 showed the most favorable survival, while the subgroup with low expression for both had the worst RFS (log-rank p=0.05) (Supplementary Fig. S9).

Supplementary Figure S9

The results section page #14 has been updated with this information:
 “We subsequently confirmed our observations on an independent, clinically similar set of 176 breast cancer cases and showed that high expression of HLA-DQA1 as a single biomarker

had a significantly better survival (log-rank $p=0.02$) and a trend was seen for high TAP1 as a single biomarker (log-rank $p=0.09$). These data also confirmed that tumors with high IHC expression for both TAP1 and HLA-DQA1 showed the most favorable survival, while the subgroup with low expression for both had the worst RFS (log-rank $p=0.05$) (Supplementary Table 3; Supplementary Fig. S9).

The Supplementary methods in the Supplementary Information file page #27 and Supplementary Table 3 include information on the characteristics of this IHC validation cohort:

“IHC validation cohort: A tissue microarray for an independent set of 176 breast cancer cases was used to validate observations on the 08-13 cohort for the key protein IHC biomarkers. This validation cohort had clinicopathological characteristics similar to the 08-13 cohort and was analyzed for IHC biomarker association with clinical outcomes. The median follow-up for the IHC validation cohort was 10 years and cases were treated in accordance with contemporary guidelines”. Characteristics of this cohort appear in the new updated Supplementary Table 3.

Supplementary Table 3

Characteristic	IHC Validation cohort (n=176)
Age at diagnosis (median)	53 years
Tumor size (median)	2 cm
Tumor grade	
1, 2	44 (25%)
3	127 (72%)
Missing	5 (3%)
Nodal status	
Negative	105 (60%)
Positive	66 (37%)
Missing	5 (3%)
IHC subtype	
Luminal ([ER+ or PR+])	69 (39%)
ER-, PR-, HER2+	32 (18%)
ER-, PR-, HER2-	71 (40%)
Missing	4 (3%)
Disease specific death	
No	134 (76%)
Yes	35 (20%)
Missing	7 (4%)
CD8 iTILs	
<1%	42 (24%)
≥1%	129 (73%)
Missing	5 (3%)
TAP1/HLA-DQA1 IHC groups	
TAP1 high /HLA-DQA1 high	35 (20%)
TAP1 low /HLA-DQA1 high	22 (13%)
TAP1 high /HLA-DQA1 low	50 (28%)
TAP1 low /HLA-DQA1 low	65 (37%)
Missing	4 (2%)

Good!

Second part of the reviewer’s comment: The Spearman correlation between the MS data and the H score for the IHC validated markers was found to be 0.51 for TAP1 and S100A8, 0.31 for HLA-DQA1, and 0.11 for IFIT2 as shown in the figure below. Of note, the assessment of the validated markers by IHC was performed on the carcinoma cells.

This data should also be in the paper together with the same kind of analysis for ESR1 and PGR. Why do you think the correlations are weak? For TAP1 and HLA-DQA1 that performs well together, what is the difference in signal that is picked up by IHC and MS? Both are prognostic but show weak correlations indicating different signal/information that they pick up.

Per the reviewer's request, this analysis has been added as a new Supplementary Figure S7.

The analysis according to the IHC ER score with ESR1 proteomic abundance and the IHC PR score with PGR proteomic abundance is described in comment #2 and included in the manuscript under new Supplementary Figure S7. In addition, the results page #13 were updated accordingly:

"When assessing the correlation between the MS data and the IHC scores for the validated biomarkers, a low-moderate correlation was noted (Supplementary Fig. S7)".

Regarding the weak-moderate correlation between IHC and MS data, there are several explanations. Firstly, as explained in the methods section, the assessment of the validated markers by IHC was performed following practical and established IHC methodologies to assess their expression **only** on the invasive carcinoma cells and using the H score that in addition to **positivity** also takes into account the **intensity** when reporting the IHC expression. These 2 components of positivity x intensity are multiplied to give the overall score. Importantly, for these biomarkers scores were reported for the **cytoplasmic staining only** that was observed in the invasive breast tumor cells, using a tissue microarray format with duplicate cores for each specimen. Intensity scores were reported as (0: none, 1: weak, 2: moderate, 3: strong) and the positivity proportion scores were reported as (1-100%) for each core. The averaged cytoplasmic H score between the duplicate cores per case was used for the scoring of the protein expression by IHC. Secondly, when analyzing tumor specimens by MS, the whole section is analyzed and the expression of specific proteins is not measured in the context of spatial expression on invasive carcinoma cells only and considering appropriate subcellular (cytoplasmic) expression only. Furthermore, the representative cores assessed on tissue microarray do not always represent the expression on the whole slide taken from the source block, but rather represent the expression of the relevant biomarker specifically in the most histologically-representative viable invasive carcinoma areas punched out as cores to construct these tissue microarrays. Thirdly, there are several analytical and preanalytical differences related to IHC as an antibody-based assay vs. MS that contribute to the correlations observed with these biomarkers. IHC is semi-quantitative due to the fact that it is antibody hybridization-based (with the signal amplified using secondary antibodies and linked enzymatic chromogen activation) while the inference of the protein level in MS is based on the peptide level quantification that is more quantitative than IHC. Altogether, these are reasons why while MS-IHC data would be expected to show weak-moderate correlation, they could still both be prognostic.

These reasons were briefly summarized and included in the discussion page #21.

"Of note, the assessment of the validated markers by IHC was performed only on the carcinoma cells and using the H score that in addition to positivity, takes into account staining intensity when reporting the IHC expression. These variables along with using a TMA format and the differences related to IHC as a multi-step antibody-based assay vs. MS contribute to the weak-moderate correlations observed with these biomarkers".

Third part of the reviewer's comment: The selection of the biomarkers for IHC validation was based on biology rather than clinical outcomes. In response to the reviewer's comment, we performed a Cox proportional-hazards analysis on the protein abundance (in MS data) and recurrence free survival for the protein candidates we assessed by IHC. MS protein levels are significantly correlated with improved outcome for TAP1 and IFIT2, while a trend is shown for HLA-DQA1 and S100A8 as follows:

Protein	Survival analysis for RFS HR (95% CI), P -value	Adjusted P -value
TAP1	0.34 (0.18-0.65), 0.001	0.04
HLA-DQA1	0.87 (0.69-1.10), 0.24	0.71
S100A8	0.87 (0.73-1.06), 0.16	0.62
IFIT2	0.38 (0.18-0.80), 0.01	0.19

Additional comments:

9. From introduction: “This method can query large FFPE material cohorts linked to outcome data, enabling comprehensive quantification of protein expression from lower input quantities of routinely-available patient specimens, and employs a more highly efficient workflow than other MS-based methods for protein profiling of clinical FFPE tissues^{21,22}. “

Based on the data, the MS workflow seems efficient, but there is really no data to comparing all other methods to support your claim of “more highly efficient workflow than other MS-based methods...”? Many of the large MS proteomics groups have published their versions of FFPE sample preparation methods. See for example Coscia 2020 Modern Pathology, Griesser 2020 MCP, Marchione 2020 JPR, Zhu 2019 Molecular Oncology.

We thank the reviewer for this comment. We have updated this sentence in the introduction page #5 accordingly:

“This method can be used to query large FFPE material cohorts linked to outcome data, enabling comprehensive quantification of protein expression from lower input quantities of routinely-available patient specimens, and employing a more highly efficient workflow than other MS-based methods for protein profiling of clinical FFPE tissues^{21,22} .

Ok

10. In the abstract and in figure one, 300 samples are mentioned as included in the study. The number is correct but it is bit misleading since it's divided up in 2 cohorts. The overview in Figure 1A is not useful since this collection of samples are not used together later on in the paper. The overview presented in fig S1A are much more useful since it gives an overview of the samples used together in each of the later analyses. Also the number of samples drop after QC and removal of replicates. To make it clearer for the reader I suggest you make a combination figure of fig S1A and S2H with the tumor characteristics and the numbers that make up each cohort used in the downstream analysis. Also include the info of how the TNBC cohort was made. This took time to figure out and with a figure outlining the 2 cohorts, it would be much clearer from the beginning for the readability of the entire paper. To make it even clearer one could add what type of analysis / aim you have with each cohort. There is also normal samples for which it is unclear of their purpose/how they are used. Did not find any comparison to the normal samples in the text?

Per the reviewer's recommendation we have moved the original Figures S1A and S2H to Figure 1. Now they appear as Fig. 1b and Fig. 1c.

Given that normals were sourced from independent reduction mammoplasties, they are very biologically different from tumors and thus they are not helpful in the subtyping or performing direct comparisons with tumor samples. The normals were included in the UMAP plots where they form a clearly separated cluster from tumors, added to the heatmaps (Figures 2c and 5a) as a reference to illustrate that proteins and pathways of interest for the proteome clusters were not high in normals, and as a visual comparator for the expression of key breast cancer associated proteins in Supplementary Figure S6d.

In addition, when we picked specific proteins of interest for validation in IHC, we used candidates that were not highly expressed in normals. We updated the text to include this specific information in page #12.

“We selected four that were among the top differentially-expressed proteins between the immune hot cluster vs. others (Supplementary Data S2c), had available antibodies applicable to FFPE, and had a practical scoring methodology on carcinoma cells: TAP1 (MHC class I), HLA-DQA1 (MHC class II), IFIT2 (type I interferon signaling) and S100A8 by IHC (Figs. 4b-4c). In addition, these proteins were not highly expressed in the normal reduction mammoplasty samples”.

The authors have gone some way to make the paper clearer when it comes to the patient cohorts. However, the results section starts with: A cohort of 300 archival FFPE breast tumor primary tissues,.... All the samples are never used together as a cohort. So this sentence and fig 1A, B are

misleading and need to be changed. You need to make it clear in the figure texts and abstract that you are analyzing 2 different cohorts, not one with 300 samples.

We thank the reviewer for this point. The design of this study was to include 300 total samples such that in sum they would represent 75 samples from each 4 main PAM50 subtype. Among the total 300 samples, luminals were mostly collected from an older cohort, so as to allow meaningful clinical outcomes that can only be captured by using long-term follow-up for these clinically less-aggressive cases, as described in the first paragraph of the results section page #6. The original intent of the study design was to analyze the 300 samples as a single cohort and thus a mix of cases from "08-13" and "86-92" were spread across the 38 11-plexes when we ran the study. Thus, the MS data were collected as a single cohort design. However, as explained in detail in page #7, due to the batch effects observed we analyzed the total 300 cases as two separate cohorts.

Per the reviewer's request, the paragraph in page #6 has been updated accordingly:

A total of 300 archival FFPE breast tumor primary tissues, representing 75 from each of the RNA PAM50 subtypes⁴, and 38 normal reduction mammoplasty samples, were obtained (Fig. 1a-1b). Samples were assembled with an original aim to be analyzed as one cohort, thus the MS data were obtained per this design, from patients diagnosed with invasive breast cancer using tissue obtained prior to adjuvant systemic therapy in 2008-2013 (n=178; the 08-13 cohort) and 1986-1992 (n=122; the 86-92-cohort). The 08-13 cohort included 75 basal-like, 62 Her2-Enriched, 30 luminal B, and 11 luminal A PAM50 defined cases. The 86-92 cohort provided the long-term outcome data required to gather sufficient outcome events for luminal A breast cancers and included 64 luminal A, 45 luminal B, and 13 Her2-Enriched PAM50 cases (Fig. 1b).

Figure 1b shows the breakdown of the two cohorts included according to "time of collection" to indicate the difference between 08-13 vs. 86-92 cohorts that were analyzed separately. The word "cohort" and the description of the cohorts has been added to the x axis in Figure 1b for further clarity. In addition, per the reviewer's request, the legend for Figure 1 has been updated:

Figure 1. Proteomic analysis of FFPE breast cancer tissue samples

- (a) *The clinical features of the 300-tumor study cohort across the four PAM50 breast cancer subtypes. Samples were assembled from patients diagnosed with invasive breast cancer using tissue obtained prior to adjuvant systemic therapy in 2008-2013 (n=178; the 08-13 cohort) and 1986-1992 (n=122; the 86-92-cohort). While the MS data were obtained with the 08-13 and 86-92 samples intermixed (see Fig S1b batch design), these two cohorts were analyzed separately. Pathological primary tumor size is defined as (T1 <=2cm), (T2 2-5cm), (T3 >5cm); recurrence, (local, regional, distant). The feature list is in Supplementary Data S1e. LVI, lympho-vascular invasion; TNBC, triple-negative breast cancer.*
- (b) *The distribution of the PAM50 subtypes for the 300 tumor samples described in (a) across the 86-92 and 08-13 cohorts. The study also included 38 normal breast reduction mammoplasty samples. Within the 08-13 cohort, a set of 88 cases were classified as TNBC by IHC and were analyzed as a separate cohort.*

We further updated the abstract per the reviewer's request:

"Despite advances in genomic classification of breast cancer, current clinical tests and treatment decisions are commonly based on protein level information. Formalin-fixed paraffin-embedded (FFPE) tissue specimens with extended clinical outcomes are widely available. We performed comprehensive proteomic profiling of 300 FFPE breast cancer surgical specimens, 75 of each PAM50 subtype, from patients diagnosed in 2008-2013 (n=178) and 1986-1992 (n=122) with linked clinical outcomes. These two cohorts were analyzed separately and we quantified 4214 proteins across all 300 samples...."

11. PAM50 is defined both by RNA and by surrogate IHC markers in the manuscript. However, it is unclear when each definition is used in the manuscript, which makes it confusing to read at times.

PAM50 per definition only refers to RNA not IHC as PAM50 is a RNA-based assay. There is no definition of PAM50 by IHC in the manuscript. We have however now added the word "RNA-based" before the word PAM50 in the section that included IHC data for further clarity.

page #15: "We analyzed 88 IHC defined TNBC cases (profiled by RNA-based PAM50 as: 61 basal-like, 22 Her2-Enriched, and 5 luminal B), all in the 08-13 cohort (Fig. 1b)

Ok

12. The authors use a new method denoted isodoping, with the aim to increase the overlap of identifications between TMT sets. The dynamic range in the orbitrap is max 3 orders of magnitude and the practical with TMT is closer to 2 orders of magnitude. To the pool of samples, 4.26 pmol of each peptide is added as isodoping. What is the evidence that you have not

added 2 orders of magnitude of your spike in peptide compared to the endogenous levels?
Adding spike in peptide amounts in excess of 2 orders of magnitude would make the other TMT channels hover around background and lose quantitative accuracy. How is it checked that this don't affect the quantification used? Could the same scenario happen for the SuperMix channel?

The reviewers make an astute point that issues with the dynamic quantification range can arise when implementing TMT. As shown in Supplementary Fig. S2f, when we compared the average S/N ratio, before normalization, across different sample types we detected an average difference of 3.7x between SuperMix and tumor samples, with all SuperMix samples showing an average S/N comparable to the tumor samples with higher signal.

In Supplementary Fig. S2g, it is displayed that there is only a 3.2x difference between the average abundance of isoDoped peptides and endogenous peptides for isodoped proteins in the PIS+isoDoping channel. When comparing the average S/N of the isoDoping peptides in the tumor samples and the spiked in channel we detected an 8.6x difference, below the suggested limit of 20x (Cheung TK et al. "Defining the carrier proteome limit for single-cell proteomics" Nature Methods, 2021).

Ok, I would be curious to see how the TMT profiles compare between isodoped and not isodoped peptides from the same protein.

For the reviewer's request, the figure below shows the correlation between the protein abundance measured by isoDoped peptides only vs protein abundance measured by the endogenous peptides only (for the same proteins). The proteins shown on the plot are the ones from the 4214 set of proteins identified across all the samples for which at least 3 isoDoping and 3 endogenous peptides were included.

13. The isodoping is presented in fig1. This to me indicates that it is one of the main concepts in the paper since if it comes in the first main figure. However, this is a technicality which the authors say that they are preparing a manuscript for and could be moved to supplementary. Per the reviewer's recommendation we have moved the isodoping performance to Supplementary Figure S2.

Supplementary Figure S2

14. It is also unclear how the isodoping peptides were selected. Usually peptides are selected due to their good ionization capabilities which could explain much of the results in fig 1b? Figure 1C is unclear to me. How do you reach 74 isodoping dependent proteins? Can you update the figure legend or make a new clearer figure?

As elaborated on in the methods section, the set of synthetic peptides was selected to fulfill the following criteria: (i) include unique peptides for the protein, and (ii) peptides should be between 6 and 20 amino acids long and/or (iii) have physiochemical properties amenable to MS detection. Our isoDoping methodology has been updated and improved in subsequent experiments for which we have a manuscript under review and can be made available upon request once it is in pre-print. We have also removed Figure 1C from the manuscript.

Ok

15. From results: "The cases in the 08-13 cohort were treated in accordance with contemporary guidelines and contained cases from all four PAM50 subtypes, including all 75 basal-like cases (Supplementary Figs. S1a, S2h, Supplementary Data S1d)."

Which contemporary guidelines are you referring to?

The contemporary guidelines refer to the updated recent guidelines recommended to treat breast cancer commonly used in practice. A reference (Cardoso F et al. Early breast cancer: ESMO Clinical Practice Guidelines for diagnosis, treatment and follow-up. Annals of Oncology 2019) has now been added to support this statement.

Good

16. LVI, lymphovascular invasion is mentioned. Don't find it in materials & methods.

Lymphovascular invasion is found in the methods as part of the survival analysis section. The acronym (LVI) has been added to page #35 as well.

Ok

17. When the tumor groups are defined they are given numbers. However, when they are first introduced in figures (for example 2b & c, 5a, 7a) they are not in numerical order. It would maybe be much easier to follow if the clusters are renumbered in numerical order in the first figure where they appear.

The assignment of numbers of the clusters in figures 2b,2c, 5a and 7a is not random and were not manually chosen, but derived from the consensus clustering algorithm we used. The numbers assigned for each cluster are based on the consensus clustering algorithm output and determined in an unsupervised manner by the ConsensusClusterPlus function. If we were to manually change the numbers in figure 2b to be in a numerical order, we would need to force changing the figure itself to follow that order. This will consequently result in changing the numerical order of the clusters in figure 2c again and the reader would not be able to match the cluster names with the consensus matrices plots present in Supplementary figures S5, S10 and S12. This is described in the consensus clustering algorithm of the ConsensusClusterPlus package where it makes cluster number decisions based on the purity of members in the clusters {Wilkerson MD; ConsensusClusterPlus: a class discovery tool with confidence assessments and item tracking. Bioinformatics 2010}.

The lack of numerical order in multiple figures of clustering is confusing and makes the paper more difficult to read and understand. This will translate into fewer people understanding the paper and thus fewer citations etc..

If you want to make it easier for the reader, you can change the order of the clusters manually and just transfer that order between figures. It can all be done easily by a bioinformatician in the R-code.

Per the reviewer's request, we have manually changed the order of the clusters and transferred that order between figures. Figure 2, Figure 5 and Figure 7 have been updated accordingly.

Figure 2

Figure 5

Figure 7

a

b

18. The identification and quantification of 4214 proteins across all samples is a good result for MS analysis of FFPE samples. But could some of the results be explained by not reaching deep enough into the proteome, considering that there should be around 14000 proteins in a tissue according to ProteinAtlas. Could this be a reason for the grouping of Luminal A tumors with Her2 in fig 2b? In the Krug et al Cell 2020 paper their tumor grouping almost exclusively only mix luminal A and Bs. No HER2 based on 7679 proteins quantified across all 122 samples. Can the lum A mixing with Her2 be reproduced with the same proteins? Or is this an effect of FFPE? The composition of our 08-13 cohort is different from CPTAC as our cohort included only 11 luminal A cases compared to 73 basal-like, 62 Her2-Enriched, and 28 luminal B. The 4 clusters displayed in Fig 2b were the best to segregate this cohort by consensus clustering and thus with only 11 cases, luminal A tumors were not found as a unique cluster, but grouped with clusters 1 and 4 that included luminal B and Her2-Enriched in Fig 2b. In these clusters 1 and 4, luminal B and Her2-Enriched were often intermixed which is a commonly known phenomenon in breast cancer subtyping (Prat, A. et al. Molecular features and survival outcomes of the intrinsic subtypes within HER2-positive breast cancer. JNCI 2014) and is consistent with the proteomics breast cancer data in (Johansson et al. Nat Comm 2019). The cluster membership of our cohort compared to the CPTAC breast cancer cohort was dependent on a different combination of cases and in turn our analysis of the 86-92 with more luminal A cases was more powered showing distinctions of two subgroups within the luminal A subtype including a unique luminal A “stromal enriched” cluster, and a cluster that was more a mix of luminal A and B. Thus, overall our results are driven by the biology and the composition of our 08-13 cohort rather than an artifact or a technical limitation.

Ok

19. In fig S3a, you refer to biological replicates. How is biological replicates defined in clinical samples? For the technical replicates, it would have been better if they were spread out in different TMT sets.

The biological replicates refer to different specimens taken from the same patients. We acknowledge that technical replicates were in the same TMT set.

We have added the definition of biological replicates to the text on page #7-8:

“High reproducibility was observed between the biological replicates (referring to different specimens taken from the same patient) (mean $r=0.71$) and the technical replicates (mean $r=0.88$) (Supplementary Figs. S4a-S4b)”.

Ok

20. The PAM50 subtypes have got standard color code. See TCGA 2012 Nature or Krug et al Cell 2020. To avoid confusion I strongly recommended to use the same color code.

As per the reviewers' request to make it easier for a reader to compare our results with recent breast cancer 'omic studies we changed the colors to match the color code used in Johansson et al and Krug et al.

21. In general the authors make a good job in describing their findings. But to make it easier to follow I would suggest to add ER, PR and HER2 status to fig 2C. For example in the text it says: “Most cases in Cluster-2 and -3 were associated with ER, PR and Her2 negativity by IHC clinical tests, high proliferation index (Ki67), and the “core basal” phenotype (defined as ER-, PR-, Her2- and [EGFR+ or CK5+])²⁹ (Supplementary Table 1).” Adding the clinicopathological markers to the heatmap in fig 2c would make it easy to see this in addition to the table. But this is a matter of taste and you can ignore if you like.

We thank the reviewer for this suggestion. Supplementary Table 1, Supplementary Table 2, Supplementary Data S1e and the “results” section describe and elaborate on the correlation between these clinicopathological variables and clusters. Figure 2c is already rich in information and different types of analysis and so we feel that the main emphasis for readers should be the PAM50 subtype membership in each proteome cluster.

22. In fig 2c, 5a, there is a column called immune with 2 categories, Immune related and Other. How are they defined? Also, for the protein groups there are enrichments, how were the enrichments done? Specify in fig text how the terms were selected, representative/ cutoff?

Immune related proteins were defined based on their protein function involvement in immuneresponse biological processes. Proteins belonging to any of these gene ontology (GO) categories were labeled as Immune:

"GO_DEFENSE_RESPONSE_TO_VIRUS", "GO_RESPONSE_TO_VIRUS",

"GO_RESPONSE_TO_TYPE_I_INTERFERON",
"GO_CELLULAR_RESPONSE_TO_INTERFERON_GAMMA",
"GO_RESPONSE_TO_INTERFERON_GAMMA",
"GO_REGULATION_OF_INNATE_IMMUNE_RESPONSE",
"GO_CYTOKINE_MEDIATED_SIGNALING_PATHWAY",
"GO_ANTIGEN_RECEPTOR_MEDIATED_SIGNALING_PATHWAY",
"GO_IMMUNE_EFFECTOR_PROCESS",
"GO_ACTIVATION_OF_INNATE_IMMUNE_RESPONSE",
"GO_ANTIGEN_PROCESSING_AND_PRESENTATION_OF_PEPTIDE_ANTIGEN_VIA_MHC_CLASS_I",
"GO_FC_EPSILON_RECEPTOR_SIGNALING_PATHWAY",
"GO_POSITIVE_REGULATION_OF_INNATE_IMMUNE_RESPONSE"

For each protein cluster, the most representative terms were selected based on gprofiler enrichment analysis with the following parameters: organism = "hsapiens", ordered_query = FALSE, multi_query = FALSE, significant = TRUE, exclude_ia = TRUE, measure_underrepresentation = FALSE, evcodes = TRUE, user_threshold = 0.05, correction_method = "g_SCS", domain_scope = "annotated", custom_bg = NULL, numeric_ns = "", sources = NULL, term_size < 150 and source in GO:MF, GO:BP or REACTOME' Raudvere, U., Kolberg, L., Kuzmin, I., Arak, T., Adler, P., Peterson, H., & Vilo, J. (2019). Reference: g:Profiler: a web server for functional enrichment analysis and conversions of gene lists (2019 update). *Nucleic Acids Research*, 47(W1), W191–W198. <https://doi.org/10.1093/nar/gkz369>.

The legends of figures 2c and 5a were updated to include this information.

"Immune related is defined based on the protein function as involved in immune-response biological process and for each protein cluster, the most representative terms displayed on the heatmap were selected based on g:profiler4 enrichment analysis".

The methods section page #32 was updated to include information on the terms selected from the enrichment analysis.

For each protein cluster, the most representative terms were selected and presented on heatmaps based on g:profiler77 enrichment analysis with the following parameters: organism = "hsapiens", ordered_query = FALSE, multi_query = FALSE, significant = TRUE, exclude_ia = TRUE, measure_underrepresentation = FALSE, evcodes = TRUE, user_threshold = 0.05, correction_method = "g_SCS", domain_scope = "annotated", custom_bg = NULL, numeric_ns = "", sources = NULL, term_size < 150 and source in GO:MF, GO:BP or REACTOME'.

Ok, good

23. Fig 2a, is this using all or the most varying proteins? In 2b it does not say that the grouping is based on consensus clustering.

UMAP in Fig 2a is based on using all proteins quantified in every sample (4214). The figure legend has been updated accordingly.

The legend of Fig. 2b has been updated to show that the grouping of the different clusters is based on consensus clustering.

(a) Uniform Manifold Approximation and Projection of the 08-13 cohort for the basal-like, luminal A, luminal B, and Her2-Enriched PAM50 subtypes based on all proteins quantified in every samples (4214).

(b) Alluvial plot shows the relationship between PAM50 subtypes and the four proteomic consensus clusters in the 08-13 cohort.

24. In figure S2b-c, the authors show number of peptides per protein. Bit unclear to what it refers to when mentioning peptide? Is that unique peptides? Nr of peptides per protein, is that the per set or total across all TMT sets or mean/median?

It refers to the total number of peptides identified per protein across all TMT sets. The legends for these figures have been updated accordingly.

(a) Percentage of the total number of proteins detected in different number of samples.

(b and c) Number and percentages of proteins identified according to total number of peptides per protein. Yellow bars in the histogram show the number of proteins identified by different numbers of peptides per protein. Blue dots show the percentage of total proteins identified per minimal number of peptides per protein.

In my version of suppl info it reads: Number and percentages of proteins identified according to number of peptides per protein. Not total. Mean nr across TMT sets would be more informative,

since some sets might have many peptides and some might have few peptides.

We have updated this sentence in the Supplementary Fig. S2b-c legends with the correct definition.

“(b and c) Numbers and percentages of the total number of proteins detected in different number of samples according to number of peptides per protein”.

A measure of the plex variability is given in Supplementary Figure S2d, showing the average number of PSM per plex that closely relates to the average number of peptides.

25. The number of unique peptides per protein, nr of psm per protein and nr of psms/protein for TMT quantification is missing from the supplementary table with all MS data. Please add this, since it is important when it comes to judging the quantitative robustness. Having said that, must give all the credits for clear clinical information and that the authors include it in the same document so it is easy to access!

We thank the reviewer for this point. As also requested in the reviewer's comment #1, we have added the total number of peptides for each protein, number of unique peptides per protein and number of PSMs used for quantification per protein to the Supplementary Data S1c.

26. In figure 3e, the y-axis says abundance. Is this log2 ratio to the pool of samples? ESR1 is high in cluster 2 which is one of the basal enriched clusters, which is surprising. Could this be due to isodoping or poor quantification? KRT18 and FOXA1 on the other hand behave as expected.

Protein abundance shown is based on a log2 ratio for PSM abundances divided by the relative PIS value in each TMT plex. Then for each protein, the median ratio of the 5 most abundant PSMs was used as relative abundance. This is explained in the methods section page #31 and has been added to the legend of Fig. 3e.

“Protein abundance values are based on log2 ratio for PSMs abundances divided by the relative PIS value in each TMT plex. For each protein, the median ratio of the 5 most abundant PSMs was used as relative abundance”.

The abundance for ESR1 was significantly lower in Cluster-3 than the mean against “all” while ESR1 was non-significantly high in Cluster-2. This could be due to challenges in quantifying ESR1 as endogenous peptides for this protein were only detected in less than 10% of the samples. Using isoDoping, 3 isoDoping peptides for ESR1 were detected in the majority of samples and thus challenges in ESR1 quantification might explain the non-significantly higher levels observed for Cluster-2.

What is the justification for limiting ratio calculation to top 5 most abundant PSMs? Should you not obtain a more robust median with more values (if available)? How is abundance in this case defined?

The unexpected behavior of ESR1 and PGR are concerning. How does the IHC data correlate to the proteomics data?

Since the averaged S/N ratio is directly anti-correlated with the coefficient of variation on repeated measurements, we prioritized the PSMs with the highest S/N ratio in an attempt to reduce the quantification's background noise. Abundance is defined as the signal to noise ratio as reported in the methods section page #31.

We added estrogen and progesterone receptor IHC and proteome measurement comparisons in the new Supplementary Fig. S7; explanations have been included in comment #2 and #8.

27. PECA is used for calculating p-values. Wonder if that inflates the p-values and makes them smaller just because you have a lot of peptides per protein?

<https://pubs.acs.org/doi/10.1021/acs.jproteome.5b00363>

PECA method leverages the number of peptides per protein to assign higher confidence to proteins with higher peptide coverage. While we agree that this method tends to drive the p-value of certain proteins with a particularly high number of peptides, we find it useful to separate proteins with a small number of peptides since these are the ones with lower confidence in quantification levels. We directly compared PECA performance to another differential expression algorithm (DEqMS, Zhu, Y., Orre, L. M., Zhou Tran, Y., Mermelekas, G., Johansson, H. J., Malyutina, A., Anders, S., & Lehtiö, J. (2020). DEqMS: A Method for Accurate Variance Estimation in Differential Protein Expression Analysis. *Molecular & Cellular Proteomics*, 19(6), 1047–1057. <https://doi.org/10.1074/mcp.tir119.001646>) on the first differential expression

contrast (Cluster1 vs Cluster2-3-4). We found that the two methods give comparable results in terms of calling differentially expressed (DE) proteins (adjusted p-value < 0.05). We found an overall agreement by DE status on 86% of the proteins: 6% of the proteins differentially expressed in PECA and not in DEqMS, 9% of proteins differentially expressed in DEqMS and not PECA, 11% consistently identified as DE in both methods, and 75% consistently identified as not differentially expressed.

While several differential expression analysis methods are routinely used in the proteomics field and their evaluation over multiple types of data and experiments would be of great interest, we believe that a technical evaluation of PECA and/or comparison with other methods are beyond the scope of this paper.

Ok, this is a point that is good to include in the paper since the general breast cancer biologists reading the paper will not be aware of the inflated p-values and may draw wrong conclusions about the data.

We note that while PECA can achieve improved or comparable overall performance with other differential expression methods [PMID:34373457], it tends to estimate particularly low p-values for proteins with a high number of quantified peptides.

Ref: Kalxdorf, M., Müller, T., Stegle, O. et al. IceR improves proteome coverage and data completeness in global and single-cell proteomics. Nat Commun 12, 4787 (2021). <https://doi.org/10.1038/s41467-021-25077-6>

To add clarity to the readers, we added this point to the methods section page #31:

“While PECA can achieve improved or comparable overall performance with other differential expression methods [34373457], it can obtain very low p-values for proteins with a high number of quantified peptides”.

28. Full credit for uploading the immunohistochemistry slides to <http://www.gpec.ubc.ca/prot>. But why limit to representative images. For the dataset to be useful, all images needs to be available. In addition, there should be an easy way to download all data for image analysis. Our IT team at the Genetic Pathology Evaluation Centre has diligently uploaded all images to <http://www.gpec.ubc.ca/prot>. This information has been updated under the section of “Data availability”, page #35, in the methods.

“Images from immunohistochemistry slides of tissue microarrays used in the study coded as “11-012” and “14-004” are available for public access via the website of Genetic Pathology Evaluation Center (<http://www.gpec.ubc.ca/prot>).

Data image analysis and clinical outcome data for the cases used in this study can be made available through the Genetic Pathology Evaluation Centre and Breast Cancer Outcomes Unit of BC Cancer Centre, upon completion of a Data Transfer Agreement and confirmation of ethical approval for qualified researchers”.

Good

29. To make the data analysis part transparent and reproducible, analysis code should be uploaded to Github or similar repository.

“Code Availability” section has been added to the methods after the “Data Availability” section as requested. Code used for proteomics data analysis is available at GitHub <https://github.com/glnegri/brca>.

Good, but the code seems to only cover basic functions and processing. You need to add the code for consensus clustering and the figures. Also, make sure that the input data is readily available/pointed to.

The available code takes Proteome Discoverer search output data and performs all the steps for filtering and normalization that result in the data used throughout the manuscript. Clustering, differential expression analysis, survival analysis and all other downstream analyses have been performed with publicly available R packages using the parameters described in the methods section. The level of detail provided is sufficient to reproduce the analysis and is in line with several similar articles published recently by Nature Communications:

Liu, W., Xie, L., He, YH. et al. Large-scale and high-resolution mass spectrometry-based proteomics profiling defines molecular subtypes of esophageal cancer for therapeutic targeting. Nat Commun 12, 4961 (2021). <https://doi.org/10.1038/s41467-021-25202-5>

Franciosa, G., Smits, J.G.A., Minuzzo, S. et al. Proteomics of resistance to Notch1 inhibition in acute lymphoblastic leukemia reveals targetable kinase signatures. *Nat Commun* 12, 2507 (2021). <https://doi.org/10.1038/s41467-021-22787-9>

Satpathy, S., Jaehnig, E.J., Krug, K. et al. Microscaled proteogenomic methods for precision oncology. *Nat Commun* 11, 532 (2020). <https://doi.org/10.1038/s41467-020-14381-2>

30. Orbitrap MS2 data was matched with 0.5 Daltons tolerance. This is a very large window that is usually used for iontrap data. For orbitrap MS2, the tolerance should be around 0.02 Dalton to reduce the risk of miss assigning transitions. Since you are also using methylation of lysine as a variable modification, this in combination with a large tolerance will increase your FDR. You should research at least parts of the data and compare the results to your present results to determine if all data needs to be researched. In relation to this, what is the protein FDR of the dataset, q-value, pep value for each protein?

We thank the reviewer for this point. This was actually a typographical error in the text; the data were in fact searched with 0.05 Da tolerance. We have updated the "methods" page #30 accordingly. The full parameters used for the Proteome Discoverer search, together with the results output are available at the PRIDE repository with the dataset identifier PXD024322.

Ok

31. From results: FFPE samples were macro-dissected from 3-6 sections to obtain >80% tumor content and analyzed using the SP3-CTP multiplex MS proteomics protocol²⁴ (Supplementary Fig. S1b).

Should it not be ref 19 instead of 24?

Reference #24 {Hughes, C.S., et al. Single-pot, solid-phase-enhanced sample preparation for proteomics experiments. *Nat Protoc* 14, 68-85 (2019)} is a more detailed and up-to-date protocol for the methods used in this study when compared to Reference #19 {Hughes, C.S., et al. Quantitative Profiling of Single Formalin Fixed Tumour Sections: proteomics for translational research. *Sci Rep* 6, 34949 (2016)}. Given that Reference #19 included work done on FFPE (in ovarian cancer) we now include both references #19 and #24 to support our statement.

Ok

New questions based on the updated manuscript

32. In the discussion the authors write: Furthermore, the 9088 total proteins identified is comparable to that achieved using fresh frozen materials (10,107 proteins) by the CPTAC breast cancer project¹⁵. The 4214 proteins quantified in every sample (n=342) across a large-scale breast cancer project using minimal tissue demonstrates the efficiency and high sensitivity of the SP3-CTP approach for FFPE cancer proteomics studies³⁶⁻³⁹.

The total number of identifications are easily achievable in MS based proteomics. The difficulties are to achieve good overlap in quantification across samples. To make it a fair comparison you should also include the nr of overlapping proteins with quantification. You have also used Johansson et al dataset and should add that also in the comparison.

Given that our study quantified 4214 proteins across a high number (n= 342) of FFPE samples, we did not compare the coverage across all samples to CPTAC and Johansson et al as those studies have a significantly smaller number of cases and are from fresh frozen samples. Given that the reviewer has mentioned that the total number of proteins identified is easily achievable, we have simply removed this sentence from our discussion to avoid any confusion.

33. In the discussion the authors write: Our result is consistent with a proteomic profiling study of 2 "basal immune hot" cases vs. 7 "basal immune cold" cases using >10mg of frozen tissue¹⁶. For max TMT labelling, 100 µg of peptides are used. This usually equates to 1-2 mg wet weight of tissue. Not >10 mg. where in paper 16 did you find this statement?

We thank the reviewer for this point. We reviewed the Johansson et al and we did not find that it used >10 mg. It was actually a mistake in the reference added to this sentence as the CPTAC paper is the one that mentioned the use of 200 mg of material for analysis in their methods:

In Krug 2020 Cell the methods state: "Samples were qualified for the study if two or more tumor tissue core biopsies or surgical resection segments had a minimum mass of 200 mg and demonstrated greater than 60% tumor cell nuclei and less than 20% tumor necrosis on frozen tissue section review".

We decided it is best to remove this sentence from the discussion, as follows (page #19):

“Our result is consistent with a proteomic profiling study of 2 “basal immune hot” cases vs. 7 “basal immune cold” cases using >10mg of frozen tissue¹⁶”.

34. Many of the supplementary figures are a bite blurry, which needs to be fixed.

The Supplementary Figures in the PDF files have been fixed and seem to be clear now. Hopefully, these will look OK to the reviewer when merged in the *Nat. Commun* journal submission system. All figure files used for the final publication will be in the Ai format for high resolution rendering as per *Nat Commun* requirements.

35. Check panel labeling in fig S5. The word robust is mentioned multiple times here and in the paper in general. What do the authors mean by robust? What is the criteria(s) that needs to be meet to be called robust?

The word “robust” has been mentioned multiple times when describing results based on the inspection of consensus matrix and delta plots examining the change in consensus cumulative distribution function (CDF) area to assign the number of consensus clusters to be used in the analysis.

Based on the ConsensusClusterPlus package in R, Consensus Clustering provides quantitative and visual ‘stability’ evidence **derived from repeated subsampling and clustering**. The Consensus Clustering reports a consensus of these repetitions (multiple iterations), which is **robust** relative to the resampling variability. Thus, the word “robust” has been used in this regard.

Nevertheless, to avoid over-use and less specific uses of the term, the word “robust” page #8 has been removed:

“An overview clustering of all the samples included in our study showed that the 38 SuperMix replicates had the highest correlation across the 38 plexes (range 0.68-0.81) when compared to the breast tumors and normal samples. (Supplementary Figs. S4c-S4d)”.

Other “robust” wordings in a context other than the bioinformatic consensus cluster assignments as described above have been removed from the manuscript as follows:

Discussion, page #20:

“Thus, our outcome-linked proteomic data could aid the development of ~~robust~~ protein biomarkers for clinical tests to distinguish TNBC/basal-like patients with favorable versus poor prognosis that may benefit from therapies beyond standard chemotherapies”.

Discussion, page #24:

“The findings on immune distinctions, ECM, and lipid metabolism pathways are potentially clinically relevant as standard clinical tests do not yet interrogate this level of heterogeneity for breast cancer subtyping. Furthermore, this study identifies protein candidates for in-depth analysis of existing archived clinical trial FFPE specimens, providing a valuable resource to develop ~~robust~~ diagnostic and prognostic biomarkers in breast cancer”.

Reviewer #2, expert in bioinformatics and subtype classification (Remarks to the Author):

In this study, the authors carry out mass-spec proteomic profiling of 300 FFPE breast cancer surgical specimens. The specimens are separated into two cohorts based on batch effects. The 08-13 cohort included 75 basal-like, 62 Her2-Enriched, 30 luminal B, and 11 luminal A PAM50 defined cases. The 86-92 cohort provided the long-term outcome data required for luminal cases and included 64 luminal A, 45 luminal B, and 13 Her2-Enriched PAM50 cases. The 08-13 cohort was used for subtype discovery, both across all tumors and within the TNBC subset. ER+ subtypes examined in the 08-13 cohort were examined in the 86-92 cohort.

Specific comments:

1. Batch effects were found between the 08-13 and 86-92 cohorts, likely due to differences in collection techniques, pre-analytical handling, and fixation procedures. Could the authors try to harmonize the two datasets using Combat (<https://rdrr.io/bioc/sva/man/ComBat.html>)? In practice, Combat is very good at removing batch effect differences. Data from different platforms (RPPA, RNA-seq, DNA methylation) have been successfully processed with Combat, and the method is independent of nature of the batch effect. The PAM50 subtype could be used as the experimental group. There would be advantages in having one harmonized dataset of 300 samples. It seems worth a try. As

currently written, the Abstract suggests that there is one dataset that was analyzed, rather than two separate cohorts.

As has been shown before, ComBat can lead to overestimating ratios and many in the field believe should be avoided. (Methods that remove batch effects while retaining group differences may lead to exaggerated confidence in downstream analyses <https://academic.oup.com/biostatistics/article/17/1/29/1744261>), especially considering that the batch effect observed in our study is mostly driven by missed identification of peptides cleaved at lysines and not by artifacts on quantification, as shown in figures S3c and S3d. Furthermore, some of the subtypes are completely (basal-like) or almost completely (luminal A) confounded with the 'cohort' batch effect. While Combat will always transform the data to minimize batch differences, we believe that for the reasons above, its application in this dataset would lead to serious artifacts in the data.

We would also like to note that the decision to include cases from the 86-92 cohort in our study design was based on clinical and translational considerations. In order for analysis to be meaningful for luminal cases, a long enough follow-up was necessary to obtain sufficient events for outcome analyses. Thus, the majority of luminal PAM50 cases were derived from patients diagnosed with invasive breast cancer in the period January 1986 to September 1992. Forcing the two cohorts to be lumped together for subtyping does not allow obtaining clinically-relevant results for the subtypes found, and could compromise any clinical relevant observations.

We have updated the abstract to highlight that for the 300 cases included there were 2 datasets analyzed rather than one.

"We performed comprehensive proteomic profiling of 300 FFPE breast cancer surgical specimens, 75 of each PAM50 subtype, from patients diagnosed in 2008-2013 (n=178) and 1986-1992 (n=122) with linked clinical outcomes".

2. Page 8: "Cluster-1 (n=34) consisted mostly of luminal B and Her2-Enriched PAM50 cases. Clusters-2 (n=50) was enriched for basal-like subtype, included few Her2-Enriched, but had no luminal cases. Cluster-3 (n=47) was primarily basal-like cases but included Her2-Enriched cases. Cluster-4 (n=43) was mostly Her2-Enriched but included luminal A and luminal B cases." It seems that actual numbers to reflect the noted associations would be helpful here, e.g. exactly how many basal-like cases and Her2 cases were in Cluster-3, and was Cluster-2 SIGNIFICANTLY enriched for basal-like.

Cluster-2 is enriched for basal-like (pval<1.16e-11, Fisher's test), Cluster-3 is enriched for basal-like (pval<1.3e-4, Fisher's test), Cluster-4 is enriched for Her2-Enriched (pval<1.9e-4, Fisher's test).

The numbers reflecting the breakdown for each PAM50 subtype within each proteome cluster as they appear in Fig. 2b have also been added to the text, page #9.

"Cluster-1 (n=34) consisted mostly of luminal B (n=18) and Her2-Enriched (n=13) PAM50 cases. Clusters-2 (n=50) was significantly enriched for basal-like subtype (n=41), included few Her2-Enriched, but had no luminal cases (p-value<1.16e-11, Fisher's test). Cluster-3 (n=47) was primarily basal-like cases (n=31) but included Her2-Enriched cases (n=14) (p-value<1.3e-4, Fisher's test). Cluster-4 (n=43) was mostly Her2-Enriched (n=26) but included luminal A (n=8) and luminal B (n=8) cases (p-value<1.9e-4, Fisher's test)".

3. In general, where the word "significantly" appears in the main text, it would be good to include a p-value and associated test to support the claim. The figures referred to likely include the test, but reflecting this in the main text as well would be helpful to the reader. For example, page 11: "The immune hot cluster also had significantly higher CD8+ TILs in the intratumoral compartment compared to other clusters (Fig. 4a)." by what p-value and test?

The p-values and tests are now updated across the text where the word "significantly" appears.

4. Wherever a p-value appears in the main text, the test used to derive that p-value should also be indicated. For example, page 12: "The subgroups with a high expression for only one of these biomarkers were characterized with intermediate RFS (Supplementary Fig. S5b). 70% (21/30) of the cases classified as (TAP1 high/HLA-DQA1 high) were in Cluster-3, while 90% (76/84) of (TAP1 low/HLA-DQA1 low) cases were in other clusters (p-value<0.00001) (Supplementary Table 1)." What test was used here (we can save the reader from having to go the Table for the answer)?

The test used was the Chi-square test. The text in page #13 has been updated to include this information.

"70% (21/30) of the cases classified as (TAP1 high/HLA-DQA1 high) were in Cluster-3, while 90% (76/84) of (TAP1 low/HLA-DQA1 low) cases were in other clusters (Chi-square p-value<0.00001) (Supplementary Table 1)".

5. Page 15: "Multiple correction testing identified fatty acid-binding protein-7 (FABP7) as a candidate biomarker most significantly associated with >10-year RFS on tamoxifen treatment..." Was this the only protein that was significant? Were other proteins significant and using what statistical test and cutoff?

The association between the continuous increase in each individual protein identified in the cohort 86-92 and the endpoint of 10-years RFS was tested using a Cox regression model and stratified log-rank test. This analysis is displayed in Supplementary Data S4f. Only protein biomarkers that had a significant log-rank p-value <0.05 when adjusted for multiplicity testing by the Benjamini-Hochberg test were selected. **Only** FABP7 protein was found to meet these criteria as displayed in Supplementary Data S4f.

The relevant text for the 86-92 analysis page #18 has been updated to include this information.

"Multiple correction testing identified fatty acid-binding protein-7 (FABP7) as the only candidate biomarker associated with >10-year RFS on tamoxifen treatment (log-rank BHadj p=0.00004) (Supplementary Data S4f, Supplementary Fig. S12e)".

6. Discussion, page 16. Many journals are uncomfortable with the phrase "(manuscript in preparation)." It seems that the method indicated should be described in sufficient detail in the Methods, if it isn't already.

We believe that the methods regarding the isoDoping methodology are now described in sufficient detail in the methods section of this manuscript for the reader to be able to reproduce the experiment as was intended. While we are currently preparing an even more detailed and comprehensive description of the general isoDoping strategy for a separate primary methodology-oriented publication, to avoid confusion we have deleted the mention of a "manuscript in preparation." From pages #6 and #18.

7. In addition to making the raw data available on ProteomeXchange, it would be most helpful to include the processed proteinXsample tables as Supplementary Data with the published paper. CPTAC has done a similar thing with their past publications.

The proteinXsample data are included in the original Supplementary Data S1c. As requested by reviewer #1, we have also added the peptides identified across the cohort to the Supplementary data S1 along with the total number of unique peptides per protein and number of PSMs used in quantification per protein (Supplementary Data S1c-S1d).

8. For boxplots in the figures, please define the ranges involved.

Boxplot whiskers range extends to the most extreme data point which is no more than 1.5 times the interquartile range from the box. This definition has been added to the legends of Fig. 3e and Fig. 7b.

Reviewer #3, expert in breast cancer subtypes (Remarks to the Author):

1. The authors present their previously described highly sensitive MS-based methodology termed "Single-Pot, Solid-Phase enhanced, Sample Preparation"-Clinical Tissue Proteomics (SP3-CTP). This technology has been shown to capture known and novel features in FFPE tumor samples. The authors have previously shown that this method can be applied on large FFPE material cohorts linked to outcome data. Comprehensive quantification of protein expression can be achieved even from lower input quantities of patient specimens such as small biopsies. Here is would have been useful to know how small?

This is described in the methods section and supplementary Figure S1a. One to six unstained 10µm tissue sections were cut for each sample to obtain an aggregate total area of ~1cm x 1cm x 10µm, with >80% tumor content.

2. In this paper they have applied the method to 300 well-characterized archival FFPE breast cancer specimens in terms of clinical outcome, IHC, and PAM50 RNA-based intrinsic subtypes. The authors demonstrate that at the protein level one can identify groups characterized by high expression of immune-response proteins and favorable clinical outcomes.

Does this paper bring a sufficient novelty? While it is true that "classifications do not always guide therapeutic choices, due to the extensive heterogeneity that still characterizes breast cancers" can this be solved by adding one more, at the level of proteomics?

As described in the introduction, we performed the current study because genomic classifications of breast cancer are inherently limited as clinical decisions are generally based on the protein level. The underlying technology's application to FFPE breast cancer material is novel. To the extent that some of the findings overlap with genomic classifications, our study still provides an important verification at the protein level, where most drugs act.

Q1. How do this extension to 300 cases add to what we know from Johansson et al Nat Comm, 2019?

As highlighted in the introduction, Johansson et al. *Nat Comm* 2019 only profiled 9 tumor samples from each of the four main breast cancer PAM50 subtypes, a set which also lacked clinical outcome associations and was insufficient

to characterize the biological heterogeneity of breast cancers in relation to clinical behavior and treatment response. In addition, their work required fresh-frozen tissues that are not routinely available from patients, unlike the FFPE clinical specimens we were able to use that can be accessed in larger numbers allowing meaningfully powered linkages to clinical outcomes.

Q2. How does the heterogeneity described here match what is known from RNA based classification (basal also divided in several immune clusters)

The PAM50 subtypes used in this study are an RNA-based classification and the associations of each proteome cluster membership with each PAM50 subtype are described in detail in the manuscript. Within the basal-like RNA-based subtype, there are two distinct proteomic groups that differ in immune response. In the results section, we describe how the heterogeneity of triple negative breast cancer relates to what is known from RNA-based classifications by comparing our findings with those by Burstein M et al. *CCR* 2015, showing that our triple negative clusters were highly correlated with their corresponding RNA subtypes of 'luminal-androgen receptor', 'mesenchymal', 'basal-immune suppressed' and 'basal-immune activated'.

Q3. If the authors were to make biomarkers based on protein as they suggest, which ones would they chose?

TAP1 and HLA-DQA1, as described in detail in the results and discussion sections. These choices are further supported by the supplementary validation work done in response to reviewer #1, comment #8 as described above (based on the data shown in Supplementary Figures S7 and the new figure S8). We do note that TAP1 and HLA-DQA1 were chosen, in part, because of the availability of quality IHC grade antibodies; it remains possible that other proteins may perform better on IHC-based tests when quality antibodies are available. Indeed, this is one of the prime utilities of our results for the breast cancer community, to spur additional biomarker research using our data.

The discussion page #21 has been updated with this information.

"Other proteins elevated in the immune hot cluster with available quality antibodies could also be used and developed as candidate biomarkers".

Q4. An introduction of 5 pages and large number of references (81) makes it into a difficult read. This paper as rigorously performed and described, would benefit from some clarity and simplification, just highlighting the results that move the field forward.

The original work was written in a way that fits the requirements of *Nature Communications*. The introduction here is 2.5 pages double spaced rather than 5 pages as pointed out by the reviewer and the authors hold that this is adequate to succinctly review the pertinent literature, making it hard to remove any essential information from the introduction. As this research sits at a crossroads of breast cancer, bioinformatics, and analytical chemistry the authors believe it is important to provide key background information for scientists from a breadth of related and interested fields to fully appreciate the work. 84 references are merely supporting information for the interested reader to pursue, a number that complies with the *Nature Communications* guidelines (and we are aware of several detailed and comprehensive publications in *Nature Communications* that have a similar or even higher number of references used to properly cover the scientific data presented).

Reviewer #1:

Remarks to the Author:

NCOMMS-21-10792, Response to reviews

RESPONSE TO REVIEWER COMMENTS

Reviewer #1, expert in proteomics (Remarks to the Author):

Asleh et al have performed quantitative proteomics of 300 breast tumors from formalin fixed paraffin embedded (FFPE) material. The general idea and its potential value to the community of this work is great.

The main Merits of the paper are: 1) the acquisition of proteomics data from FFPE samples across a large number of breast cancer samples with clinical follow up that can serve as a resource, 2) directly linking protein based sample groups with immune infiltration to improved outcome, 3) suggestion of potential biomarkers for tumor groups 4) identification of 4 TNBC groups as previously suggested at the RNA level and linking the immune infiltrated subgroup to good outcome, 5) identification of 3 ER positive tumor groups with a stromal enriched group.

We appreciate the reviewer's view that the work will provide important value to both the fields of breast cancer and of proteomic analysis of patient samples in general.

Limitations regarding merits above:

1. To function as a resource the data needs to be judged as robust.

To evaluate protein quantitative robustness, the number of peptides used for quantification per protein needs to be available and visualized. Now it is lacking from the supplementary data table with all ratios. A panel can also be added to figure 2 to show nr of psms/protein used for quantification.

We thank the reviewer for this point. We have added the total number of peptides for each protein, number of unique peptides per protein and number of PSMs used in quantification per protein to the Supplementary Data S1c. We have also added the data on the peptide abundance per protein (now appears as new Supplementary Data S1d) and PSMs per protein in Supplementary Figure S2d.

Good! However, you need to fix the x-axis. Now it reads:

Fig text: Average number of quantified PSMs per protein, across the full cohort – is that for the subset with quantification across all or including all proteins?

In suppl data S1C the column header says: set_1_number_PSMs – that is nr of psms used for quantification I presume? When you add this information, it would also be informative to add the nr of unique peptides/protein per set. Also, protein scores and q-values are missing from the table. Add a column to easily select the proteins that you have used in your data analysis.

The x axis relates to the PSM data from the 4214 proteins quantified across all samples rather than all 9088 proteins quantified in total. We have updated the text of Figure S2d legends accordingly.

“(d) Average number of quantified PSMs per protein, across the full cohort (corresponding to the 4214 quantified across all samples)”.

Supplementary Data S1c shows the number of PSM per protein after filtering as described in the methods section, which were then used for quantification.

Per the reviewer's request, we added additional columns to Supplementary Data S1c with the PSMs #/protein per set, protein scores per set and q-values per set. In addition, we added a column to quickly identify the proteins used in the analysis (TRUE vs. FALSE).

Good!

Supplementary Figure S2

2. The supermix is present in all TMT sets and should represent how well quantifications can be reproduced between TMT sets. Can the supermix data be used for robustness evaluation between the sets? For example a heatmap for overview, variation of supermix in relation to the breast samples and particular sets with deviation on supermix-sample.

We thank the reviewer for this point. The 38 SuperMix replicates included in our experiment showed a high high correlation across the 38 plexes. Unsupervised clustering of our data for all samples including breast tumors, normals, and SuperMix show the SuperMix samples clustered together and are clearly separated from the breast tumor and normal samples. The correlation between the SuperMix samples was the highest when compared to the breast tumor and normal samples, supporting the robustness of the evaluation of SuperMix samples across the sets (appears as new Supplementary Fig. S4c). Pairwise correlation between the 38 SuperMix replicates (ranged between 0.68-0.81, median 0.75) was significantly higher than the pairwise correlation across the 38 normals (ranged between 0.53- 0.85, median 0.71). These findings are shown in new Supplementary Fig. S4d.

The following information has been added to the results section page #8:

“An overview clustering of all the samples included in our study showed that the 38 SuperMix replicates had the highest correlation across the 38 plexes (range 0.68-0.81) when compared to the breast tumors and normal samples (Supplementary Figs. S4c-S4d).

The small difference in correlation between the Supermix, that should be exactly the same sample in all TMT sets, and the normal samples, which are biologically different are surprising. The supermix should represent technical variation and in this case are very close to the biological variation. The large number of proteins used in the sample to sample correlation analysis will provide a relatively high correlation, which limits this analysis.

To be able to support the claim of the dataset as resource, the reader needs to be able to better understand the technical variation in the dataset. For example, you could calculate coefficient of variation for each protein based on the supermix and plot that.

Also, you have IHC data for some proteins as ESR1, PGR etc, how these measurements correlate to the proteome data would be useful for judging the quality of the data.

Indeed, as shown in the plot below, we expected a tighter correlation between SuperMix replicates since they should represent the technical variation across TMT plexes. However, we found that the SuperMix shows an average higher variation across the cohort compared to the one observed in the (biologically distinct) normal samples. We believe that this increase in variation is the result of the very different background matrix composition of the samples as the SuperMix includes 13 different cancer models cultured in vitro (as described in the methods) while the rest of the cohort consists of breast tissue, which was FFPE preserved. This under-representation (1/11th of the channels) of the SuperMix matrix makes it more likely to be affected from isolation interference and background noise leak from the breast tissue FFPE samples. For these reasons, while the SuperMix is an important reference standard that allows future comparisons with any cohorts that will include a SuperMix control in the design, it doesn't completely reflect the true technical variation in this cohort. However, a better representation of the technical variation can be estimated from the technical (n= 3) and biological (n= 3) tumor replicates that we included as part of the cohort (Supplementary Fig. S4a-b).

Supplementary Figure S4

We include the comparison of ER, PR and HER2 IHC results with the proteome data in Supplementary Fig. S7 along with the validated IHC markers mentioned in the reviewer's comment #8.

Supplementary Figure S7

a

b

Supplementary Fig. S7. Correlation between proteomic abundance scores vs. IHC for selected proteins.
(a) Relative abundance of ESR, PGR and HER2 by Mass spectrometry according to their IHC categories.
(b) Correlation of protein expression values for protein candidates by mass spectrometry vs. IHC. Scoring values of the S100A8, TAP1, IFIT2 and HLA-DQA1 IHC biomarkers were reported using the H scoring system (intensity x

positivity) for the cytoplasmic staining observed in the invasive breast tumor cells. Spearman correlations are shown on each panel. Abbreviations: IHC, immunohistochemistry.

The categories of ER, PR and Her2 are assigned per the available pathological data extracted from the patients' charts reporting hormone receptor status and HER2 as positive vs. negative. A highly significant association was observed between the proteomic relative abundance of HER2 and clinical HER2 status. A significant association between the proteomic relative abundance of ESR1 and ER IHC status was also observed. PGR relative abundance was overall higher in PR+ by IHC, but this result was not significant.

In addition, the results page #12 were updated accordingly to include these results:

"When testing the association between the MS data for ESR1, PGR, HER2 and their IHC categories, results were significant for HER2 ($p < 0.0001$) and ER ($p = 0.02$) IHC expression (Supplementary Fig. S7)".

As is also explained in detail in our response to comment #8, there are several reasons why different IHC biomarkers could differ in their association with the proteomic data. ER and PR assessment were performed per the current established guidelines that evaluate their **nuclear staining on carcinoma cells only, using pre-established clinically validated cutpoints to report results categorically as positive vs. negative**. In contrast, the MS relative abundance does not consider **this spatial information** when reporting the overall protein scores. The inference of the protein level in MS is based on peptide level quantification, while IHC is semi-quantitative with the inherent limitations of being an antibody-based assay with analytical and preanalytical issues that can affect the results.

Ok, Good!

I would be a bit cautious to use the supermix to compare between TMT sets in the future since it shows higher CVs than the rest of the conditions (comment not related to this manuscript).

3. An overview clustering of the 2 cohorts with replicates would also be useful to judge how the whole dataset behaves. Does the technical replicates cluster together?

Per the reviewer's request, we generated a heatmap showing the overview clustering for all the samples, as also requested in the previous comment. As now shown in Supplementary Fig. S4c, the three technical replicates indeed have clustered adjacent to each other (T_rep 5, T_rep 6, T_rep 7). Regarding the biological replicates, 2 of 3 replicates clustered adjacent to each other while the 3rd biological replicates clustered very closely together, a variance in line with expectations for intratumoral regional sampling. The normal samples clearly separated from tumor samples and showed an overall correlation of 0.70. An overall correlation of 0.5-0.6 was observed for the different breast tumor clusters and these included a mix of samples from both 08-13 and 86-92 cohorts.

This information has been added to the results section page #8:

"All the technical replicates and 2 out of 3 biological replicates clustered adjacent to each other, while the 3rd biological replicates clustered very closely together, a variance in line with expectations for intra-tumoral regional sampling (Supplementary Fig. S4c).

Ok

4. The data should also confirm with previous knowledge, as ER, PR, HER2, MKI67 levels in different PAM50 subtypes, and this would be good to show in a supplementary figure.

HER2 (ERBB2) and MKI67 expression levels across the different PAM50 subtypes are found in Supplementary Fig. S6d. ER (ESR1) and PR (PGR) expression levels across the different PAM50 subtypes are now also included in Supplementary Fig. S6d.

Ok, see my comment to question 2.

We include the comparison of ER, PR and HER2 IHC results with the proteome data as explained in the above response to comment #2.

Ok!

5. Proteomics have previously identified immune infiltration in breast cancer subgroups without directly linking them to outcome (Krug 2020 Cell, Johansson et al 2019 Nat Comm). Tumorinfiltrating lymphocytes (TILs) have also been linked to better outcome in breast cancer subtypes (Dieci 2021 Cells). The strength of this study is the direct link between proteomics data with "immune hot" tumors and outcome.

Relation to published data

In general, anchoring the novel findings further, e.g. by validation of findings in other breast proteomics data sets would be valuable to show the usefulness of the data as a resource and strengthen the findings. There is several decent datasets published now on breast cancer proteome so this should be done.

Per the reviewer's request, we performed a validation of our findings on previous proteomic datasets published by Krug et al. Cell 2020 (CPTAC) and Johansson et al. Nat. Commun 2019 (OSLO2).

Validation using the Krug et al. 2020 CPTAC breast tumor cohort: In order to compare our results with available published datasets, we performed consensus clustering with the same parameters used in our cohort on the CPTAC Cell 2020 cohort, using the 939 proteins from the CPTAC data that overlap with the 1054 mostly highly-variant proteins of our 08-13 cohort. This analysis identified four main proteome clusters that highly resembled the original CPTAC NMF clusters of "LumA-I", "LumB-I", "Basal-I", "HER2-I". Two of these were almost entirely similar to the original NMF clusters of "Basal-I", and "LumA-I". Another cluster highly resembled NMF "LumB-I" and consistent with Krug et al consisted of 54% luminal A cases (compared to 55% luminal A cases assigned as "LumB-I" in the original NMF CPTAC clusters by Krug et al). Similar to the original NMF CPTAC clustering composition, the NMF CPTAC "HER2-I" cluster identified had a mix of Her2-Enriched, luminal A and luminal B breast cancers. Of note, the original Krug et al 2020 study of 122 breast tumors included a majority of luminal A PAM50 subtype (n=57, 47%), followed by basal-like (n=29, 24%), luminal B (n=17, 14%) and Her2-Enriched (n=13, 11%) when compared to the composition of our 08-13 cohort which consisted of a higher number of basal-like (n=73, 42%) and Her2-Enriched (n=62, 36%) cases, but few luminal A (n=11, 6%). Despite this, our analysis further reproduced the existence of subsets enriched for immune response pathways at the proteome level within the basal-like and Her2-Enriched subtypes not captured in the CPTAC analysis. Consistent with our analysis on the 08-13 cohort, stromal pathways were enriched in luminal A tumors and lipid metabolism was enriched within luminal B and Her2-Enriched tumors. A description of these findings is displayed in Supplementary Fig. S10a.

In the results section you write: Our analysis reproduced the existence of subsets enriched for immune response pathways at the... These subsets are within your clusters. They don't come out as defined clusters. You need to make that clear. It looks though as it should be possible to separate out immune enriched samples.

We agree with the reviewer. Our analysis of the CPTAC breast tumor cohort did not demonstrate these as separate defined clusters, though it seemed possible to separate out some immune enriched samples that were classified as basal-like and Her2-Enriched. In contrast, the analysis of our 08-13 cohort revealed an "immune hot" cluster that was referred to as a defined and distinct cluster. These differences might be because of the reasons explained above in our original response regarding the composition of our 08-13 cohort, which includes a much higher number of basal-like and Her2-Enriched cases when compared to the CPTAC cohort.

Overall, our analysis on the CPTAC cohort illustrates that there is a fraction (subset) within the basal-like and the Her2-Enriched subtypes that are enriched for immune response pathways. For clarity, we have replaced the word "reproduced" with "demonstrated" in the sentence mentioned in the reviewer's comment and updated this sentence in the results section page #14 and in the legend of Supplementary Figure S10a as below, highlighting that these were not captured as defined clusters in the CPTAC analysis.

"Our analysis demonstrated the existence of subsets enriched for immune response pathways at the proteome level and these included basal-like and Her2-Enriched subtypes. In contrast to the 08-13 cohort, these subsets were not captured as separate and defined clusters by CPTAC analysis".

Ok, good!

Validation using the Johansson et al 2019 "OSLO2 breast cancer landscape cohort":

To validate our findings on the 36 cases of the 4 main subtypes (9 for each PAM50 type) in the "OSLO2 landscape cohort", we performed consensus clustering with the same parameters used in our analysis, using the 775 proteins from the OSLO2 data that overlap with the 1054 mostly highly-variant proteins of our 08-13 cohort. This analysis identified 4 clusters that highly resembled the main consensus core tumor clusters (CoTCs) and their biological functions as reported in Johansson et al. These clusters consisted of CoTC1 (basal-like immune cold), CoTC2 (basal-like immune hot), CoTC3 with few CoTC6 cases (luminal A-enriched) and CoTC6 (luminal B and Her2-Enriched). Importantly, the immune distinctions within the basal-like

subtype were entirely reproduced using our highly variant proteins showing that the two basal-like samples of OSL.3EB and OSL.449 (CoTC2) were consistently classified as “basal immune hot cluster” when compared to other basal cases characterized as “basal immune cold”. These findings are displayed in Supplementary Fig. S10b.

The results section page #14 has been updated to include our comparison analysis using the Krug et al 2020 and Johansson et al 2019 proteomics datasets, as a new section entitled “Comparison with previous breast cancer proteomics studies”.

The number of immune hot samples are a little bite low, but in the other hand supports your findings.

Indeed. We agree with the reviewer that the number used in Johansson et al. is extremely low when compared to our dataset and we highlight that in the comparison we make in discussion section pages #19-20. To date, the only proteomic published data preceding our current study which showed the existence of defined immune hot vs. immune cold clusters consisting of basal-like cases is Johansson et al, and thus despite its limitations serves as the best available proteomic dataset for comparison. It does support our findings as highlighted in the introduction page #4 and the results section page #15.

Ok, good!

Supplementary Figure S10

a) Validation using the Krug et al. 2020 CPTAC breast tumor cohort

b) Validation using the Johansson et al 2019 OSLO2 breast cancer landscape cohort

6. The 4 TNBC groups are correlating to their suggested RNA based groups. To strengthen the finding of 4 TNBC subtypes, can they be identified also at the protein level, for example in Krug 2020 Cell data?

We validated our TNBC proteome clusters using the 935 proteins from Krug et al that overlap with the 1055 mostly highly-variant proteins in our 08-13 TNBC (n=88) subset on the set of 28 TNBC cases included in the CPTAC breast cancer cohort by Krug et al. Our analysis reproduced the existence of the four main proteome TNBC subgroups and the biological features of 'luminal-androgen receptor', 'mesenchymal', 'basal-immune suppressed', and 'basal-immune activated' as now shown in Supplementary Fig. S12.

The results section page #16 has been updated to include this information:

“The existence of these TNBC proteome clusters and their biological features were validated when applying consensus clustering, with identical parameters, on the 935 proteins overlapping with the 1055 mostly highly-variant proteins of our 08-13 TNBC subset on the proteomic data for a set of 28 TNBC cases included in the CPTAC breast cancer cohort by Krug et al (Supplementary Fig. S12).

Supplementary Figure S12

Ok, good!

7. How generalizable are the 3 ER positive tumor subgroups identified in the manuscript? The authors cite Krug 2020 Cell in the discussion as consistent with the stromal-enriched subtype. But to my knowledge, the data in the Krug paper don't show a separate luminal A subgroup enriched for stroma. Dennison 2016 CCR, however show a stromal subtype of ER positive tumors that are or mixed subtype but enriched in Luminal A with a favorable clinical outcome. Are the same proteins (in RPPA and your MS data) deterministic of the stromal subgroup?

We agree with the reviewer that Krug 2020 Cell did not identify a separate stromal enriched subtype as a unique cluster by mass spectrometry, but described a subset of luminal A tumors as stromal-enriched since these tumors were classified originally as “reactive” in the TCGA 2012 RPPA data. In the subsequent Nature 2016 CPTAC proteomics profiling breast cancer publication, the proteomic cluster that was highly correlated with the “reactive” RPPA cluster was referred to as stromal-enriched.

The Dennison 2016 CCR study basically tried to characterize the biological and clinical features of the

stromal enriched tumors as a whole (i.e. reactive tumors) identified in the TCGA based on RPPA data. The majority of these tumors were found to be classified as luminal A by PAM50, and among the luminal A as a group those that had high stromal protein expression displayed favorable clinical outcomes.

Comparing the proteins in our MS data that are in common with the RPPA proteins (n=30) used to classify the “stromal-enriched” vs. the “ER positive cancer derived” subtypes in Dennison 2016 CCR, we found 5 proteins in the RPPA “stromal-enriched” Dennison 2016 CCR that were also characteristic for our luminal A stromal enriched proteomics cluster ($\log_2FC > 0.20$, adjusted p-value < 0.05). These were fibronectin, annexin, collagen VI, caveolin, and MYH11.

While our data correlate with those results, the RPPA data only cover a small percentage of the proteome that was quantified in our experiment; thus, our data characterize the luminal A stromal enriched cluster in a more comprehensive manner and identify protein candidates that are beyond those captured by the restricted number of proteins in the antibody-based RPPA assay.

The discussion page #23 has been updated to highlight this information.

“Our analysis of ER+ cases with mature clinical data identified a stromal-enriched subset (86-92-Cluster-2) consistent with previous reports^{57,63}, which could help sub-classify luminal breast cancer. However, our data characterize the luminal A stromal enriched cluster in a more comprehensive manner and identify protein candidates that are beyond those captured by the restricted number of proteins in the antibody-based RPPA assay”.

OK

8. IHC validation of S100A8, TAP1, IFIT2, HLA-DQA1 and CD8 as suggested biomarkers of immune infiltration and better outcome are done on the same cohort as the proteomics. To consolidate the findings, validation in an independent cohort would be valuable. Also, what is correlation between the MS data and the IHC validated markers? Are the MS protein levels also related to outcome?

First part of the reviewer’s comment: Per the reviewer’s request, we have now performed a validation of these IHC biomarkers on an independent set of 176 breast cancer cases with similar clinicopathological characteristics to the 08-13 cohort. Our analysis confirmed that high expression of HLA-DQA1 as a single biomarker had a significantly better survival (log-rank $p=0.02$) and a similar trend was seen with high TAP1 as a single biomarker (log-rank $p=0.09$). The findings further confirmed that tumors with IHC expression for both TAP1 and HLA-DQA1 showed the most favorable survival, while the subgroup with low expression for both had the worst RFS (log-rank $p=0.05$) (Supplementary Fig. S9).

Supplementary Figure S9

The results section page #14 has been updated with this information:
 “We subsequently confirmed our observations on an independent, clinically similar set of 176 breast cancer cases and showed that high expression of HLA-DQA1 as a single biomarker

had a significantly better survival (log-rank $p=0.02$) and a trend was seen for high TAP1 as a single biomarker (log-rank $p=0.09$). These data also confirmed that tumors with high IHC expression for both TAP1 and HLA-DQA1 showed the most favorable survival, while the subgroup with low expression for both had the worst RFS (log-rank $p=0.05$) (Supplementary Table 3; Supplementary Fig. S9).

The Supplementary methods in the Supplementary Information file page #27 and Supplementary Table 3 include information on the characteristics of this IHC validation cohort:

“IHC validation cohort: A tissue microarray for an independent set of 176 breast cancer cases was used to validate observations on the 08-13 cohort for the key protein IHC biomarkers. This validation cohort had clinicopathological characteristics similar to the 08-13 cohort and was analyzed for IHC biomarker association with clinical outcomes. The median follow-up for the IHC validation cohort was 10 years and cases were treated in accordance with contemporary guidelines”. Characteristics of this cohort appear in the new updated Supplementary Table 3.

Supplementary Table 3

Characteristic	IHC Validation cohort (n=176)
Age at diagnosis (median)	53 years
Tumor size (median)	2 cm
Tumor grade	
1, 2	44 (25%)
3	127 (72%)
Missing	5 (3%)
Nodal status	
Negative	105 (60%)
Positive	66 (37%)
Missing	5 (3%)
IHC subtype	
Luminal ([ER+ or PR+])	69 (39%)
ER-, PR-, HER2+	32 (18%)
ER-, PR-, HER2-	71 (40%)
Missing	4 (3%)
Disease specific death	
No	134 (76%)
Yes	35 (20%)
Missing	7 (4%)
CD8 iTILs	
<1%	42 (24%)
≥1%	129 (73%)
Missing	5 (3%)
TAP1/HLA-DQA1 IHC groups	
TAP1 high /HLA-DQA1 high	35 (20%)
TAP1 low /HLA-DQA1 high	22 (13%)
TAP1 high /HLA-DQA1 low	50 (28%)
TAP1 low /HLA-DQA1 low	65 (37%)
Missing	4 (2%)

Good!

Second part of the reviewer’s comment: The Spearman correlation between the MS data and the H score for the IHC validated markers was found to be 0.51 for TAP1 and S100A8, 0.31 for HLA-DQA1, and 0.11 for IFIT2 as shown in the figure below. Of note, the assessment of the validated markers by IHC was performed on the carcinoma cells.

This data should also be in the paper together with the same kind of analysis for ESR1 and PGR. Why do you think the correlations are weak? For TAP1 and HLA-DQA1 that performs well together, what is the difference in signal that is picked up by IHC and MS? Both are prognostic but show weak correlations indicating different signal/information that they pick up.

Per the reviewer's request, this analysis has been added as a new Supplementary Figure S7.

The analysis according to the IHC ER score with ESR1 proteomic abundance and the IHC PR score with PGR proteomic abundance is described in comment #2 and included in the manuscript under new Supplementary Figure S7. In addition, the results page #13 were updated accordingly:

"When assessing the correlation between the MS data and the IHC scores for the validated biomarkers, a low-moderate correlation was noted (Supplementary Fig. S7)".

Regarding the weak-moderate correlation between IHC and MS data, there are several explanations. Firstly, as explained in the methods section, the assessment of the validated markers by IHC was performed following practical and established IHC methodologies to assess their expression **only** on the invasive carcinoma cells and using the H score that in addition to **positivity** also takes into account the **intensity** when reporting the IHC expression. These 2 components of positivity x intensity are multiplied to give the overall score. Importantly, for these biomarkers scores were reported for the **cytoplasmic staining only** that was observed in the invasive breast tumor cells, using a tissue microarray format with duplicate cores for each specimen. Intensity scores were reported as (0: none, 1: weak, 2: moderate, 3: strong) and the positivity proportion scores were reported as (1-100%) for each core. The averaged cytoplasmic H score between the duplicate cores per case was used for the scoring of the protein expression by IHC. Secondly, when analyzing tumor specimens by MS, the whole section is analyzed and the expression of specific proteins is not measured in the context of spatial expression on invasive carcinoma cells only and considering appropriate subcellular (cytoplasmic) expression only. Furthermore, the representative cores assessed on tissue microarray do not always represent the expression on the whole slide taken from the source block, but rather represent the expression of the relevant biomarker specifically in the most histologically-representative viable invasive carcinoma areas punched out as cores to construct these tissue microarrays. Thirdly, there are several analytical and preanalytical differences related to IHC as an antibody-based assay vs. MS that contribute to the correlations observed with these biomarkers. IHC is semi-quantitative due to the fact that it is antibody hybridization-based (with the signal amplified using secondary antibodies and linked enzymatic chromogen activation) while the inference of the protein level in MS is based on the peptide level quantification that is more quantitative than IHC. Altogether, these are reasons why while MS-IHC data would be expected to show weak-moderate correlation, they could still both be prognostic.

These reasons were briefly summarized and included in the discussion page #21.

"Of note, the assessment of the validated markers by IHC was performed only on the carcinoma cells and using the H score that in addition to positivity, takes into account staining intensity when reporting the IHC expression. These variables along with using a TMA format and the differences related to IHC as a multi-step antibody-based assay vs. MS contribute to the weak-moderate correlations observed with these biomarkers".

Ok,

Third part of the reviewer's comment: The selection of the biomarkers for IHC validation was based on biology rather than clinical outcomes. In response to the reviewer's comment, we performed a Cox proportional-hazards analysis on the protein abundance (in MS data) and recurrence free survival for the protein candidates we assessed by IHC. MS protein levels are significantly correlated with improved outcome for TAP1 and IFIT2, while a trend is shown for HLA-DQA1 and S100A8 as follows:

Protein	Survival analysis for RFS HR (95% CI), P -value	Adjusted P -value
TAP1	0.34 (0.18-0.65), 0.001	0.04
HLA-DQA1	0.87 (0.69-1.10), 0.24	0.71
S100A8	0.87 (0.73-1.06), 0.16	0.62
IFIT2	0.38 (0.18-0.80), 0.01	0.19

Additional comments:

9. From introduction: “This method can query large FFPE material cohorts linked to outcome data, enabling comprehensive quantification of protein expression from lower input quantities of routinely-available patient specimens, and employs a more highly efficient workflow than other MS-based methods for protein profiling of clinical FFPE tissues^{21,22}. “

Based on the data, the MS workflow seems efficient, but there is really no data to comparing all other methods to support your claim of “more highly efficient workflow than other MS-based methods...”? Many of the large MS proteomics groups have published their versions of FFPE sample preparation methods. See for example Coscia 2020 Modern Pathology, Griesser 2020 MCP, Marchione 2020 JPR, Zhu 2019 Molecular Oncology.

We thank the reviewer for this comment. We have updated this sentence in the introduction page #5 accordingly:

“This method can be used to query large FFPE material cohorts linked to outcome data, enabling comprehensive quantification of protein expression from lower input quantities of routinely-available patient specimens, and employing a more highly efficient workflow than other MS-based methods for protein profiling of clinical FFPE tissues^{21,22} .

Ok

10. In the abstract and in figure one, 300 samples are mentioned as included in the study. The number is correct but it is bit misleading since it's divided up in 2 cohorts. The overview in Figure 1A is not useful since this collection of samples are not used together later on in the paper. The overview presented in fig S1A are much more useful since it gives an overview of the samples used together in each of the later analyses. Also the number of samples drop after QC and removal of replicates. To make it clearer for the reader I suggest you make a combination figure of fig S1A and S2H with the tumor characteristics and the numbers that make up each cohort used in the downstream analysis. Also include the info of how the TNBC cohort was made. This took time to figure out and with a figure outlining the 2 cohorts, it would be much clearer from the beginning for the readability of the entire paper. To make it even clearer one could add what type of analysis / aim you have with each cohort. There is also normal samples for which it is unclear of their purpose/how they are used. Did not find any comparison to the normal samples in the text?

Per the reviewer's recommendation we have moved the original Figures S1A and S2H to Figure 1. Now they appear as Fig. 1b and Fig. 1c.

Given that normals were sourced from independent reduction mammoplasties, they are very biologically different from tumors and thus they are not helpful in the subtyping or performing direct comparisons with tumor samples. The normals were included in the UMAP plots where they form a clearly separated cluster from tumors, added to the heatmaps (Figures 2c and 5a) as a reference to illustrate that proteins and pathways of interest for the proteome clusters were not high in normals, and as a visual comparator for the expression of key breast cancer associated proteins in Supplementary Figure S6d.

In addition, when we picked specific proteins of interest for validation in IHC, we used candidates that were not highly expressed in normals. We updated the text to include this specific information in page #12.

“We selected four that were among the top differentially-expressed proteins between the immune hot cluster vs. others (Supplementary Data S2c), had available antibodies applicable to FFPE, and had a practical scoring methodology on carcinoma cells: TAP1 (MHC class I), HLA-DQA1 (MHC class II), IFIT2 (type I interferon signaling) and S100A8 by IHC (Figs. 4b-4c). In addition, these proteins were not highly expressed in the normal reduction mammoplasty samples”.

The authors have gone some way to make the paper clearer when it comes to the patient cohorts. However, the results section starts with: A cohort of 300 archival FFPE breast tumor primary tissues,.... All the samples are never used together as a cohort. So this sentence and fig 1A, B are

misleading and need to be changed. You need to make it clear in the figure texts and abstract that you are analyzing 2 different cohorts, not one with 300 samples.

We thank the reviewer for this point. The design of this study was to include 300 total samples such that in sum they would represent 75 samples from each 4 main PAM50 subtype. Among the total 300 samples, luminals were mostly collected from an older cohort, so as to allow meaningful clinical outcomes that can only be captured by using long-term follow-up for these clinically less-aggressive cases, as described in the first paragraph of the results section page #6. The original intent of the study design was to analyze the 300 samples as a single cohort and thus a mix of cases from “08-13” and “86-92” were spread across the 38 11-plexes when we ran the study. Thus, the MS data were collected as a single cohort design. However, as explained in detail in page #7, due to the batch effects observed we analyzed the total 300 cases as two separate cohorts.

Per the reviewer’s request, the paragraph in page #6 has been updated accordingly:

A total of 300 archival FFPE breast tumor primary tissues, representing 75 from each of the RNA PAM50 subtypes⁴, and 38 normal reduction mammoplasty samples, were obtained (Fig. 1a-1b). Samples were assembled with an original aim to be analyzed as one cohort, thus the MS data were obtained per this design, from patients diagnosed with invasive breast cancer using tissue obtained prior to adjuvant systemic therapy in 2008-2013 (n=178; the 08-13 cohort) and 1986-1992 (n=122; the 86-92-cohort). The 08-13 cohort included 75 basal-like, 62 Her2-Enriched, 30 luminal B, and 11 luminal A PAM50 defined cases. The 86-92 cohort provided the long-term outcome data required to gather sufficient outcome events for luminal A breast cancers and included 64 luminal A, 45 luminal B, and 13 Her2-Enriched PAM50 cases (Fig. 1b).

Figure 1b shows the breakdown of the two cohorts included according to “time of collection” to indicate the difference between 08-13 vs. 86-92 cohorts that were analyzed separately. The word “cohort” and the description of the cohorts has been added to the x axis in Figure 1b for further clarity. In addition, per the reviewer’s request, the legend for Figure 1 has been updated:

Figure 1. Proteomic analysis of FFPE breast cancer tissue samples

- (a) *The clinical features of the 300-tumor study cohort across the four PAM50 breast cancer subtypes. Samples were assembled from patients diagnosed with invasive breast cancer using tissue obtained prior to adjuvant systemic therapy in 2008-2013 (n=178; the 08-13 cohort) and 1986-1992 (n=122; the 86-92-cohort). While the MS data were obtained with the 08-13 and 86-92 samples intermixed (see Fig S1b batch design), these two cohorts were analyzed separately. Pathological primary tumor size is defined as (T1 <=2cm), (T2 2-5cm), (T3 >5cm); recurrence, (local, regional, distant). The feature list is in Supplementary Data S1e. LVI, lympho-vascular invasion; TNBC, triple-negative breast cancer.*
- (b) *The distribution of the PAM50 subtypes for the 300 tumor samples described in (a) across the 86-92 and 08-13 cohorts. The study also included 38 normal breast reduction mammoplasty samples. Within the 08-13 cohort, a set of 88 cases were classified as TNBC by IHC and were analyzed as a separate cohort.*

We further updated the abstract per the reviewer’s request:

“Despite advances in genomic classification of breast cancer, current clinical tests and treatment decisions are commonly based on protein level information. Formalin-fixed paraffin-embedded (FFPE) tissue specimens with extended clinical outcomes are widely available. We performed comprehensive proteomic profiling of 300 FFPE breast cancer surgical specimens, 75 of each PAM50 subtype, from patients diagnosed in 2008-2013 (n=178) and 1986-1992 (n=122) with linked clinical outcomes. These two cohorts were analyzed separately and we quantified 4214 proteins across all 300 samples....”

Ok, good!

11. PAM50 is defined both by RNA and by surrogate IHC markers in the manuscript. However, it is unclear when each definition is used in the manuscript, which makes it confusing to read at times.

PAM50 per definition only refers to RNA not IHC as PAM50 is a RNA-based assay. There is no definition of PAM50 by IHC in the manuscript. We have however now added the word “RNA-based” before the word PAM50 in the section that included IHC data for further clarity.

page #15: “We analyzed 88 IHC defined TNBC cases (profiled by RNA-based PAM50 as: 61 basal-like, 22 Her2-Enriched, and 5 luminal B), all in the 08-13 cohort (Fig. 1b)

Ok

12. The authors use a new method denoted isodoping, with the aim to increase the overlap of identifications between TMT sets. The dynamic range in the orbitrap is max 3 orders of magnitude and the practical with TMT is closer to 2 orders of magnitude. To the pool of

samples, 4.26 pmol of each peptide is added as isodoping. What is the evidence that you have not added 2 orders of magnitude of your spike in peptide compared to the endogenous levels? Adding spike in peptide amounts in excess of 2 orders of magnitude would make the other TMT channels hover around background and lose quantitative accuracy. How is it checked that this don't affect the quantification used? Could the same scenario happen for the SuperMix channel?

The reviewers make an astute point that issues with the dynamic quantification range can arise when implementing TMT. As shown in Supplementary Fig. S2f, when we compared the average S/N ratio, before normalization, across different sample types we detected an average difference of 3.7x between SuperMix and tumor samples, with all SuperMix samples showing an average S/N comparable to the tumor samples with higher signal.

In Supplementary Fig. S2g, it is displayed that there is only a 3.2x difference between the average abundance of isoDoped peptides and endogenous peptides for isodoped proteins in the PIS+isoDoping channel. When comparing the average S/N of the isoDoping peptides in the tumor samples and the spiked in channel we detected an 8.6x difference, below the suggested limit of 20x (Cheung TK et al. "Defining the carrier proteome limit for single-cell proteomics" Nature Methods, 2021).

Ok, I would be curious to see how the TMT profiles compare between isodoped and not isodoped peptides from the same protein.

For the reviewer's request, the figure below shows the correlation between the protein abundance measured by isoDoped peptides only vs protein abundance measured by the endogenous peptides only (for the same proteins). The proteins shown on the plot are the ones from the 4214 set of proteins identified across all the samples for which at least 3 isoDoping and 3 endogenous peptides were included.

Ok, there are some varying correlations but if you have multiple peptides for each protein the quantification should be fine.

13. The isodoping is presented in fig1. This to me indicates that it is one of the main concepts in the paper since if it comes in the first main figure. However, this is a technicality which the authors say that they are preparing a manuscript for and could be moved to supplementary. Per the reviewer's recommendation we have moved the isodoping performance to Supplementary Figure S2.

Supplementary Figure S2

14. It is also unclear how the isodoping peptides were selected. Usually peptides are selected due to their good ionization capabilities which could explain much of the results in fig 1b? Figure 1C is unclear to me. How do you reach 74 isodoping dependent proteins? Can you update the figure legend or make a new clearer figure?

As elaborated on in the methods section, the set of synthetic peptides was selected to fulfill the following criteria: (i) include unique peptides for the protein, and (ii) peptides should be between 6 and 20 amino acids long and/or (iii) have physiochemical properties amenable to MS detection. Our isoDoping methodology has been updated and improved in subsequent experiments for which we have a manuscript under review and can be made available upon request once it is in pre-print. We have also removed Figure 1C from the manuscript.

Ok

15. From results: "The cases in the 08-13 cohort were treated in accordance with contemporary guidelines and contained cases from all four PAM50 subtypes, including all 75 basal-like cases (Supplementary Figs. S1a, S2h, Supplementary Data S1d)."

Which contemporary guidelines are you referring to?

The contemporary guidelines refer to the updated recent guidelines recommended to treat breast cancer commonly used in practice. A reference (Cardoso F et al. Early breast cancer: ESMO Clinical Practice Guidelines for diagnosis, treatment and follow-up. Annals of Oncology 2019) has now been added to support this statement.

Good

16. LVI, lymphovascular invasion is mentioned. Don't find it in materials & methods.

Lymphovascular invasion is found in the methods as part of the survival analysis section. The acronym (LVI) has been added to page #35 as well.

Ok

17. When the tumor groups are defined they are given numbers. However, when they are first introduced in figures (for example 2b & c, 5a, 7a) they are not in numerical order. It would maybe be much easier to follow if the clusters are renumbered in numerical order in the first figure where they appear.

The assignment of numbers of the clusters in figures 2b,2c, 5a and 7a is not random and were not manually chosen, but derived from the consensus clustering algorithm we used. The numbers assigned for each cluster are based on the consensus clustering algorithm output and determined in an unsupervised manner by the ConsensusClusterPlus function. If we were to manually change the numbers in figure 2b to be in a numerical order, we would need to force changing the figure itself to follow that order. This will consequently result in changing the numerical order of the clusters in figure 2c again and the reader would not be able to match the cluster names with the consensus matrices plots present in Supplementary figures S5, S10 and S12. This is described in the consensus clustering algorithm of the ConsensusClusterPlus package where it makes cluster number decisions based on the purity of members in the clusters {Wilkerson MD; ConsensusClusterPlus: a class discovery tool with confidence assessments and item tracking. Bioinformatics 2010}.

The lack of numerical order in multiple figures of clustering is confusing and makes the paper more difficult to read and understand. This will translate into fewer people understanding the paper and thus fewer citations etc..

If you want to make it easier for the reader, you can change the order of the clusters manually and just transfer that order between figures. It can all be done easily by a bioinformatician in the R-code.

Per the reviewer's request, we have manually changed the order of the clusters and transferred that order between figures. Figure 2, Figure 5 and Figure 7 have been updated accordingly.

Perfect!

Figure 2

Figure 5

Figure 7

a

b

18. The identification and quantification of 4214 proteins across all samples is a good result for MS analysis of FFPE samples. But could some of the results be explained by not reaching deep enough into the proteome, considering that there should be around 14000 proteins in a tissue according to ProteinAtlas. Could this be a reason for the grouping of Luminal A tumors with Her2 in fig 2b? In the Krug et al Cell 2020 paper their tumor grouping almost exclusively only mix luminal A and Bs. No HER2 based on 7679 proteins quantified across all 122 samples. Can the lum A mixing with Her2 be reproduced with the same proteins? Or is this an effect of FFPE? The composition of our 08-13 cohort is different from CPTAC as our cohort included only 11 luminal A cases compared to 73 basal-like, 62 Her2-Enriched, and 28 luminal B. The 4 clusters displayed in Fig 2b were the best to segregate this cohort by consensus clustering and thus with only 11 cases, luminal A tumors were not found as a unique cluster, but grouped with clusters 1 and 4 that included luminal B and Her2-Enriched in Fig 2b. In these clusters 1 and 4, luminal B and Her2-Enriched were often intermixed which is a commonly known phenomenon in breast cancer subtyping (Prat, A. et al. Molecular features and survival outcomes of the intrinsic subtypes within HER2-positive breast cancer. JNCI 2014) and is consistent with the proteomics breast cancer data in (Johansson et al. Nat Comm 2019). The cluster membership of our cohort compared to the CPTAC breast cancer cohort was dependent on a different combination of cases and in turn our analysis of the 86-92 with more luminal A cases was more powered showing distinctions of two subgroups within the luminal A subtype including a unique luminal A “stromal enriched” cluster, and a cluster that was more a mix of luminal A and B. Thus, overall our results are driven by the biology and the composition of our 08-13 cohort rather than an artifact or a technical limitation.

Ok

19. In fig S3a, you refer to biological replicates. How is biological replicates defined in clinical samples? For the technical replicates, it would have been better if they were spread out in different TMT sets.

The biological replicates refer to different specimens taken from the same patients. We acknowledge that technical replicates were in the same TMT set.

We have added the definition of biological replicates to the text on page #7-8:

“High reproducibility was observed between the biological replicates (referring to different specimens taken from the same patient) (mean $r=0.71$) and the technical replicates (mean $r=0.88$) (Supplementary Figs. S4a-S4b)”.

Ok

20. The PAM50 subtypes have got standard color code. See TCGA 2012 Nature or Krug et al Cell 2020. To avoid confusion I strongly recommended to use the same color code.

As per the reviewers' request to make it easier for a reader to compare our results with recent breast cancer 'omic studies we changed the colors to match the color code used in Johansson et al and Krug et al.

21. In general the authors make a good job in describing their findings. But to make it easier to follow I would suggest to add ER, PR and HER2 status to fig 2C. For example in the text it says: “Most cases in Cluster-2 and -3 were associated with ER, PR and Her2 negativity by IHC clinical tests, high proliferation index (Ki67), and the “core basal” phenotype (defined as ER-, PR-, Her2- and [EGFR+ or CK5+])²⁹ (Supplementary Table 1).” Adding the clinicopathological markers to the heatmap in fig 2c would make it easy to see this in addition to the table. But this is a matter of taste and you can ignore if you like.

We thank the reviewer for this suggestion. Supplementary Table 1, Supplementary Table 2, Supplementary Data S1e and the “results” section describe and elaborate on the correlation between these clinicopathological variables and clusters. Figure 2c is already rich in information and different types of analysis and so we feel that the main emphasis for readers should be the PAM50 subtype membership in each proteome cluster.

22. In fig 2c, 5a, there is a column called immune with 2 categories, Immune related and Other. How are they defined? Also, for the protein groups there are enrichments, how were the enrichments done? Specify in fig text how the terms were selected, representative/ cutoff?

Immune related proteins were defined based on their protein function involvement in immuneresponse biological processes. Proteins belonging to any of these gene ontology (GO) categories were labeled as Immune:

"GO_DEFENSE_RESPONSE_TO_VIRUS", "GO_RESPONSE_TO_VIRUS",

"GO_RESPONSE_TO_TYPE_I_INTERFERON",
"GO_CELLULAR_RESPONSE_TO_INTERFERON_GAMMA",
"GO_RESPONSE_TO_INTERFERON_GAMMA",
"GO_REGULATION_OF_INNATE_IMMUNE_RESPONSE",
"GO_CYTOKINE_MEDIATED_SIGNALING_PATHWAY",
"GO_ANTIGEN_RECEPTOR_MEDIATED_SIGNALING_PATHWAY",
"GO_IMMUNE_EFFECTOR_PROCESS",
"GO_ACTIVATION_OF_INNATE_IMMUNE_RESPONSE",
"GO_ANTIGEN_PROCESSING_AND_PRESENTATION_OF_PEPTIDE_ANTIGEN_VIA_MHC_CLASS_I",
"GO_FC_EPSILON_RECEPTOR_SIGNALING_PATHWAY",
"GO_POSITIVE_REGULATION_OF_INNATE_IMMUNE_RESPONSE"

For each protein cluster, the most representative terms were selected based on gprofiler enrichment analysis with the following parameters: organism = "hsapiens", ordered_query = FALSE, multi_query = FALSE, significant = TRUE, exclude_iea = TRUE, measure_underrepresentation = FALSE, evcodes = TRUE, user_threshold = 0.05, correction_method = "g_SCS", domain_scope = "annotated", custom_bg = NULL, numeric_ns = "", sources = NULL, term_size < 150 and source in GO:MF, GO:BP or REACTOME' Raudvere, U., Kolberg, L., Kuzmin, I., Arak, T., Adler, P., Peterson, H., & Vilo, J. (2019). Reference: g:Profiler: a web server for functional enrichment analysis and conversions of gene lists (2019 update). *Nucleic Acids Research*, 47(W1), W191–W198. <https://doi.org/10.1093/nar/gkz369>.

The legends of figures 2c and 5a were updated to include this information.

"Immune related is defined based on the protein function as involved in immune-response biological process and for each protein cluster, the most representative terms displayed on the heatmap were selected based on g:profiler4 enrichment analysis".

The methods section page #32 was updated to include information on the terms selected from the enrichment analysis.

For each protein cluster, the most representative terms were selected and presented on heatmaps based on g:profiler77 enrichment analysis with the following parameters: organism = "hsapiens", ordered_query = FALSE, multi_query = FALSE, significant = TRUE, exclude_iea = TRUE, measure_underrepresentation = FALSE, evcodes = TRUE, user_threshold = 0.05, correction_method = "g_SCS", domain_scope = "annotated", custom_bg = NULL, numeric_ns = "", sources = NULL, term_size < 150 and source in GO:MF, GO:BP or REACTOME'.

Ok, good

23. Fig 2a, is this using all or the most varying proteins? In 2b it does not say that the grouping is based on consensus clustering.

UMAP in Fig 2a is based on using all proteins quantified in every sample (4214). The figure legend has been updated accordingly.

The legend of Fig. 2b has been updated to show that the grouping of the different clusters is based on consensus clustering.

(a) Uniform Manifold Approximation and Projection of the 08-13 cohort for the basal-like, luminal A, luminal B, and Her2-Enriched PAM50 subtypes based on all proteins quantified in every samples (4214).

(b) Alluvial plot shows the relationship between PAM50 subtypes and the four proteomic consensus clusters in the 08-13 cohort.

24. In figure S2b-c, the authors show number of peptides per protein. Bit unclear to what it refers to when mentioning peptide? Is that unique peptides? Nr of peptides per protein, is that the per set or total across all TMT sets or mean/median?

It refers to the total number of peptides identified per protein across all TMT sets. The legends for these figures have been updated accordingly.

(a) Percentage of the total number of proteins detected in different number of samples.

(b and c) Number and percentages of proteins identified according to total number of peptides per protein. Yellow bars in the histogram show the number of proteins identified by different numbers of peptides per protein. Blue dots show the percentage of total proteins identified per minimal number of peptides per protein.

In my version of suppl info it reads: Number and percentages of proteins identified according to number of peptides per protein. Not total. Mean nr across TMT sets would be more informative,

since some sets might have many peptides and some might have few peptides.

We have updated this sentence in the Supplementary Fig. S2b-c legends with the correct definition.

“(b and c) Numbers and percentages of the total number of proteins detected in different number of samples according to number of peptides per protein”.

A measure of the plex variability is given in Supplementary Figure S2d, showing the average number of PSM per plex that closely relates to the average number of peptides.

Ok,

25. The number of unique peptides per protein, nr of psm per protein and nr of psms/protein for TMT quantification is missing from the supplementary table with all MS data. Please add this, since it is important when it comes to judging the quantitative robustness. Having said that, must give all the credits for clear clinical information and that the authors include it in the same document so it is easy to access!

We thank the reviewer for this point. As also requested in the reviewer's comment #1, we have added the total number of peptides for each protein, number of unique peptides per protein and number of PSMs used for quantification per protein to the Supplementary Data S1c.

26. In figure 3e, the y-axis says abundance. Is this log2 ratio to the pool of samples? ESR1 is high in cluster 2 which is one of the basal enriched clusters, which is surprising. Could this be due to isodoping or poor quantification? KRT18 and FOXA1 on the other hand behave as expected.

Protein abundance shown is based on a log2 ratio for PSM abundances divided by the relative PIS value in each TMT plex. Then for each protein, the median ratio of the 5 most abundant PSMs was used as relative abundance. This is explained in the methods section page #31 and has been added to the legend of Fig. 3e.

“Protein abundance values are based on log2 ratio for PSMs abundances divided by the relative PIS value in each TMT plex. For each protein, the median ratio of the 5 most abundant PSMs was used as relative abundance”.

The abundance for ESR1 was significantly lower in Cluster-3 than the mean against “all” while ESR1 was non-significantly high in Cluster-2. This could be due to challenges in quantifying ESR1 as endogenous peptides for this protein were only detected in less than 10% of the samples. Using isoDoping, 3 isoDoping peptides for ESR1 were detected in the majority of samples and thus challenges in ESR1 quantification might explain the non-significantly higher levels observed for Cluster-2.

What is the justification for limiting ratio calculation to top 5 most abundant PSMs? Should you not obtain a more robust median with more values (if available)? How is abundance in this case defined?

The unexpected behavior of ESR1 and PGR are concerning. How does the IHC data correlate to the proteomics data?

Since the averaged S/N ratio is directly anti-correlated with the coefficient of variation on repeated measurements, we prioritized the PSMs with the highest S/N ratio in an attempt to reduce the quantification's background noise. Abundance is defined as the signal to noise ratio as reported in the methods section page #31.

We added estrogen and progesterone receptor IHC and proteome measurement comparisons in the new Supplementary Fig. S7; explanations have been included in comment #2 and #8.

Ok

27. PECA is used for calculating p-values. Wonder if that inflates the p-values and makes them smaller just because you have a lot of peptides per protein?

<https://pubs.acs.org/doi/10.1021/acs.jproteome.5b00363>

PECA method leverages the number of peptides per protein to assign higher confidence to proteins with higher peptide coverage. While we agree that this method tends to drive the p-value of certain proteins with a particularly high number of peptides, we find it useful to separate proteins with a small number of peptides since these are the ones with lower confidence in quantification levels. We directly compared PECA performance to another differential expression algorithm (DEqMS, Zhu, Y., Orre, L. M., Zhou Tran, Y., Mermelekas, G., Johansson, H. J., Malyutina, A., Anders, S., & Lehtiö, J. (2020). DEqMS: A Method for Accurate Variance

Estimation in Differential Protein Expression Analysis. *Molecular & Cellular Proteomics*, 19(6), 1047–1057. <https://doi.org/10.1074/mcp.tir119.001646>) on the first differential expression contrast (Cluster1 vs Cluster2-3-4). We found that the two methods give comparable results in terms of calling differentially expressed (DE) proteins (adjusted p-value < 0.05). We found an overall agreement by DE status on 86% of the proteins: 6% of the proteins differentially expressed in PECA and not in DEqMS, 9% of proteins differentially expressed in DEqMS and not PECA, 11% consistently identified as DE in both methods, and 75% consistently identified as not differentially expressed.

While several differential expression analysis methods are routinely used in the proteomics field and their evaluation over multiple types of data and experiments would be of great interest, we believe that a technical evaluation of PECA and/or comparison with other methods are beyond the scope of this paper.

Ok, this is a point that is good to include in the paper since the general breast cancer biologists reading the paper will not be aware of the inflated p-values and may draw wrong conclusions about the data.

We note that while PECA can achieve improved or comparable overall performance with other differential expression methods [PMID:34373457], it tends to estimate particularly low p-values for proteins with a high number of quantified peptides.

Ref: Kalxdorf, M., Müller, T., Stegle, O. et al. IceR improves proteome coverage and data completeness in global and single-cell proteomics. *Nat Commun* 12, 4787 (2021). <https://doi.org/10.1038/s41467-021-25077-6>

To add clarity to the readers, we added this point to the methods section page #31:

“While PECA can achieve improved or comparable overall performance with other differential expression methods [34373457], it can obtain very low p-values for proteins with a high number of quantified peptides”.

Ok,

28. Full credit for uploading the immunohistochemistry slides to <http://www.gpec.ubc.ca/prot>. But why limit to representative images. For the dataset to be useful, all images needs to be available. In addition, there should be an easy way to download all data for image analysis. Our IT team at the Genetic Pathology Evaluation Centre has diligently uploaded all images to <http://www.gpec.ubc.ca/prot>. This information has been updated under the section of “Data availability”, page #35, in the methods.

“Images from immunohistochemistry slides of tissue microarrays used in the study coded as “11-012” and “14-004” are available for public access via the website of Genetic Pathology Evaluation Center (<http://www.gpec.ubc.ca/prot>).

Data image analysis and clinical outcome data for the cases used in this study can be made available through the Genetic Pathology Evaluation Centre and Breast Cancer Outcomes Unit of BC Cancer Centre, upon completion of a Data Transfer Agreement and confirmation of ethical approval for qualified researchers”.

Good

29. To make the data analysis part transparent and reproducible, analysis code should be uploaded to Github or similar repository.

“Code Availability” section has been added to the methods after the “Data Availability” section as requested. Code used for proteomics data analysis is available at GitHub

<https://github.com/glnegri/brca>.

Good, but the code seems to only cover basic functions and processing. You need to add the code for consensus clustering and the figures. Also, make sure that the input data is readily available/pointed to.

The available code takes Proteome Discoverer search output data and performs all the steps for filtering and normalization that result in the data used throughout the manuscript. Clustering, differential expression analysis, survival analysis and all other downstream analyses have been performed with publicly available R packages using the parameters described in the methods section. The level of detail provided is sufficient to reproduce the analysis and is in line with several similar articles published recently by Nature Communications:

Liu, W., Xie, L., He, YH. et al. Large-scale and high-resolution mass spectrometry-based proteomics profiling defines molecular subtypes of esophageal cancer for therapeutic targeting. *Nat Commun* 12, 4961 (2021).

<https://doi.org/10.1038/s41467-021-25202-5>

Franciosa, G., Smits, J.G.A., Minuzzo, S. et al. Proteomics of resistance to Notch1 inhibition in acute lymphoblastic leukemia reveals targetable kinase signatures. *Nat Commun* 12, 2507 (2021). <https://doi.org/10.1038/s41467-021-22787-9>

Satpathy, S., Jaehnig, E.J., Krug, K. et al. Microscaled proteogenomic methods for precision oncology. *Nat Commun* 11, 532 (2020). <https://doi.org/10.1038/s41467-020-14381-2>

Ok, then it is up to the editor if all figures should be reproducible by code in github or not.

30. Orbitrap MS2 data was matched with 0.5 Daltons tolerance. This is a very large window that is usually used for iontrap data. For orbitrap MS2, the tolerance should be around 0.02 Dalton to reduce the risk of miss assigning transitions. Since you are also using methylation of lysine as a variable modification, this in combination with a large tolerance will increase your FDR. You should research at least parts of the data and compare the results to your present results to determine if all data needs to be researched. In relation to this, what is the protein FDR of the dataset, q-value, pep value for each protein?

We thank the reviewer for this point. This was actually a typographical error in the text; the data were in fact searched with 0.05 Da tolerance. We have updated the "methods" page #30 accordingly. The full parameters used for the Proteome Discoverer search, together with the results output are available at the PRIDE repository with the dataset identifier PXD024322.

Ok

31. From results: FFPE samples were macro-dissected from 3-6 sections to obtain >80% tumor content and analyzed using the SP3-CTP multiplex MS proteomics protocol²⁴ (Supplementary Fig. S1b).

Should it not be ref 19 instead of 24?

Reference #24 {Hughes, C.S., et al. Single-pot, solid-phase-enhanced sample preparation for proteomics experiments. *Nat Protoc* 14, 68-85 (2019)} is a more detailed and up-to-date protocol for the methods used in this study when compared to Reference #19 {Hughes, C.S., et al. Quantitative Profiling of Single Formalin Fixed Tumour Sections: proteomics for translational research. *Sci Rep* 6, 34949 (2016)}. Given that Reference #19 included work done on FFPE (in ovarian cancer) we now include both references #19 and #24 to support our statement.

Ok

New questions based on the updated manuscript

32. In the discussion the authors write: Furthermore, the 9088 total proteins identified is comparable to that achieved using fresh frozen materials (10,107 proteins) by the CPTAC breast cancer project¹⁵. The 4214 proteins quantified in every sample (n=342) across a large-scale breast cancer project using minimal tissue demonstrates the efficiency and high sensitivity of the SP3-CTP approach for FFPE cancer proteomics studies³⁶⁻³⁹.

The total number of identifications are easily achievable in MS based proteomics. The difficulties are to achieve good overlap in quantification across samples. To make it a fair comparison you should also include the nr of overlapping proteins with quantification. You have also used Johansson et al dataset and should add that also in the comparison.

Given that our study quantified 4214 proteins across a high number (n= 342) of FFPE samples, we did not compare the coverage across all samples to CPTAC and Johansson et al as those studies have a significantly smaller number of cases and are from fresh frozen samples. Given that the reviewer has mentioned that the total number of proteins identified is easily achievable, we have simply removed this sentence from our discussion to avoid any confusion.

Ok,

33. In the discussion the authors write: Our result is consistent with a proteomic profiling study of 2 "basal immune hot" cases vs. 7 "basal immune cold" cases using >10mg of frozen tissue¹⁶. For max TMT labelling, 100 µg of peptides are used. This usually equates to 1-2 mg wet weight of tissue. Not >10 mg. where in paper 16 did you find this statement?

We thank the reviewer for this point. We reviewed the Johansson et al and we did not find that it used >10 mg. It was actually a mistake in the reference added to this sentence as the CPTAC paper is the one that mentioned the use of 200 mg of material for analysis in their methods:

In Krug 2020 Cell the methods state: "Samples were qualified for the study if two or more tumor tissue core biopsies or surgical resection segments had a minimum mass of 200 mg and demonstrated greater than 60% tumor cell nuclei and less than 20% tumor necrosis on frozen tissue section review".

We decided it is best to remove this sentence from the discussion, as follows (page #19):

"Our result is consistent with a proteomic profiling study of 2 "basal immune hot" cases vs. 7 "basal immune cold" cases using >10mg of frozen tissue¹⁶".

Ok,

34. Many of the supplementary figures are a bite blurry, which needs to be fixed.

The Supplementary Figures in the PDF files have been fixed and seem to be clear now. Hopefully, these will look OK to the reviewer when merged in the *Nat. Commun* journal submission system. All figure files used for the final publication will be in the Ai format for high resolution rendering as per *Nat Commun* requirements.

35. Check panel labeling in fig S5. The word robust is mentioned multiple times here and in the paper in general. What do the authors mean by robust? What is the criteria(s) that needs to be meet to be called robust?

The word "robust" has been mentioned multiple times when describing results based on the inspection of consensus matrix and delta plots examining the change in consensus cumulative distribution function (CDF) area to assign the number of consensus clusters to be used in the analysis.

Based on the ConsensusClusterPlus package in R, Consensus Clustering provides quantitative and visual 'stability' evidence **derived from repeated subsampling and clustering**. The Consensus Clustering reports a consensus of these repetitions (multiple iterations), which is **robust** relative to the resampling variability. Thus, the word "robust" has been used in this regard.

Nevertheless, to avoid over-use and less specific uses of the term, the word "robust" page #8 has been removed:

"An overview clustering of all the samples included in our study showed that the 38 SuperMix replicates had the highest correlation across the 38 plexes (range 0.68-0.81) when compared to the breast tumors and normal samples. (Supplementary Figs. S4c-S4d)".

Other "robust" wordings in a context other than the bioinformatic consensus cluster assignments as described above have been removed from the manuscript as follows:

Discussion, page #20:

"Thus, our outcome-linked proteomic data could aid the development of ~~robust~~ protein biomarkers for clinical tests to distinguish TNBC/basal-like patients with favorable versus poor prognosis that may benefit from therapies beyond standard chemotherapies".

Discussion, page #24:

"The findings on immune distinctions, ECM, and lipid metabolism pathways are potentially clinically relevant as standard clinical tests do not yet interrogate this level of heterogeneity for breast cancer subtyping. Furthermore, this study identifies protein candidates for in-depth analysis of existing archived clinical trial FFPE specimens, providing a valuable resource to develop ~~robust~~ diagnostic and prognostic biomarkers in breast cancer".

Ok,

Reviewer #2, expert in bioinformatics and subtype classification (Remarks to the Author):

In this study, the authors carry out mass-spec proteomic profiling of 300 FFPE breast cancer surgical specimens. The specimens are separated into two cohorts based on batch effects. The 08-13 cohort included 75 basal-like, 62 Her2-Enriched, 30 luminal B, and 11 luminal A PAM50 defined cases. The 86-92 cohort provided the long-term outcome data required for luminal cases and included 64 luminal A, 45 luminal B, and 13 Her2-Enriched PAM50 cases. The 08-13 cohort was used for subtype discovery, both across all tumors and within the TNBC subset. ER+ subtypes examined in the 08-13 cohort were examined in the 86-92 cohort.

Specific comments:

1. Batch effects were found between the 08-13 and 86-92 cohorts, likely due to differences in collection techniques, pre-analytical handling, and fixation procedures. Could the authors try to harmonize the two datasets using Combat (<https://rdrr.io/bioc/sva/man/ComBat.html>)? In practice, Combat is very good at removing batch effect differences. Data from different platforms (RPPA, RNA-seq, DNA methylation) have been successfully processed with Combat, and the method is independent of nature of the batch effect. The PAM50 subtype could be used as the experimental group. There would be advantages in having one harmonized dataset of 300 samples. It seems worth a try. As currently written, the Abstract suggests that there is one dataset that was analyzed, rather than two separate cohorts.

As has been shown before, ComBat can lead to overestimating ratios and many in the field believe should be avoided. (Methods that remove batch effects while retaining group differences may lead to exaggerated confidence in downstream analyses <https://academic.oup.com/biostatistics/article/17/1/29/1744261>), especially considering that the batch effect observed in our study is mostly driven by missed identification of peptides cleaved at lysines and not by artifacts on quantification, as shown in figures S3c and S3d. Furthermore, some of the subtypes are completely (basal-like) or almost completely (luminal A) confounded with the 'cohort' batch effect. While Combat will always transform the data to minimize batch differences, we believe that for the reasons above, its application in this dataset would lead to serious artifacts in the data.

We would also like to note that the decision to include cases from the 86-92 cohort in our study design was based on clinical and translational considerations. In order for analysis to be meaningful for luminal cases, a long enough follow-up was necessary to obtain sufficient events for outcome analyses. Thus, the majority of luminal PAM50 cases were derived from patients diagnosed with invasive breast cancer in the period January 1986 to September 1992. Forcing the two cohorts to be lumped together for subtyping does not allow obtaining clinically-relevant results for the subtypes found, and could compromise any clinical relevant observations.

We have updated the abstract to highlight that for the 300 cases included there were 2 datasets analyzed rather than one.

"We performed comprehensive proteomic profiling of 300 FFPE breast cancer surgical specimens, 75 of each PAM50 subtype, from patients diagnosed in 2008-2013 (n=178) and 1986-1992 (n=122) with linked clinical outcomes".

2. Page 8: "Cluster-1 (n=34) consisted mostly of luminal B and Her2-Enriched PAM50 cases. Clusters-2 (n=50) was enriched for basal-like subtype, included few Her2-Enriched, but had no luminal cases. Cluster-3 (n=47) was primarily basal-like cases but included Her2-Enriched cases. Cluster-4 (n=43) was mostly Her2-Enriched but included luminal A and luminal B cases." It seems that actual numbers to reflect the noted associations would be helpful here, e.g. exactly how many basal-like cases and Her2 cases were in Cluster-3, and was Cluster-2 SIGNIFICANTLY enriched for basal-like.

Cluster-2 is enriched for basal-like (pval<1.16e-11, Fisher's test), Cluster-3 is enriched for basal-like (pval<1.3e-4, Fisher's test), Cluster-4 is enriched for Her2-Enriched (pval<1.9e-4, Fisher's test).

The numbers reflecting the breakdown for each PAM50 subtype within each proteome cluster as they appear in Fig. 2b have also been added to the text, page #9.

"Cluster-1 (n=34) consisted mostly of luminal B (n=18) and Her2-Enriched (n=13) PAM50 cases. Clusters-2 (n=50) was significantly enriched for basal-like subtype (n=41), included few Her2-Enriched, but had no luminal cases (p-value<1.16e-11, Fisher's test). Cluster-3 (n=47) was primarily basal-like cases (n=31) but included Her2-Enriched cases (n=14) (p-value<1.3e-4, Fisher's test). Cluster-4 (n=43) was mostly Her2-Enriched (n=26) but included luminal A (n=8) and luminal B (n=8) cases (p-value<1.9e-4, Fisher's test)".

3. In general, where the word "significantly" appears in the main text, it would be good to include a p-value and associated test to support the claim. The figures referred to likely include the test, but reflecting this in the main text as well would be helpful to the reader. For example, page 11: "The immune hot cluster also had significantly higher CD8+ TILs in the intratumoral compartment compared to other clusters (Fig. 4a)." by what p-value and test?

The p-values and tests are now updated across the text where the word "significantly" appears.

4. Wherever a p-value appears in the main text, the test used to derive that p-value should also be indicated. For example, page 12: "The subgroups with a high expression for only one of these biomarkers were characterized with intermediate RFS (Supplementary Fig. S5b). 70% (21/30) of the cases classified as (TAP1 high/HLA-DQA1 high) were in Cluster-3, while 90% (76/84) of (TAP1 low/HLA-DQA1 low) cases were in other clusters (p-value<0.00001) (Supplementary Table 1)." What test was used here (we can save the reader from having to go the Table for the answer)?

The test used was the Chi-square test. The text in page #13 has been updated to include this information.

"70% (21/30) of the cases classified as (TAP1 high/HLA-DQA1 high) were in Cluster-3, while 90% (76/84) of (TAP1 low/HLA-DQA1 low) cases were in other clusters (Chi-square p-value<0.00001) (Supplementary Table 1)".

5. Page 15: "Multiple correction testing identified fatty acid-binding protein-7 (FABP7) as a candidate biomarker most significantly associated with >10-year RFS on tamoxifen treatment..." Was this the only protein that was significant? Were other proteins significant and using what statistical test and cutoff?

The association between the continuous increase in each individual protein identified in the cohort 86-92 and the endpoint of 10-years RFS was tested using a Cox regression model and stratified log-rank test. This analysis is displayed in Supplementary Data S4f. Only protein biomarkers that had a significant log-rank p-value <0.05 when adjusted for multiplicity testing by the Benjamini-Hochberg test were selected. **Only** FABP7 protein was found to meet these criteria as displayed in Supplementary Data S4f.

The relevant text for the 86-92 analysis page #18 has been updated to include this information.

"Multiple correction testing identified fatty acid-binding protein-7 (FABP7) as the only candidate biomarker associated with >10-year RFS on tamoxifen treatment (log-rank BHadj p=0.00004) (Supplementary Data S4f, Supplementary Fig. S12e)".

6. Discussion, page 16. Many journals are uncomfortable with the phrase "(manuscript in preparation)." It seems that the method indicated should be described in sufficient detail in the Methods, if it isn't already.

We believe that the methods regarding the isoDoping methodology are now described in sufficient detail in the methods section of this manuscript for the reader to be able to reproduce the experiment as was intended. While we are currently preparing an even more detailed and comprehensive description of the general isoDoping strategy for a separate primary methodology-oriented publication, to avoid confusion we have deleted the mention of a "manuscript in preparation." From pages #6 and #18.

7. In addition to making the raw data available on ProteomeXchange, it would be most helpful to include the processed proteinXsample tables as Supplementary Data with the published paper. CPTAC has done a similar thing with their past publications.

The proteinXsample data are included in the original Supplementary Data S1c. As requested by reviewer #1, we have also added the peptides identified across the cohort to the Supplementary data S1 along with the total number of unique peptides per protein and number of PSMs used in quantification per protein (Supplementary Data S1c-S1d).

8. For boxplots in the figures, please define the ranges involved.

Boxplot whiskers range extends to the most extreme data point which is no more than 1.5 times the interquartile range from the box. This definition has been added to the legends of Fig. 3e and Fig. 7b.

Reviewer #3, expert in breast cancer subtypes (Remarks to the Author):

1. The authors present their previously described highly sensitive MS-based methodology termed "Single-Pot, Solid-Phase enhanced, Sample Preparation"-Clinical Tissue Proteomics (SP3-CTP). This technology has been shown to capture known and novel features in FFPE tumor samples. The authors have previously shown that this method can be applied on large FFPE material cohorts linked to outcome data. Comprehensive quantification of protein expression can be achieved even from lower input quantities of patient specimens such as small biopsies. Here is would have been useful to know how small?

This is described in the methods section and supplementary Figure S1a. One to six unstained 10µm tissue sections were cut for each sample to obtain an aggregate total area of ~1cm x 1cm x 10µm, with >80% tumor content.

2. In this paper they have applied the method to 300 well-characterized archival FFPE breast cancer specimens in terms of clinical outcome, IHC, and PAM50 RNA-based intrinsic subtypes. The authors demonstrate that at the protein level one can identify groups characterized by high expression of immune-response proteins and favorable clinical outcomes.

Does this paper bring a sufficient novelty? While it is true that "classifications do not always guide therapeutic choices, due to the extensive heterogeneity that still characterizes breast cancers" can this be solved by adding one more, at the level of proteomics?

As described in the introduction, we performed the current study because genomic classifications of breast cancer are inherently limited as clinical decisions are generally based on the protein level. The underlying technology's

application to FFPE breast cancer material is novel. To the extent that some of the findings overlap with genomic classifications, our study still provides an important verification at the protein level, where most drugs act.

Q1. How do this extension to 300 cases add to what we know from Johansson et al. *Nat Comm*, 2019?

As highlighted in the introduction, Johansson et al. *Nat Comm* 2019 only profiled 9 tumor samples from each of the four main breast cancer PAM50 subtypes, a set which also lacked clinical outcome associations and was insufficient to characterize the biological heterogeneity of breast cancers in relation to clinical behavior and treatment response. In addition, their work required fresh-frozen tissues that are not routinely available from patients, unlike the FFPE clinical specimens we were able to use that can be accessed in larger numbers allowing meaningfully powered linkages to clinical outcomes.

Q2. How does the heterogeneity described here match what is known from RNA based classification (basal also divided in several immune clusters)

The PAM50 subtypes used in this study are an RNA-based classification and the associations of each proteome cluster membership with each PAM50 subtype are described in detail in the manuscript. Within the basal-like RNA-based subtype, there are two distinct proteomic groups that differ in immune response. In the results section, we describe how the heterogeneity of triple negative breast cancer relates to what is known from RNA-based classifications by comparing our findings with those by Burstein M et al. *CCR* 2015, showing that our triple negative clusters were highly correlated with their corresponding RNA subtypes of 'luminal-androgen receptor', 'mesenchymal', 'basal-immune suppressed' and 'basal-immune activated'.

Q3. If the authors were to make biomarkers based on protein as they suggest, which ones would they chose?

TAP1 and HLA-DQA1, as described in detail in the results and discussion sections. These choices are further supported by the supplementary validation work done in response to reviewer #1, comment #8 as described above (based on the data shown in Supplementary Figures S7 and the new figure S8). We do note that TAP1 and HLA-DQA1 were chosen, in part, because of the availability of quality IHC grade antibodies; it remains possible that other proteins may perform better on IHC-based tests when quality antibodies are available. Indeed, this is one of the prime utilities of our results for the breast cancer community, to spur additional biomarker research using our data.

The discussion page #21 has been updated with this information.

"Other proteins elevated in the immune hot cluster with available quality antibodies could also be used and developed as candidate biomarkers".

Q4. An introduction of 5 pages and large number of references (81) makes it into a difficult read. This paper as rigorously performed and described, would benefit from some clarity and simplification, just highlighting the results that move the field forward.

The original work was written in a way that fits the requirements of *Nature Communications*. The introduction here is 2.5 pages double spaced rather than 5 pages as pointed out by the reviewer and the authors hold that this is adequate to succinctly review the pertinent literature, making it hard to remove any essential information from the introduction. As this research sits at a crossroads of breast cancer, bioinformatics, and analytical chemistry the authors believe it is important to provide key background information for scientists from a breadth of related and interested fields to fully appreciate the work. 84 references are merely supporting information for the interested reader to pursue, a number that complies with the *Nature Communications* guidelines (and we are aware of several detailed and comprehensive publications in *Nature Communications* that have a similar or even higher number of references used to properly cover the scientific data presented).